# DRL: Decomposed Representation Learning for Tabular Anomaly Detection

**Hangting Ye[1], He Zhao[2], Wei Fan[3], Mingyuan Zhou[4], Dandan Guo[1]\*, Yi Chang[1 5 6]\***

School of Artificial Intelligence, Jilin University[1] CSIRO's Data61[2] University of Oxford[3]
The University of Texas at Austin[4] International Center of Future Science, Jilin University[5]
Engineering Research Center of Knowledge-Driven Human-Machine Intelligence, MOE, China[6]
`yeht2118@mails.jlu.edu.cn`, `he.zhao@data61.csiro.au`, `wei.fan@wrh.ox.ac.uk`,
`mingyuan.zhou@mccombs.utexas.edu`, `{guodandan,yichang}@jlu.edu.cn`

## Abstract

Anomaly detection, indicating to identify the anomalies that significantly deviate from the majority normal instances of data, has been an important role in machine learning and related applications. Despite the significant success achieved in anomaly detection on image and text data, the accurate Tabular Anomaly Detection (TAD) has still been hindered due to the lack of clear prior structure information in the tabular data. Most state-of-the-art TAD studies are along the line of reconstruction, which first reconstruct training data and then use reconstruction errors to decide anomalies; however, reconstruction on training data can still hardly distinguish anomalies due to the data entanglement. To address this problem, in this paper, we propose a novel approach Decomposed Representation Learning (DRL), to re-map data into a tailor-designed constrained space, in order to capture the underlying shared patterns of normal samples and differ anomalous patterns for TAD. Specifically, we enforce the representation of each normal sample in the latent space to be decomposed into a weighted linear combination of randomly generated orthogonal basis vectors, where these basis vectors are both data-free and training-free. Furthermore, we enhance the discriminative capability between normal and anomalous patterns in the latent space by introducing a novel constraint that amplifies the discrepancy between these two categories, supported by theoretical analysis. Finally, extensive experiments on 40 tabular datasets and 16 competing tabular anomaly detection algorithms show that our method achieves state-of-the-art performance.

## 1 Introduction

Anomaly detection (AD), which aims to identify the anomalies that significantly deviate from the majority normal instances, is a crucial machine learning task. Tabular data, usually represented as vectors of heterogeneous features, is a vital data type in AD (Han et al., 2022; Yin et al., 2024) with numerous applications, including cyber-security (Ahmad et al., 2021), rare disease diagnosis (Fernando et al., 2021; Ye et al., 2023a) and financial fraud detection (Al-Hashedi & Magalingam, 2021). In most real-world scenarios, labeled anomalies usually need to be manually annotated by domain experts, which is both expensive and time-consuming, and marking all types of anomalies is usually impractical (Chandola et al., 2009; Ye et al., 2023b; Guo et al., 2023). Therefore, tabular anomaly detection (TAD) is usually implemented in a one-class classification setting, where a model is trained to identify anomalies in the absence of labeled anomalous samples, relying solely on the normal instances available during training (Schölkopf et al., 1999; Sohn et al., 2021; Yin et al., 2024). This approach fundamentally differs from supervised learning tasks, which have access to examples from all data classes.

Existing approaches in this literature leverage the assumption that anomalies deviate from characteristic patterns present in the normal training data. By capturing these normal patterns during training, one can identify anomalies at the inference time. When considering perceptual data (e.g., image

---

*Corresponding authors.

and text), many methods (Golan & El-Yaniv, 2018; Liznerski et al., 2020) have demonstrated significant success by leveraging the structure of the input data. For example, images can be rotated, and the ability to distinguish between different rotations varies between anomalies and normal samples (Golan & El-Yaniv, 2018). However, unlike image or text, tabular data features can have values with different types (numerical and categorical), ranges and distributions. Thus, there is no prior information on the structure of tabular data (Borisov et al., 2022; Shenkar & Wolf, 2022; Chang et al., 2023; Thimonier et al., 2024; Ye et al., 2024). One of the general ideas for TAD is to train deep learning models that reconstruct training data and use reconstruction error to decide whether a test datum is normal or anomalous following the assumption that a model can accurately reconstruct normal samples while failing to reconstruct anomalous ones. Methods (Yin et al., 2024; Thimonier et al., 2024) built on top of this idea have shown promising performance.

Although fruitful progress has been made in the last several years, capturing the comprehensive normal patterns that are distinct from anomalous patterns for tabular data remains a challenging task, as real-world data may exhibit data entanglement between normal and anomalous samples. This paper develops DRL, a decomposed representation learning framework for TAD that learns the underlying normal patterns shared among normal samples by re-mapping observations into a tailor-designed constrained latent space, where normal and anomalous patterns are more effectively distinguished. To learn the shared information that distinguishes normal samples from anomalies, we enforce each normal sample's representation in the latent space to be decomposed into a linear combination of randomly generated orthogonal basis vectors with sample-specific weights. Notably, these basis vectors are fixed during training and are data-free. In addition, to enhance the discriminative power between normal and anomalous patterns within the latent space, we introduce a novel constraint that amplifies the discrepancy between these two patterns, supported by theoretical analysis.

The contributions of this paper include: **1)** Inspired by the effectiveness and popularity of reconstruction-based approaches in TAD, we propose a novel method that performs representation decomposition in a constrained latent space, effectively addressing the potential data entanglement issue between normal and anomalous samples, resulting in significant improvements over existing methods. **2)** We propose a decomposed representation learning framework to enforce that each normal representation in the latent space could be decomposed into a linear combination of shared orthogonal basis vectors with sample-specific weights, thereby capturing the underlying unique normal patterns effectively. Importantly, the choice of basis vectors is data-free and the basis vectors themselves are training-free. **3)** Additionally, we develop a simple and theoretically sound constraint to further amplify the discrepancy between normal and anomalous patterns within the latent space. **4)** We conduct extensive experiments on 40 TAD datasets and 16 competing benchmark algorithms, and report comprehensive results along with analysis and visualizations. These experiments show that our method achieves state-of-the-art performance.

## 2 PRELIMINARIES

**Problem Formulation.** Following previous works (Yin et al., 2024; Thimonier et al., 2024), we implement AD in a one-class classification setting, i.e., only normal samples are available during training. We denote the training set as $\mathcal{D}_{train} = \{\mathbf{x}_i\}_{i=1}^N$, where $\mathbf{x}_i \in \mathcal{X} \subseteq \mathbb{R}^D$, $N$ is the number of normal samples in the training set and $D$ is the number of input features. The test set $\mathcal{D}_{test}$ includes both anomalous and normal samples. The goal of AD is to learn a detection model from $\mathcal{D}_{train}$. During inference, the detection model takes sample $\mathbf{x} \in \mathcal{D}_{test}$ as input and outputs the predicted anomaly score of $\mathbf{x}$, where a higher score indicates a higher confidence that $\mathbf{x}$ is an anomaly.

**Reconstruction-based Approaches.** These methods (Yin et al., 2024; Thimonier et al., 2024) have demonstrated promising efficacy in tabular AD by training models to accurately reconstruct normal samples while failing to reconstruct anomalous ones. Standard reconstruction-based approaches consider learning a mapping $A(\cdot; \Theta) : \mathbb{R}^D \to \mathbb{R}^D$ to minimize the reconstruction loss within $\mathcal{D}_{train}$. Typically, $A(\cdot; \Theta)$ first maps the sample from observation space to latent space, and then maps it back to the observation space to obtain the reconstruction of the sample. The $\Theta$ can be optimized by minimizing the reconstruction loss on normal training samples:

$$\min_\Theta \frac{1}{N} \sum_{i=1}^N d(\mathbf{x}_i, \tilde{\mathbf{x}}_i) = \min_\Theta \frac{1}{N} \sum_{i=1}^N d(\mathbf{x}_i, A(\mathbf{x}_i; \Theta)), \tag{1}$$

Figure 1: T-SNE visualization for original data space and deep TAD models' latent space on real-world datasets. (a-d) are on the Abalone dataset and (e-h) are on the Cardiotocography dataset. "blue" and "orange" points indicate normal and anomalous samples respectively. (a) and (e) show that the normal and anomalous samples are entangled in the observed data space. (b) and (f) show the prediction for observations by MCM. In (c) and (g), which depict the representations learned by MCM, we observe a notable overlap between normal and anomalous representations in the latent space. In contrast, (d) and (h) demonstrate that our proposed DRL achieves a more discriminative separation for normal and anomalous representations in the latent space.

where $d(\mathbf{x}_i, \tilde{\mathbf{x}}_i)$ measures the reconstruction loss for $\mathbf{x}_i$, which is often set to be a distance measurement such as the Euclidean distance. The reconstruction loss is typically employed as the anomaly score to detect anomalies. Recently, MCM (Yin et al., 2024) introduces a learnable masking strategy to the input and aims to reconstruct normal samples well with access to only unmasked entries of the input, where generating the effective masks is still challenging in this field. Motivated by the importance of incorporating both feature-feature and sample-sample dependencies in tabular learning (Kossen et al., 2021; Gorishniy et al., 2024), NPT-AD (Thimonier et al., 2024) trains Non-Parametric Transformers (Kossen et al., 2021) to reconstruct masked features of normal samples by utilizing the whole training set. However, NPT-AD involves a high computational cost in terms of memory and time, due to its reliance on the training set during inference. Both NPT and MCM rely on the reconstruction about the samples, which usually suffer from the potential data entanglement in observation space, as shown in Fig. 1 and Fig. 7 of Appendix A.1.

## 3 METHOD

### 3.1 MOTIVATION

Despite the effectiveness of aforementioned works based on their specially designed masks and reconstruction strategies in the observed data space, the issue of potential data entanglement remains unaddressed, which may impede the ability for TAD. As illustrated in (a) and (e) of Fig. 1, real-world data may exhibit observation entanglement between normal and anomalous samples. Our empirical observations indicate that this phenomenon also exists for many other real world tabular datasets (see Fig. 7 of Appendix A.1). (b) and (f) of Fig. 1 indicate that while MCM (Yin et al., 2024), one of the best-performing methods, demonstrates promising performance in detecting anomalies, some mistakes caused by data entanglement persist. Ignoring observation entanglement in TAD under the one-class classification setting can lead to diminished discriminative power between learned normal and anomalous patterns, as the overlap between normal and anomalous representations within the latent space of deep models may obscure the distinction between them, which is illustrated in (c) and (g) of Fig. 1. We attribute this challenge to the intrinsic heterogeneity of features in tabular data, which aligns with recent findings (Grinsztajn et al., 2022) indicating that neural networks struggle to learn irregular patterns, particularly when confronted with numerous uninformative features present in tabular data. That is to say, using the reconstruction loss about the observed samples to distinguish the anomalous samples from the normal samples might be insufficient.

### 3.2 PROPOSED METHOD

To address the above-mentioned challenges, we focus on learning the underlying unique normal patterns from the representation $\mathbf{h} = f(\mathbf{x}; \theta_f)$ produced by a feature extractor $f(\cdot; \theta_f) : \mathbb{R}^D \to \mathbb{R}^E$ and performing representation decomposition in the constrained latent space, where the normal and anomalous patterns are more discriminative, rather than from the raw observations directly. In the following, we will elaborate on the specific learning procedure to capture the unique patterns of normal representations in Section 3.2.1; then, we will provide a constraint to further enhance the discriminative power between normal and anomalous patterns within the latent space in Section 3.2.2; we also give the overall implementation of the proposed method in Section 3.2.3. As shown in (d) and (h) of Fig. 1, with our proposed method, the representations of normal and anomalous samples in the latent space are discriminative. An overview of the proposed framework is depicted in Fig. 2.

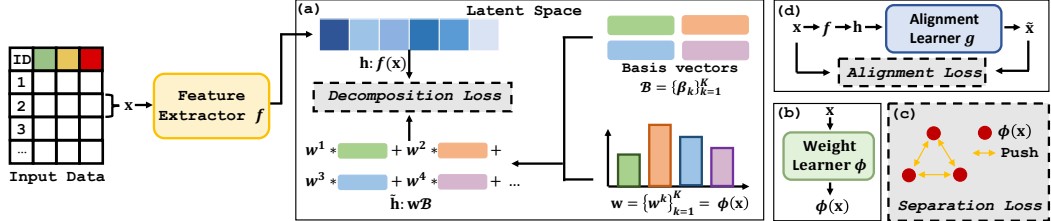

Figure 2: The DRL framework. During training, in step (a), to capture the unique patterns of normal representations in the latent space, the representation $\mathbf{h} = f(\mathbf{x}; \theta_f)$ for each normal sample is enforced by decomposition loss (Eq. 3) to be decomposed into a linear combination of shared orthogonal basis vectors $\mathcal{B} = \{\boldsymbol{\beta}_k\}_{k=1}^K$ with sample-specific weights $\mathbf{w}$, where $\mathbf{w}$ is computed by weight learner $\phi(\mathbf{x}; \theta_\phi)$ in step (b); in step (c), a separation constraint (Eq. 4) is introduced to further enhance the discriminative power between normal and anomalous patterns within the latent space; in step (d), we incorporate alignment loss (Eq. 5) to preserve intrinsic feature correlation of $\mathbf{x}$ motivated by standard reconstruction-based method. During inference, $d(\mathbf{h}, \tilde{\mathbf{h}})$ derived from the decomposition loss (Eq. 3) optimized during training in step (a) is used as the anomaly score.

### 3.2.1 DECOMPOSED REPRESENTATION LEARNING IN LATENT SPACE

Normal samples, which are drawn from the same distribution, are considered to represent the "normal" state. Thus, it is reasonable to assume that these samples share common statistical information that distinguishes them from anomalies. Inspired by techniques from dictionary learning, topic modeling, and matrix factorization (Tošić & Frossard, 2011; Vayansky & Kumar, 2020), we can learn the shared information by enforcing that each normal sample's representation is decomposed into a linear combination of shared basis vectors (analogous to "topics" in topic modeling) with sample-specific weights (analogous to "topic proportion" in topic modeling). Unlike traditional dictionary learning or topic modeling mainly performed in the data space, our approach assumes that the representation of each normal sample in the latent space can be effectively modeled as a mixture of fixed basis vectors with specific mixture proportions.

To accurately capture the statistical characteristics of normal samples while distinguishing them from anomalies, it is crucial that the shared basis vectors are sufficiently diverse to encapsulate the global structure of the normal data. To this end, we eliminate the dependencies among basis vectors by leveraging a set of orthogonal vectors as basis vectors $\mathcal{B} = \{\boldsymbol{\beta}_k\}_{k=1}^K \in \mathbb{R}^{K \times E}$, where $\boldsymbol{\beta}_k \in \mathbb{R}^E$ denotes the $k$-th basis vector and $K$ denotes the number of basis vectors with $K < E$. These basis vectors are derived using the classical Gram–Schmidt process (Leon et al., 2013), which constructs an orthogonal sequence $\{\boldsymbol{\beta}_k\}_{k=1}^K$ from a linearly independent sequence $\{\mathbf{q}_k\}_{k=1}^K \in \mathbb{R}^{K \times E}$ by defining $\boldsymbol{\beta}_k$ inductively as follows:

$$\boldsymbol{\beta}_1 = \mathbf{q}_1, \quad \boldsymbol{\beta}_k = \mathbf{q}_k - \sum_{i=1}^{k-1} \frac{\boldsymbol{\beta}_i \cdot \mathbf{q}_k}{\|\boldsymbol{\beta}_i\|^2} \boldsymbol{\beta}_i, k \geq 2, \tag{2}$$

where $\boldsymbol{\beta}_i \cdot \mathbf{q}_k$ denotes the dot product. In practice, we randomly sample $\{\mathbf{q}_k\}_{k=1}^K$ from the standard Gaussian distribution and then obtain $K$ orthogonal vectors as basis vectors by Eq. 2, which are fixed during the whole training procedure. Additionally, experimental results demonstrate that ours is robust and alternative orthogonalization methods can also effectively build basis vectors.

Our objective is to decompose each normal sample representation $\mathbf{h} = f(\mathbf{x}; \theta_f)$ into a linear combination of orthogonal basis vectors $\mathcal{B} = \{\boldsymbol{\beta}_k\}_{k=1}^K$, weighted by a sample-specific vector $\mathbf{w} = \{w^k\}_{k=1}^K \in \sum^K$, where $w^k$ denotes the weight associated with the $k$-th basis vector, and $\sum^K$ denotes the probability simplex in $\mathbb{R}^K$. Considering that the weights $\mathbf{w}$ should be sample-specific to encapsulate the unique representation information in the latent space for an observed sample $\mathbf{x}$, we apply a weight learner $\phi(\mathbf{x}; \theta_\phi) : \mathbb{R}^D \to \mathbb{R}^K$ to explicitly compute the corresponding weight vector $\mathbf{w}$. Consequently, a normal sample representation $\mathbf{h}$ that can be successfully decomposed in the latent space is expressed as $\tilde{\mathbf{h}} = \sum_{k=1}^K w^k \boldsymbol{\beta}_k$, where $\tilde{\mathbf{h}}$ serves as the reconstructed version of $\mathbf{h}$ in the latent space. To ensure that normal sample representations are accurately decomposed using the orthogonal basis vectors, we formulate the optimization problem to minimize the discrepancy

between $\mathbf{h}$ and $\tilde{\mathbf{h}}$ as follows:

$$
\begin{aligned}
\min_{\theta_f, \theta_\phi} \mathcal{L}_{decomposition} &= \min_{\theta_f, \theta_\phi} \frac{1}{N} \sum_{i=1}^{N} d(\mathbf{h}_i, \tilde{\mathbf{h}}_i) = \min_{\theta_f, \theta_\phi} \frac{1}{N} \sum_{i=1}^{N} d(f(\mathbf{x}_i; \theta_f), \sum_{k=1}^{K} w_i^k \boldsymbol{\beta}_k) \\
&= \min_{\theta_f, \theta_\phi} \frac{1}{N} \sum_{i=1}^{N} d(f(\mathbf{x}_i; \theta_f), \phi(\mathbf{x}_i; \theta_\phi) \mathcal{B}),
\end{aligned}
\tag{3}
$$

where $N$ is the number of training samples, $d(\cdot, \cdot)$ is the distance measurement and $\{w_i^k\}_{k=1}^{K} = \phi(\mathbf{x}_i; \theta_\phi)$. By ensuring that each normal training sample's representation is mapped to a region where it can be expressed as a linear combination of fixed orthogonal basis vectors, with the associate weights explicitly computed, this approach effectively models the shared statistical information of normal patterns within the latent space. This shared information encompasses two perspectives: (i) by utilizing all features as input, we capture the inherent interactions among features, thereby acquiring comprehensive feature-feature dependencies among normal training samples, and (ii) each training sample's representation is decomposed into a set of shared basis vectors with sample-specific weights, thereby optimizing the model to capture sample-sample dependencies among normal training samples.

### 3.2.2 CONSTRAINT: REPRESENTATION SEPARATION

To further enhance the discriminative power between normal and anomalous patterns within the latent space, we introduce a constraint designed to amplify the discrepancy between the two patterns. We now motivate the proposed constraint by giving a closer look at the discrepancy between normal and anomalous weights $\mathbf{w} = \{w^k\}_{k=1}^{K} \in \mathbb{R}^K$ computed by weight learner $\phi(\mathbf{x}; \theta_\phi)$.

**Proposition 1** *Let $\mathbf{w}_n \in \mathcal{N}$ and $\mathbf{w}_a \in \mathcal{A}$ denote the computed weights of normal and anomalous samples, where $\mathcal{N}$ and $\mathcal{A}$ denote the weight sets of normal and anomalous samples, respectively. Given $\mathbf{w} \in \sum^K$, where $\sum^K$ represents the probability simplex in $\mathbb{R}^K$, the expected discrepancy between normal and anomalous weights, $\mathbb{E}_{\mathbf{w}_n \in \mathcal{N}, \mathbf{w}_a \in \mathcal{A}} \left[ \|\mathbf{w}_n - \mathbf{w}_a\|_2^2 \right]$, can be amplified by increasing the variance of $\|\mathbf{w}_n\|_2$ among $\mathcal{N}$.*

**Proposition 2** *Denoting $\mu_{\mathbf{w}_n} = \mathbb{E}_{\mathbf{w}_n \in \mathcal{N}} [\mathbf{w}_n]$ as the centroid of the normal samples' weights, $\mathbb{E}_{\mathbf{w}_n \in \mathcal{N}, \mathbf{w}_a \in \mathcal{A}} \left[ \|\mathbf{w}_a - \mu_{\mathbf{w}_n}\|_2^2 \right] - \mathbb{E}_{\mathbf{w}_n \in \mathcal{N}} \left[ \|\mathbf{w}_n - \mu_{\mathbf{w}_n}\|_2^2 \right]$, can be amplified by increasing the variance of $\|\mathbf{w}_n\|_2$ among $\mathcal{N}$.*

The proofs of Proposition 1 and Proposition 2 are provided in Appendix A.11. Since only normal samples are available during training, we implement this constraint by enforcing separation among the weights of normal training samples, i.e., we push each normal sample's corresponding weight $\mathbf{w}_n$ away from any other one. By sufficiently separating the normal weights $\mathbf{w}_n$, we can promote increased variance in their $L_2$ norms ($\text{Var}(\|\mathbf{w}_n\|_2)$), thereby promoting greater discrepancy between normal weights $\mathbf{w}_n$ and anomalous weights $\mathbf{w}_a$. Given that the reconstructed representation $\tilde{\mathbf{h}}$ can be expressed as $\tilde{\mathbf{h}} = \sum_{k=1}^{K} w^k \boldsymbol{\beta}_k$, where basis vectors $\mathcal{B} = \{\boldsymbol{\beta}_k\}_{k=1}^{K}$ are fixed, $\mathbf{w}$ could be viewed as the computed coordinates of $\tilde{\mathbf{h}}$ in the subspace consisting of basis vectors. Therefore, the separation between $\mathbf{w}_n$ and $\mathbf{w}_a$ promotes separation among normal and anomalous $\tilde{\mathbf{h}}$. This process is achieved by minimizing the separation loss as follows:

$$
\begin{aligned}
\min_{\theta_\phi} \mathcal{L}_{separation} &= \max_{\theta_\phi} \frac{1}{N(N-1)} \sum_{i,j=1}^{N} \mathbf{1}\{i \neq j\} d(\mathbf{w}_i, \mathbf{w}_j) \\
&= \min_{\theta_\phi} -\frac{1}{N(N-1)} \sum_{i,j=1}^{N} \mathbf{1}\{i \neq j\} d(\phi(\mathbf{x}_i; \theta_\phi), \phi(\mathbf{x}_j; \theta_\phi)),
\end{aligned}
\tag{4}
$$

where $d(\cdot, \cdot)$ is the distance measurement. This constraint amplifies the discrepancy between normal and anomalous patterns in the latent space, which is verified in Fig. 5 and Appendix A.10.

### 3.2.3 OVERALL ALGORITHM AND IMPLEMENTATIONS.

Motivated by reconstruction-based methods (Yin et al., 2024; Thimonier et al., 2024), we also introduce a strategy that incorporates the intrinsic feature correlations of the sample observation $\mathbf{x}$

into representation $\mathbf{h}$ in the latent space. This approach helps prevent the model from extracting task-irrelevant representations while preserving the critical features of the normal data. Specifically, we minimize the following alignment loss:

$$\min_{\theta_f,\theta_g} \mathcal{L}_{alignment} = \min_{\theta_f,\theta_g} \frac{1}{N} \sum_{i=1}^{N} d(\mathbf{x}_i, \tilde{\mathbf{x}}_i) = \min_{\theta_f,\theta_g} \frac{1}{N} \sum_{i=1}^{N} d(\mathbf{x}_i, g(\mathbf{h}_i; \theta_g))$$

$$= \min_{\theta_f,\theta_g} \frac{1}{N} \sum_{i=1}^{N} d(\mathbf{x}_i, g(f(\mathbf{x}_i; \theta_f); \theta_g)), \tag{5}$$

where $g(\cdot; \theta_g) : \mathbb{R}^E \to \mathbb{R}^D$ is the alignment learner, $\tilde{\mathbf{x}}_i = g(\mathbf{h}_i; \theta_g) = g(f(\mathbf{x}_i; \theta_f); \theta_g)$ is the reconstruction of $\mathbf{x}_i$, and $d(\cdot, \cdot)$ is the distance measurement.

Altogether, DRL aims to minimize the following objective function w.r.t. $\theta_f, \theta_\phi, \theta_g$ during training:

$$\min_{\theta_f,\theta_\phi,\theta_g} \mathcal{L}_{all} = \min_{\theta_f,\theta_\phi} \mathcal{L}_{decomposition} + \lambda_1 \min_{\theta_\phi} \mathcal{L}_{separation} + \lambda_2 \min_{\theta_f,\theta_g} \mathcal{L}_{alignment}, \tag{6}$$

where $\lambda_1, \lambda_2$ indicates the hyper-parameters for balancing the total loss functions. The decomposed representation learning is achieved via $\mathcal{L}_{decomposition}$ (Eq. 3) to capture the unique patterns of normal sample representations in the latent space. Additionally, a separation constraint via $\mathcal{L}_{separation}$ (Eq. 4) is introduced to further enhance the discriminative power between normal and anomalous patterns within the latent space. In practice, the separation constraint is implemented within each mini-batch to reduce the computational cost. We also introduce a strategy via $\mathcal{L}_{alignment}$ (Eq. 5) that incorporates the intrinsic feature correlations of the sample observations into representations. During inference, we derive the anomaly score $s = d(\mathbf{h}, \tilde{\mathbf{h}}) = d(f(\mathbf{x}; \theta_f), \phi(\mathbf{x}; \theta_\phi)\mathcal{B})$ from the $\mathcal{L}_{decomposition}$ (Eq. 3) optimized during training, which performs reconstruction for representation $\mathbf{h}$ in the constrained latent space, rather than in the observed data space (typically adopting $d(\mathbf{x}, \tilde{\mathbf{x}})$ as anomaly score discussed in Section 2). A higher anomaly score indicates higher confidence that $\mathbf{x}$ is an anomaly. We provide the training and inference process of DRL in Algorithm 1 and 2 of Appendix A.3 respectively.

Given the potential entanglement between normal and anomalous samples in the observed data space, our proposed method offers a notable advantage by providing greater flexibility in adjusting representations within the latent space compared to the observed space (Ye et al., 2024; dan Guo et al., 2022). This flexibility allows us to enforce constraints on the learned representations, making them more discriminative between normal and anomalous patterns, and thereby capturing the distinct characteristics of normal representations more effectively. The representation decomposition process seen by the model only corresponds to the normal samples during training, and a separation constraint is applied to further enhance the distinction between normal and anomalous patterns. This approach significantly impedes the ability to exploit representation decomposition for anomalies, ensuring a more robust discrimination between the two categories.

## 4 RELATED WORK

**Classical Anomaly Detection.** One of the main settings for anomaly detection is one-class classification, that is identifying anomalies after observing a set of normal samples (Sohn et al., 2021; Yin et al., 2024). Over the decades, various classical methods have been developed, consistently showing strong performance, particularly with tabular data. *Density estimation* This approach typically involves estimating the distribution of normal data using either parametric or non-parametric techniques, allowing for anomaly detection based on sample likelihoods under the estimated distribution. Traditional models include KDE (Parzen, 1962) and GMM (Roberts & Tarassenko, 1994). Recently, ECOD (Li et al., 2022) has employed empirical cumulative distribution for this purpose. Additionally, some methods utilize local density estimation to identify anomalies, such as LOF (Breunig et al., 2000). *Classification-based methods* This category focuses on learning a decision boundary using only normal data. For example, OCSVM (Schölkopf et al., 1999) creates this boundary by maximizing the margin between the input data and the origin. *Reconstruction-based methods* These methods are among the most widely used as discussed in Section 2, including classical methods such as PCA (Shyu et al., 2003). For a comprehensive overview of anomaly detection, we recommend the following surveys (Pang et al., 2021; Ruff et al., 2021; Chandola et al., 2009).

**Deep Learning-based Anomaly Detection.** As classical methods struggle to capture complex patterns and relationships in high-dimensional spaces (Pang et al., 2021), recent studies have prompted a shift toward deep learning methodologies for tabular data. For example, ICL (Shenkar & Wolf, 2022) learns mapping relationships that maximize the mutual information between the feature subset and its rest. RAPP (Kim et al., 2019) computes anomaly scores by comparing observations and reconstructions in both the input and hidden space of a well-trained AutoEncoder (Chen et al., 2018). In contrast, our approach re-maps observations into a specially designed constrained space and decomposes the representations into basis vectors to capture normal patterns. Inspired by diffusion models (Ho et al., 2020), DTE (Livernoche et al., 2024) simplifies the diffusion process to reconstruct observations for tabular anomaly detection. Recent advancements, including MCM (Yin et al., 2024) and NPT-AD (Thimonier et al., 2024), are discussed in Section 2. Among these, the widely used methods primarily belong to reconstruction-based approaches (Kim et al., 2019; Livernoche et al., 2024; Yin et al., 2024; Thimonier et al., 2024). However, the issue of data entanglement between normal and anomalous samples is ignored.

# 5 EXPERIMENT & ANALYSIS

## 5.1 EXPERIMENT SETUP

**Datasets.** We conduct experiments on an extensive benchmark of 40 tabular anomaly detection datasets selected from Outlier Detection DataSets (ODDS) (Rayana, 2016) and Anomaly Detection Benchmark (ADBench) (Han et al., 2022), following previous works (Yin et al., 2024; Thimonier et al., 2024). These datasets span diverse domains, including healthcare, science, and social sciences, and vary in characteristics from low-dimensional, low-scale to high-dimensional, large-scale. Detailed information about the dataset properties is provided in Appendix A.1.

**Evaluation Metrics.** Per the literature (Zong et al., 2018; Bergman & Hoshen, 2020; Yin et al., 2024; Thimonier et al., 2024), we construct the training set by randomly subsampling 50% of the normal samples. The remaining 50% of the normal samples are then combined with the entire set of anomalies to form the test set. Following previous work (Han et al., 2022; Yin et al., 2024), we employ Area Under the Precision-Recall Curve (AUC-PR) and Area Under the Receiver Operating Characteristic Curve (AUC-ROC) as our evaluation criteria.

**Baseline Models.** We conduct a comparative analysis between DRL and 16 other prominent methods in the field of TAD. Specifically, we include classic techniques such as OCSVM (Schölkopf et al., 1999), KNN (Ramaswamy et al., 2000), LOF (Breunig et al., 2000), PCA (Shyu et al., 2003), IForest (Liu et al., 2008), and ECOD (Li et al., 2022), which continue to be widely used. In addition, we compare our method against recent deep learning-based approaches, including DAGMM (Zong et al., 2018), Deep SVDD (Ruff et al., 2018), AutoEncoder (Chen et al., 2018), RAPP (Kim et al., 2019), GOAD (Bergman & Hoshen, 2020), NeuTraLAD (Qiu et al., 2021), ICL (Shenkar & Wolf, 2022), DTE (Livernoche et al., 2024), MCM (Yin et al., 2024), and NPT-AD (Thimonier et al., 2024). More details about the implementation of baseline models can be found in Appendix A.2.

**DRL Details.** The DRL architecture remains consistent across all datasets. Specifically, the feature extractor $f(\cdot; \theta_f) : \mathbb{R}^D \rightarrow \mathbb{R}^E$ and weight learner $\phi(\cdot; \theta_\phi) : \mathbb{R}^D \rightarrow \mathbb{R}^K$ are implemented as a simple two-layer fully connected MLP with Leaky ReLU activation function. The last layer of weight learner is with Softmax activation function. The alignment learner $g(\cdot; \theta_g) : \mathbb{R}^E \rightarrow \mathbb{R}^D$ is a linear layer. For the distance measurement $d(\cdot, \cdot)$, we use L2 distance for $\mathcal{L}_{decomposition}$ (Eq. 3), Cosine distance for $\mathcal{L}_{separation}$ (Eq. 4) and $\mathcal{L}_{alignment}$ (Eq. 5). The default number of basis vectors $K$ is set to 5, and these basis vectors are not updated during training. $\lambda_1, \lambda_2$ is set to 0.06 and 0.1 for separation loss and alignment loss respectively. The hidden dimension $E$ is set to 128, the batch size is set to 512, and the number of epochs is set to 200. Adam optimizer is employed, bounded by an exponentially decaying learning rate controller with 0.05 as initialization. To reduce the effect of randomness, the reported performance is averaged over 10 independent runs.[1]

---

[1] The source code is available at `https://github.com/HangtingYe/DRL`.

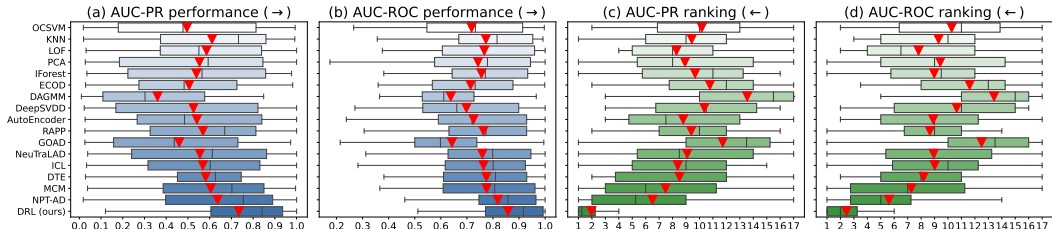

Figure 3: Comparison of all models' performance and ranking across different datasets in terms of AUC-PR and AUC-ROC. The red triangles represent the average value over all datasets.

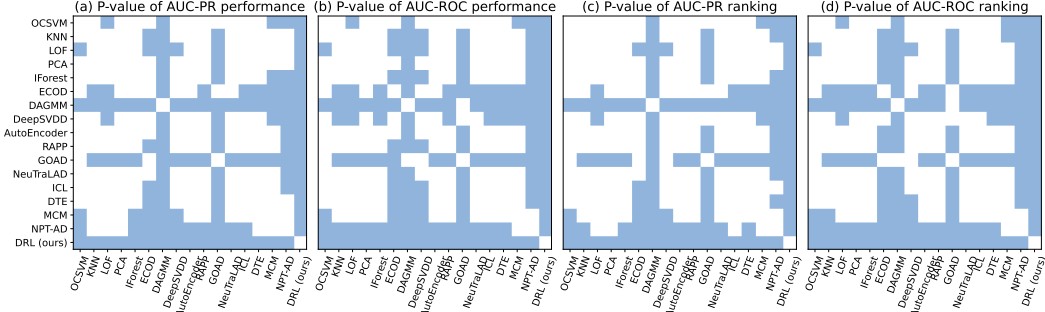

Figure 4: The wilcoxon test is conducted between different models across all datasets. The significant difference of different settings are provided. Blue color indicates the corresponding p-value is lower than 0.05 (significant), while white color indicates the corresponding p-value is higher than 0.05 (not significant).

## 5.2 EMPIRICAL RESULTS AND ANALYSIS

**Main Results.** We evaluated the performance and model ranking of DRL against 16 mainstream baseline models across 40 tabular datasets using two evaluation metrics. The results, comprising 2(performance & ranking)×(16+1)(models)×40(datasets)×2(metrics) = 2720 numerical results, are summarized in boxplots, with each boxplot representing a model's performance across various datasets, as illustrated in Fig. 3. Detailed results, average standard deviations, and analyses of the baseline methods are available in Appendix A.4. Overall, DRL demonstrates state-of-the-art performance, achieving an absolute average AUC-PR gain of at least 9.8% compared to other methods. Notably, DRL consistently enhances performance across datasets, regardless of dimensionality or size, as shown in Fig. 8 of Appendix A.4. We conduct the Wilcoxon signed-rank test (with $\alpha = 0.05$) (Woolson, 2007) to measure the improvement significance, as illustrated in Fig. 4. In all settings, the improvement of DRL over baseline models is statistically significant at the 95% confidence level. Additionally, we provide T-SNE visualizations of real-world data in Fig. 7 of Appendix A.1, revealing that data entanglement is a common issue in many tabular datasets. Even in cases without data entanglement in the observation space, DRL enhances performance by making normal and anomalous patterns more distinguishable.

**Different Basis Vector Initialization Methods.** As we utilize a set of orthogonal vectors as basis vectors to capture normal patterns, we systematically investigate various orthogonalization methods for generating these vectors, as detailed in Table 1. Specifically, Random indicates that we randomly sample linearly independent vectors from the standard Gaussian distribution without applying any orthogonalization. The Householder (Schreiber & Van Loan, 1989) transformation reflects a given vector across a hyperplane defined by a unit vector, successively generating orthogonal vectors. Givens (Shalit & Chechik, 2014) rotation applies a rotation matrix to a pair of coordinates, effectively zeroing out one component while preserving orthogonality. SVD denotes Singular Value

Table 1: Comparison of basis vector generation strategies averaged over all datasets. Orthogonalization of basis vectors is crucial for capturing normal patterns and distinguishing them from anomalies, with different methods yielding similar robustness. The Gram-Schmidt process achieves the best performance, while learnable basis vectors underperform due to optimization complexity.

| Metrics | Random | Householder | Givens | SVD | GS (0.5, 1) | GS (0, 1.5) | GS* (0, 1) | GS (0, 1) (ours) |
|---|---|---|---|---|---|---|---|---|
| AUC-PR | 0.667 (0.031) | 0.729 (0.021) | **0.736** (0.024) | 0.712 (0.023) | 0.730 (0.028) | 0.732 (0.030) | 0.697 (0.032) | 0.734 (0.028) |
| AUC-ROC | 0.831 (0.021) | 0.851 (0.011) | 0.849 (0.013) | 0.837 (0.014) | 0.851 (0.014) | 0.856 (0.015) | 0.836 (0.024) | **0.857** (0.013) |

Table 2: Analysis on the effects of different components of DRL in terms of AUC-PR and AUC-ROC averaged over 40 datasets. The full results are provided in Table 11 and Table 12 in Appendix A.10.

| Variants | Training | | | | Inference | | Evaluation Metric | |
|---|---|---|---|---|---|---|---|---|
| | $\mathcal{L}_{decomposition}$ | $\mathcal{L}_{separation}$ | $\mathcal{L}_{alignment}$ | Weight learner | $\mathcal{L}_{decomposition}$ | $\mathcal{L}_{alignment}$ | AUC-PR | AUC-ROC |
| A | ✓ | × | × | ✓ | ✓ | × | 0.6344 | 0.7882 |
| B | ✓ | × | ✓ | ✓ | ✓ | × | 0.6844 | 0.7999 |
| C | ✓ | ✓ | × | ✓ | ✓ | × | 0.7081 | 0.8123 |
| D | × | × | ✓ | × | × | ✓ | 0.5222 | 0.7066 |
| E | ✓ (Observation) | × | × | ✓ | ✓ (Observation) | × | 0.5407 | 0.7240 |
| F | ✓ | ✓ | ✓ | × (Least square) | ✓ | × | 0.6964 | 0.8205 |
| G | ✓ (MCM's representation) | × | × | ✓ | ✓ (MCM's representation) | × | 0.6013 | 0.7273 |
| H (MCM w/ DRL) | ✓ | ✓ | ✓ | ✓ | ✓ | × | 0.6750 | 0.7837 |
| I | ✓ | ✓ | ✓ | ✓ | ✓ | ✓ | 0.6928 | 0.8286 |
| J | ✓ | ✓ | ✓ | ✓ | × | ✓ | 0.6100 | 0.7720 |
| **DRL (ours)** | ✓ | ✓ | ✓ | ✓ | ✓ | × | **0.7344** | **0.8574** |

Decomposition (Klema & Laub, 1980), which decomposes a matrix into orthogonal matrices and a diagonal matrix capturing the singular values, thus revealing the matrix's latent structure. GS refers to the Gram-Schmidt process (Leon et al., 2013) utilized in our study. In this experiment, we sample linearly independent vectors from a Gaussian distribution, with the values in parentheses representing the mean and variance. GS* (0, 1) indicates that the basis vectors are learnable during training, while they remain fixed in other settings.

**Effectiveness of Key Components of DRL.** We further analyze the effectiveness of key components of DRL, as illustrated in Table 2, by comparing DRL against three groups of variants. *Variant A to E explore training DRL with different loss configurations*: (1) variant A, that removes both separation constraint and alignment loss when training DRL; (2) variant B, that removes separation constraint when training DRL; (3) variant C, that removes alignment loss when training DRL; (4) variant D, that uses only the alignment loss when training DRL, with the alignment loss directly serving as the anomaly score during inference; (5) variant E, that first constructs a set of orthogonal basis vectors, matching the dimensionality of the input features. It then employs the weight learner, as detailed in Section 3.2.1, to compute sample-specific weights. These weights are used to form a linear combination of the basis vectors, yielding the reconstructed observations. The decomposition loss is applied directly to the observations, and this loss serves as the anomaly score. The removal of any of the components during training degrades the performance of DRL. Notably, the separation constraint significantly enhances performance, which we attribute to its ability to improve the discriminative power between normal and anomalous patterns within the latent space. *Variant F to H examine training DRL with different architectures*: (6) variant F, decomposes each normal sample representation into a linear combination of orthogonal basis vectors with sample-specific weights, and computes these weights using least squares instead of a weight learner; (7) variant G, that applies the decomposition loss to the well-trained representations of MCM, assuming that the fixed normal representations can be decomposed into orthogonal basis vectors; (8) variant H, that replaces DRL's feature extractor and alignment learner with MCM's encoder and decoder, effectively training MCM within the DRL framework. While Variant F achieves promising performance, it still underperforms compared to DRL. DRL benefits from a learnable weight learner that captures the global distribution of weights across all normal training samples, enabling it to accurately represent complex normal patterns. Variant G highlights that well-trained reconstruction-based representations may not lie within the latent space defined by the basis vectors, complicating the modeling of normal information. Moreover, variant H shows that while substituting DRL's components with MCM's leads to significant improvements over MCM itself, it still falls short of DRL due to the observation-centric strategy, which can introduce data entanglement. *Variant I and J test the well-trained DRL with different anomaly score configurations during inference*: (9) variant I, that uses both decomposition and alignment losses as anomaly scores during inference; (10) variant J, that only uses alignment loss as anomaly scores during inference. The results indicate that decomposition loss is more effective than alignment loss in capturing normal information during inference.

**Different Distance Metrics.** Besides, we also provide additional experiments to validate the effectiveness of different distance metrics for decomposition loss, separation constraint and alignment loss in Table 3. By default, we use L2 distance for decomposition loss, cosine distance for separation loss and alignment loss. For separation and alignment loss, L1, L2, and cosine distance metrics all yield promising performance, with cosine distance demonstrating relatively superior and more stable performance. The separation constraint exhibited more variability depending on the distance metric used. We attribute this to the fact that when maximizing the L1 and L2 distance metrics, the

Table 3: Analysis on the effects of various distance metrics in terms of AUC-PR across different datasets. The average results over 40 datasets are also provided. Full results are in Table 13 and Table 14 in Appendix A.10.

| Different components | Backdoor | Fault | Imgseg | Lympho | Pendigits | Vowels | Wbc | Average of 40 data |
|---|---|---|---|---|---|---|---|---|
| Decomposition w/ Cosine distance | 0.8808 | 0.6534 | 0.9036 | 0.9868 | 0.9029 | 0.4307 | 0.9417 | 0.7130 |
| Separation w/ L1 distance | 0.8784 | 0.6391 | 0.9185 | 0.8391 | 0.882 | 0.44 | 0.9589 | 0.7007 |
| Alignment w/ L1 distance | 0.8868 | 0.6433 | 0.9004 | 0.9762 | 0.9218 | 0.3696 | 0.9401 | 0.7063 |
| Separation w/ L2 distance | 0.8786 | 0.6444 | 0.8998 | 0.8900 | 0.8735 | **0.4635** | 0.9655 | 0.7080 |
| Alignment w/ L2 distance | 0.8886 | 0.6576 | 0.9125 | **1.0000** | 0.9090 | 0.4425 | 0.9590 | 0.7134 |
| **DRL (ours)** | **0.8915** | **0.6649** | **0.9238** | **1.0000** | **0.9360** | 0.4506 | **0.9742** | **0.7344** |

lack of inherent range limitations can lead to excessively large distance values, which hinder the model's optimization process.

**Discussion.** In Fig. 5, we present the average discrepancy between normal and anomalous observations over all data pairs which is labeled as "Original", alongside the discrepancy of latent representations under "DRL w/o S" and "DRL", where "S" denotes the separation constraint. The results in Fig. 5 indicate that our proposed method enhances the discriminative distinction between normal and anomalous patterns within the latent space. Additionally, we visualize the T-SNE of the learned representations from DRL on the Annthyroid dataset in Fig. 6. The findings reveal that the observation entanglement between normal and anomalous samples in existing reconstruction-based methods can obscure their distinction within the latent space. In contrast, DRL effectively separates the representations into distinct regions, thereby enhancing the discriminative capability between normal and anomalous patterns.

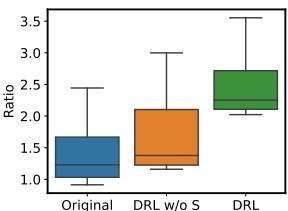

Figure 5: The ratio of the mean L2 distance between normal and anomalous samples to the mean L2 distance among pairs of normal samples across all datasets.

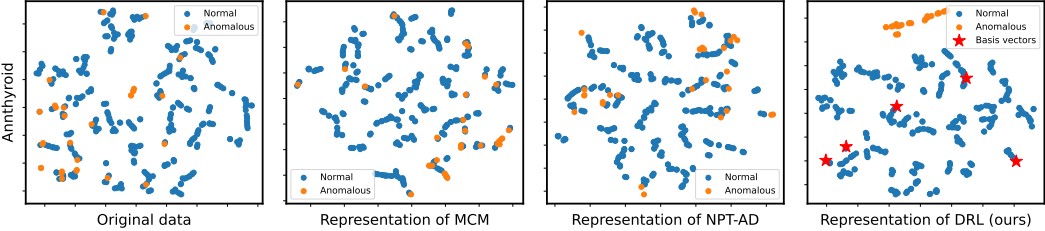

Figure 6: Visualization for original data space and deep TAD models' latent space on real-world datasets.

**Additional Experiments.** The computational efficiency is detailed in Appendix A.5. We incorporate the sensitivity analysis w.r.t. the number of basis vectors, the loss weight $\lambda_1$, $\lambda_2$, the number of training epochs and the number of batch size in Appendix A.6. We also provide the weight visualization of normal and anomalous representations in Appendix A.7, DRL's robustness to anomaly contamination in Appendix A.8, and the model performance on different types of anomalies in Appendix A.9. Furthermore, additional empirical evidence and theoretical analysis are provided in Appendix A.10 and Appendix A.11, respectively.

## 6 CONCLUSION

We study the problem of anomaly detection for tabular data, typically implemented in a one-class classification setting where the training data comprises solely normal samples. Most state-of-the-art TAD works are along the line of reconstruction; however, reconstruction on training data can still hardly distinguish anomalies due to the data entanglement. To address this problem, in this paper, we propose a novel approach Decomposed Representation Learning (DRL), to re-map data into a tailor-designed constrained space, in order to capture the underlying shared patterns of normal samples and differ anomalous patterns for tabular anomaly detection. Our approach decomposes each normal sample's representation into a weighted linear combination of randomly generated orthogonal basis vectors, where these basis vectors are both data-free and training-free. We further enhance the discriminative ability between normal and anomalous patterns through a simple and theoretically sound constraint. The empirical results demonstrated the effectiveness of DRL for tabular anomaly detection. Our work can shed some light on the development of better algorithms for similar tasks.

ACKNOWLEDGMENTS

The authors would like to thank the anonymous referees for their valuable comments. In this work, Hangting Ye, Dandan Guo and Yi Chang are supported by the National Natural Science Foundation of China (No. 623B2043, No. U2341229, No. 62306125) and the National Key R&D Program of China under Grant (No. 2023YFF0905400).

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

# A APPENDIX

In this section, we provide the details of datasets, the implementations of baseline methods, the full comparison results with baseline methods, the computational efficiency details, hyper-parameter sensitivity analysis, robustness to anomaly contamination, model performance on different types of anomalies and visualization. The source code for implementing DRL will be released after the paper is published.

## A.1 DATASETS DETAILS

We experiment on an extensive benchmark of tabular datasets following previous work (Yin et al., 2024), which spans diverse domains, including healthcare, natural science, social sciences, etc. We omit the KDD dataset since it presents a certain number of limitations (Silva et al., 2020). Instead, according to (Thimonier et al., 2024), we include three real-world datasets from (Han et al., 2022) that display relatively similar characteristics to KDD in terms of dimensions: Fraud, Campaign, and Backdoor. In addition, we extend the benchmark of (Yin et al., 2024) by including other commonly used tabular real-world datasets from ADBench (Han et al., 2022). The dataset properties are summarized in Table 4. The Table 4 shows the number of samples, the number of features, and the number of anomalies of each dataset used. Additionally, we provide T-SNE visualizations of real-world data in Fig. 7.

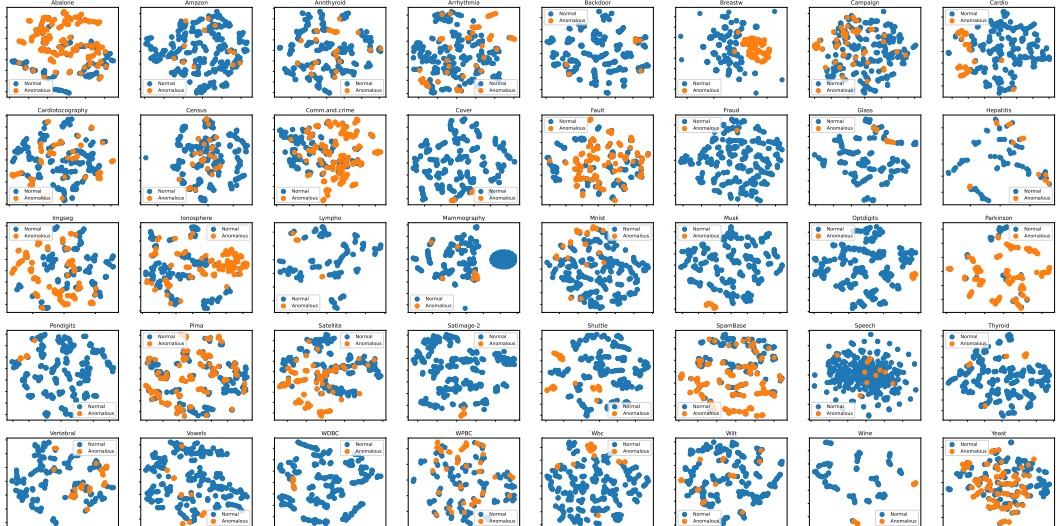

Figure 7: T-SNE visualization of real world datasets.

Table 4: Dataset properties. We use 40 commonly used tabular anomaly detection datasets in this paper.

|  | Samples | Dims | Anomalies |
|---|---|---|---|
| Abalone | 4177 | 7 | 2081 |
| Amazon | 10000 | 768 | 500 |
| Annthyroid | 7200 | 6 | 534 |
| Arrhythmia | 452 | 274 | 66 |
| Backdoor | 95329 | 196 | 2329 |
| Breastw | 683 | 9 | 239 |
| Campaign | 41188 | 62 | 4640 |
| Cardio | 1831 | 21 | 176 |
| Cardiotocography | 2114 | 21 | 466 |
| Census | 299285 | 500 | 18568 |
| Comm.and.crime | 1994 | 101 | 993 |
| Cover | 286048 | 10 | 2747 |
| Fault | 1941 | 27 | 673 |
| Fraud | 284807 | 29 | 492 |
| Glass | 214 | 9 | 9 |
| Hepatitis | 80 | 19 | 13 |
| Imgseg | 2310 | 18 | 990 |
| Ionosphere | 351 | 33 | 126 |
| Lympho | 148 | 18 | 6 |
| Mammography | 11183 | 6 | 260 |
| Mnist | 7603 | 100 | 700 |
| Musk | 3062 | 166 | 97 |
| Optdigits | 5216 | 64 | 150 |
| Parkinson | 195 | 22 | 147 |
| Pendigits | 6870 | 16 | 156 |
| Pima | 768 | 8 | 268 |
| Satellite | 6435 | 36 | 2036 |
| Satimage-2 | 5803 | 36 | 71 |
| Shuttle | 49097 | 9 | 3511 |
| SpamBase | 4207 | 57 | 1679 |
| Speech | 3686 | 400 | 61 |
| Thyroid | 3772 | 6 | 93 |
| Vertebral | 240 | 6 | 30 |
| Vowels | 1456 | 12 | 50 |
| WDBC | 367 | 30 | 10 |
| WPBC | 198 | 33 | 47 |
| Wbc | 378 | 30 | 21 |
| Wilt | 4819 | 5 | 257 |
| Wine | 129 | 13 | 10 |
| Yeast | 1484 | 8 | 507 |

## A.2 BASELINE MODELS DETAILS

We conduct a comparative analysis between DRL and 16 other prominent methods in the field of TAD, including classical non-deep methods and recent deep learning based methods. Specifically, OCSVM (Schölkopf et al., 1999), KNN (Ramaswamy et al., 2000), LOF (Breunig et al., 2000), PCA (Shyu et al., 2003), IForest (Liu et al., 2008) and ECOD (Li et al., 2022) represent classic AD approaches that continue to maintain popularity. In addition, we compare our method to recent deep learning based methods, namely DAGMM (Zong et al., 2018), Deep SVDD (Ruff et al., 2018), AutoEncoder (Chen et al., 2018), RAPP (Kim et al., 2019), GOAD (Bergman & Hoshen, 2020), NeuTraLAD (Qiu et al., 2021), ICL (Shenkar & Wolf, 2022), DTE (Livernoche et al., 2024), MCM (Yin et al., 2024) and NPT-AD (Thimonier et al., 2024). We use the popular PyOD python package (Zhao et al., 2019) to implement OCSVM, LOF, PCA, IForest, ECOD, Deep SVDD and AutoEncoder. We use the DeepOD python library (Xu et al., 2023) to implement GOAD, NeuTraLAD and ICL. The implementation of the other methods is based on their official open-source code releases. Following latest works (Yin et al., 2024; Thimonier et al., 2024), We implement all baseline models' hyperparameters following their original papers. All the methods are implemented with identical dataset partitioning and preprocessing procedures, following previous works (Yin et al., 2024).

## A.3 DRL WORKFLOW

---

**Algorithm 1** DRL training workflow.

---

**Input:** Training dataset $\mathcal{D}_{train}$, feature extractor $f(\cdot; \theta_f) : \mathbb{R}^D \to \mathbb{R}^E$, weight learner $\phi(\cdot; \theta_\phi)$, alignment learner $g(\cdot; \theta_g) : \mathbb{R}^E \to \mathbb{R}^D$, orthogonal basis vectors $\mathcal{B} = \{\boldsymbol{\beta}_k\}_{k=1}^K \subseteq \mathbb{R}^{K \times E}$ obtained by Eq. 2, hyper-parameters $\lambda_1, \lambda_2$;

1: **while** $\theta_f, \theta_\phi, \theta_g$ has not converged **do**
2:     Sample a mini-batch $B$ from $\mathcal{D}_{train}$;
3:     **for** $\mathbf{x}_i$ in $B$ **do**
4:         Obtain sample representation $\mathbf{h}_i$ by $f(\mathbf{x}_i; \theta_f)$;
5:         Calculate the corresponding weights $\mathbf{w}_i$ in latent space by $\phi(\mathbf{x}_i; \theta_\phi)$;
6:         Obtain the estimated observation $\tilde{\mathbf{x}}_i$ by $g(\mathbf{h}_i; \theta_g)$;
7:     **end for**
8:     Compute $\mathcal{L}_{decomposition} = \frac{1}{|B|} \sum_{\mathbf{x}_i \in B} d(f(\mathbf{x}_i; \theta_f), \phi(\mathbf{x}_i; \theta_\phi)\mathcal{B})$ with Eq. 3;
9:     Compute $\mathcal{L}_{separation} = -\frac{1}{|B|(|B|-1)} \sum_{i,j=1}^{|B|} \mathbf{1}\{i \neq j\}\, d(\phi(\mathbf{x}_i; \theta_\phi), \phi(\mathbf{x}_j; \theta_\phi))$ with Eq. 4;
10:    Compute $\mathcal{L}_{alignment} = \frac{1}{|B|} \sum_{\mathbf{x}_i \in B} d(\mathbf{x}_i, g(f(\mathbf{x}_i; \theta_f); \theta_g))$ with Eq. 5;
11:    Update $\theta_f, \theta_\phi, \theta_g$ by minimizing $\mathcal{L}_{all} = \mathcal{L}_{decomposition} + \lambda_1 \mathcal{L}_{separation} + \lambda_2 \mathcal{L}_{alignment}$ with Eq. 6 through gradient descent.
12: **end while**
**Output:** Trained feature extractor $f(\cdot; \theta_f)$, trained weight learner $\phi(\cdot; \theta_\phi)$.

---

---

**Algorithm 2** DRL inference workflow.

---

**Input:** Test dataset $\mathcal{D}_{test}$, trained feature extractor $f(\cdot; \theta_f)$, trained weight learner $\phi(\cdot; \theta_\phi)$, orthogonal basis vectors $\mathcal{B} = \{\boldsymbol{\beta}_k\}_{k=1}^K \subseteq \mathbb{R}^{K \times E}$ obtained by Eq. 2;

1: **for** $\mathbf{x}_i$ in $\mathcal{D}_{test}$ **do**
2:     Obtain sample representation $\mathbf{h}_i$ by $f(\mathbf{x}_i; \theta_f)$;
3:     Calculate the corresponding weights $\mathbf{w}_i$ in latent space by $\phi(\mathbf{x}_i; \theta_\phi)$;
4:     Compute $s_i = d(f(\mathbf{x}_i; \theta_f), \phi(\mathbf{x}_i; \theta_\phi)\mathcal{B})$ as anomaly score (higher score indicates higher confidence that $\mathbf{x}_i$ is an anomaly) derived from the $\mathcal{L}_{decomposition}$ (Eq. 3).
5: **end for**
**Output:** Anomaly score $\{s_i\}_{i=1}^{|\mathcal{D}_{test}|}$.

---

## A.4 Full Comparison Results with Baseline Methods

We compare the performance and model ranking of DRL against 16 mainstream baseline models across 40 tabular datasets using 2 evaluation metrics. We summarize this large-scale experiment results (2(performance & ranking)×(16+1)(models)×40(datasets)×2(metrics) = 2720 numerical results) in boxplots, where each boxplot corresponding to each model shows its results across different datasets, as shown in Fig. 3. The full results and average standard deviation are provided in table 5 and table 6. We can observe that DRL achieves the highest average performance, and the lowest average ranking overall datasets for both AUC-PR and AUC-ROC. Specifically, DRL obtains an absolute average AUC-PR gain of at least 9.8% compared to other methods. Considering the overall performance and model ranking across 40 datasets, DRL achieves state-of-the-art performance. Furthermore, DRL consistently improves performance compared to baseline models w.r.t. both dimensionality and data size, as detailed in Fig. 8. Additionally, we conduct Wilcoxon signed-rank test (with $\alpha = 0.05$) (Woolson, 2007) to measure the improvement significance, as illustrated in Fig. 4. In all settings, the improvement of DRL over baseline models is statistically significant at the 95% confidence level. This demonstrates the superior effectiveness of DRL compared to baseline models across different datasets. As the representative reconstruction-based methods for tabular anomaly detection, DTE, MCM and NPT-AD belong to the best-performing baseline methods according to the results, which is aligned with the results in their original papers. However, as discussed in Section 3.1, relying on reconstruction loss from observed samples to differentiate anomalous samples may be insufficient due to inherent data entanglement. RAPP computes anomaly scores by comparing sample observations with their autoencoder reconstructions in both the input and hidden spaces, yet it does not address the underlying issue of data entanglement between normal and anomalous samples. In contrast, DRL enhances anomaly detection in tabular data by learning the shared underlying normal patterns of normal samples through representation decomposition in a constrained latent space, where normal and anomalous patterns are more discriminative. We provide TSNE visualizations of real-world data in Fig. 7, and our empirical observations indicate that data entanglement is a prevalent issue in many real-world tabular datasets. As shown in table 5 and table 6, even in cases without data entanglement in the observation space, DRL still improves performance by making normal and anomalous patterns more distinguishable.

Table 5: Comparison of AUC-PR (↑) results between baseline methods and DRL on 40 datasets.

| | OCSVM | KNN | LOF | PCA | IForest | ECOD | DAGMM | DeepSVDD | AutoEncoder | RAPP | GOAD | NeuTraLAD | ICL | DTE | MCM | NPT-AD | DRL (ours) |
|---|---|---|---|---|---|---|---|---|---|---|---|---|---|---|---|---|---|
| Abalone | 0.8459 | 0.8596 | 0.813 | 0.8393 | 0.8481 | 0.6554 | 0.7869 | 0.7173 | 0.8295 | 0.8125 | 0.8233 | 0.7768 | 0.7654 | 0.4798 | 0.7471 | 0.4899 | 0.885 |
| Amazon | 0.105 | 0.0904 | 0.1117 | 0.1072 | 0.1091 | 0.104 | 0.1022 | 0.1003 | 0.117 | 0.1179 | 0.1086 | 0.1007 | 0.1077 | 0.1115 | 0.1083 | 0.0861 | 0.1206 |
| Annthyroid | 0.1831 | 0.3525 | 0.4513 | 0.5657 | 0.6149 | 0.4002 | 0.2189 | 0.3235 | 0.5291 | 0.3348 | 0.2132 | 0.5057 | 0.4114 | 0.6288 | 0.3215 | 0.62 | 0.6761 |
| Arrhythmia | 0.5339 | 0.6008 | 0.5277 | 0.5336 | 0.5097 | 0.4461 | 0.4668 | 0.6036 | 0.3029 | 0.4487 | 0.4091 | 0.6237 | 0.6155 | 0.4912 | 0.6107 | 0.4345 | 0.627 |
| Backdoor | 0.1674 | 0.7554 | 0.5138 | 0.1107 | 0.0926 | 0.1552 | 0.2424 | 0.7291 | 0.8435 | 0.7055 | 0.0911 | 0.8716 | 0.8226 | 0.6244 | 0.7049 | 0.8828 | 0.8915 |
| Breastw | 0.9934 | 0.9712 | 0.9923 | 0.9934 | 0.9449 | 0.9522 | 0.7584 | 0.9924 | 0.9896 | 0.9762 | 0.8335 | 0.9117 | 0.9656 | 0.8825 | 0.9952 | 0.9813 | 0.9966 |
| Campaign | 0.4749 | 0.4383 | 0.4459 | 0.4884 | 0.4608 | 0.4705 | 0.2472 | 0.2529 | 0.4857 | 0.4242 | 0.203 | 0.4033 | 0.4506 | 0.469 | 0.604 | 0.4769 | 0.5013 |
| Cardio | 0.8614 | 0.758 | 0.836 | 0.8628 | 0.7018 | 0.3636 | 0.3089 | 0.788 | 0.4778 | 0.7201 | 0.6225 | 0.4535 | 0.8037 | 0.6929 | 0.8489 | 0.7733 | 0.8325 |
| Cardiotocography | 0.6619 | 0.6162 | 0.5732 | 0.6969 | 0.6036 | 0.6968 | 0.444 | 0.4602 | 0.4705 | 0.6729 | 0.4647 | 0.6431 | 0.5955 | 0.5334 | 0.6993 | 0.6401 | 0.754 |
| Census | 0.2279 | 0.146 | 0.2343 | 0.2004 | 0.1357 | 0.1773 | 0.1066 | 0.1514 | 0.1913 | 0.1703 | 0.1445 | 0.1103 | 0.1949 | 0.1794 | 0.242 | 0.2672 | 0.2649 |
| Comm.and.crime | 0.8371 | 0.851 | 0.8519 | 0.8892 | 0.894 | 0.6854 | 0.7475 | 0.8239 | 0.8678 | 0.8751 | 0.933 | 0.9202 | 0.8962 | 0.7901 | 0.8549 | 0.8668 | 0.9164 |
| Cover | 0.019 | 0.0961 | 0.1598 | 0.0716 | 0.0362 | 0.1435 | 0.1355 | 0.0229 | 0.1422 | 0.1302 | 0.2071 | 0.0686 | 0.0714 | 0.6373 | 0.0438 | 0.0184 | 0.783 |
| Fault | 0.6062 | 0.6028 | 0.5101 | 0.6035 | 0.5948 | 0.5171 | 0.5894 | 0.563 | 0.6501 | 0.6227 | 0.5872 | 0.6083 | 0.5937 | 0.6393 | 0.6022 | 0.7578 | 0.6649 |
| Fraud | 0.349 | 0.4052 | 0.4046 | 0.2366 | 0.6939 | 0.4062 | 0.0099 | 0.332 | 0.3611 | 0.55 | 0.1249 | 0.2312 | 0.596 | 0.6214 | 0.5141 | 0.3868 | 0.6614 |
| Glass | 0.0896 | 0.1099 | 0.0923 | 0.0896 | 0.0952 | 0.1113 | 0.1019 | 0.0912 | 0.1079 | 0.0963 | 0.0948 | 0.1491 | 0.2573 | 0.2151 | 0.1905 | 0.2204 | 0.167 |
| Hepatitis | 0.2815 | 0.2744 | 0.3201 | 0.5828 | 0.4182 | 0.4049 | 0.4063 | 0.4264 | 0.4248 | 0.3306 | 0.3393 | 0.369 | 0.3357 | 0.6582 | 0.3372 | 0.3813 | 0.6627 |
| Imgseg | 0.7883 | 0.8531 | 0.8702 | 0.7724 | 0.7556 | 0.7365 | 0.6409 | 0.6699 | 0.8391 | 0.7871 | 0.7114 | 0.8747 | 0.8916 | 0.6846 | 0.8124 | 0.8644 | 0.9238 |
| Ionosphere | 0.8969 | 0.9297 | 0.9591 | 0.8969 | 0.9768 | 0.9713 | 0.7046 | 0.867 | 0.7328 | 0.8915 | 0.928 | 0.9355 | 0.9777 | 0.9683 | 0.9802 | 0.9812 | 0.9895 |
| Lympho | 0.8107 | 0.9401 | 0.9762 | 1 | 0.9593 | 0.8972 | 0.573 | 0.9749 | 0.2709 | 0.6972 | 0.7656 | 0.646 | 0.6091 | 0.8677 | 0.4204 | 0.9929 | 1 |
| Mammography | 0.4178 | 0.381 | 0.4063 | 0.4165 | 0.3334 | 0.538 | 0.1141 | 0.419 | 0.253 | 0.2944 | 0.2426 | 0.1326 | 0.1894 | 0.3985 | 0.4755 | 0.364 | 0.8406 |
| Mnist | 0.1686 | 0.761 | 0.838 | 0.6499 | 0.5349 | 0.3018 | 0.2742 | 0.4158 | 0.279 | 0.7233 | 0.7182 | 0.9021 | 0.8915 | 0.5226 | 0.7782 | 0.7485 | 0.887 |
| Musk | 0.0614 | 0.9917 | 1 | 1 | 0.5279 | 0.982 | 0.0482 | 1 | 1 | 1 | 0.5372 | 1 | 1 | 1 | 0.639 | 1 | 1 |
| Optdigits | 0.0692 | 0.8589 | 0.4363 | 0.0602 | 0.157 | 0.0669 | 0.0536 | 0.1159 | 0.1418 | 0.3436 | 0.0633 | 0.1709 | 0.1696 | 0.1534 | 0.8885 | 0.4203 | 0.9356 |
| Parkinson | 0.8892 | 0.7952 | 0.834 | 0.9297 | 0.9595 | 0.8906 | 0.8484 | 0.9192 | 0.9243 | 0.8378 | 0.8413 | 0.8172 | 0.8568 | 0.7304 | 0.7996 | 0.9637 | 0.921 |
| Pendigits | 0.5178 | 0.9692 | 0.7855 | 0.3863 | 0.5133 | 0.4145 | 0.0441 | 0.0616 | 0.8904 | 0.7321 | 0.0259 | 0.693 | 0.4039 | 0.4844 | 0.8258 | 0.9671 | 0.936 |
| Pima | 0.7008 | 0.7098 | 0.697 | 0.7008 | 0.6662 | 0.5877 | 0.5956 | 0.7165 | 0.7174 | 0.6962 | 0.5027 | 0.6168 | 0.6965 | 0.6798 | 0.7389 | 0.7527 | 0.7449 |
| Satellite | 0.7778 | 0.8515 | 0.8088 | 0.7778 | 0.8583 | 0.8334 | 0.6866 | 0.8217 | 0.8218 | 0.8136 | 0.7786 | 0.8588 | 0.8799 | 0.8479 | 0.8532 | 0.8576 | 0.8692 |
| Satimage-2 | 0.9192 | 0.9555 | 0.9692 | 0.9192 | 0.8846 | 0.7775 | 0.1142 | 0.9427 | 0.9688 | 0.9051 | 0.9726 | 0.9684 | 0.8124 | 0.6821 | 0.985 | 0.9862 | 0.9703 |
| Shuttle | 0.9488 | 0.937 | 0.9601 | 0.9627 | 0.9172 | 0.9815 | 0.4875 | 0.9818 | 0.9316 | 0.9724 | 0.9545 | 0.9971 | 0.9811 | 0.9403 | 0.9479 | 0.9151 | 0.9819 |
| SpamBase | 0.8136 | 0.7885 | 0.6764 | 0.8184 | 0.8902 | 0.7126 | 0.5721 | 0.7785 | 0.8253 | 0.818 | 0.5534 | 0.4789 | 0.6573 | 0.838 | 0.8078 | 0.8433 | 0.8413 |
| Speech | 0.0279 | 0.0197 | 0.0315 | 0.0277 | 0.0353 | 0.0287 | 0.0515 | 0.04 | 0.027 | 0.0262 | 0.0424 | 0.0386 | 0.0335 | 0.0285 | 0.038 | 0.0584 | 0.0584 |
| Thyroid | 0.8134 | 0.5903 | 0.7892 | 0.8134 | 0.6055 | 0.6807 | 0.1095 | 0.7282 | 0.8096 | 0.6655 | 0.6931 | 0.7435 | 0.6575 | 0.8167 | 0.8417 | 0.8179 | 0.8626 |
| Vertebral | 0.1517 | 0.1239 | 0.2063 | 0.1381 | 0.1342 | 0.1917 | 0.1954 | 0.159 | 0.1476 | 0.1544 | 0.1451 | 0.1658 | 0.1598 | 0.2514 | 0.1949 | 0.2279 | 0.2854 |
| Vowels | 0.2969 | 0.3146 | 0.3277 | 0.1051 | 0.0984 | 0.1772 | 0.0843 | 0.1717 | 0.3475 | 0.1183 | 0.2274 | 0.1224 | 0.1574 | 0.381 | 0.0977 | 0.9498 | 0.4506 |
| WDBC | 0.4348 | 0.9323 | 0.9573 | 0.9833 | 0.9749 | 0.7734 | 0.7105 | 0.9184 | 0.9591 | 0.9383 | 0.9276 | 0.6372 | 0.8821 | 0.6882 | 0.889 | 0.9112 | 1 |
| WPBC | 0.3974 | 0.3882 | 0.3878 | 0.394 | 0.376 | 0.3525 | 0.3981 | 0.3903 | 0.3848 | 0.4287 | 0.3738 | 0.3631 | 0.4238 | 0.4935 | 0.402 | 0.4005 | 0.5017 |
| Wbc | 0.8391 | 0.8022 | 0.8412 | 0.8391 | 0.8573 | 0.7217 | 0.2959 | 0.834 | 0.8578 | 0.775 | 0.3362 | 0.6051 | 0.7218 | 0.3997 | 0.8887 | 0.8079 | 0.9742 |
| Wilt | 0.2254 | 0.1496 | 0.2233 | 0.0641 | 0.0848 | 0.0768 | 0.0997 | 0.0705 | 0.0867 | 0.1246 | 0.1061 | 0.244 | 0.1382 | 0.2541 | 0.0759 | 0.0717 | 0.4543 |
| Wine | 0.1424 | 0.9917 | 0.1253 | 0.1325 | 0.2458 | 0.3578 | 0.4907 | 0.1476 | 0.1605 | 0.5852 | 0.1619 | 0.9078 | 0.5659 | 0.9985 | 0.9335 | 0.7746 | 1 |
| Yeast | 0.4803 | 0.4737 | 0.4866 | 0.4678 | 0.4654 | 0.4943 | 0.5411 | 0.4953 | 0.4833 | 0.4869 | 0.5877 | 0.5458 | 0.5097 | 0.4974 | 0.4631 | 0.4888 | 0.5416 |
| Average AUC-PR | 0.4957 | 0.6109 | 0.5858 | 0.5532 | 0.5391 | 0.5059 | 0.3602 | 0.5254 | 0.5413 | 0.5693 | 0.4599 | 0.5553 | 0.5687 | 0.5815 | 0.605 | 0.6362 | **0.7344** |
| Average ranking | 10.25 | 9.425 | 8.2875 | 8.85 | 9.675 | 10.775 | 13.55 | 10.3625 | 8.7875 | 9.3875 | 11.725 | 9.0875 | 8.3875 | 8.5125 | 7.475 | 6.4875 | **1.975** |
| Win | 0 | 1 | 1 | 3 | 2 | 0 | 0 | 1 | 1 | 1 | 2 | 3 | 3 | 1 | 1 | 8 | **22** |
| Average std | 0 | 0 | 0 | 0 | 0.017 | 0 | 0.056 | 0.038 | 0.022 | 0.042 | 0.015 | 0.021 | 0.033 | 0.048 | 0.064 | 0.004 | 0.028 |

Table 6: Comparison of AUC-ROC (↑) results between baseline methods and DRL on 40 datasets.

| | OCSVM | KNN | LOF | PCA | IForest | ECOD | DAGMM | DeepSVDD | AutoEncoder | RAPP | GOAD | NeuTraLAD | ICL | DTE | MCM | NPT-AD | **DRL (ours)** |
|---|---|---|---|---|---|---|---|---|---|---|---|---|---|---|---|---|---|
| Abalone | 0.7237 | 0.7894 | 0.7154 | 0.7044 | 0.7351 | 0.4867 | 0.6785 | 0.5468 | 0.7019 | 0.6842 | 0.7407 | 0.6992 | 0.6208 | 0.4624 | 0.5822 | 0.9143 | 0.8086 |
| Amazon | 0.5418 | 0.5768 | 0.584 | 0.549 | 0.5593 | 0.5379 | 0.5256 | 0.5095 | 0.6004 | 0.6095 | 0.5584 | 0.5187 | 0.5596 | 0.5671 | 0.5615 | 0.4617 | 0.5699 |
| Annthyroid | 0.5551 | 0.6903 | 0.7216 | 0.8519 | 0.9112 | 0.7845 | 0.615 | 0.5678 | 0.7295 | 0.6536 | 0.6008 | 0.8196 | 0.6997 | 0.909 | 0.6894 | 0.8682 | 0.9239 |
| Arrhythmia | 0.7689 | 0.7933 | 0.7688 | 0.7684 | 0.7734 | 0.7199 | 0.7283 | 0.7941 | 0.5682 | 0.6676 | 0.581 | 0.79 | 0.8145 | 0.5912 | 0.8114 | 0.7185 | 0.7742 |
| Backdoor | 0.8464 | 0.9379 | 0.9344 | 0.7193 | 0.7461 | 0.8385 | 0.7143 | 0.861 | 0.9354 | 0.9529 | 0.7228 | 0.926 | 0.9154 | 0.9165 | 0.9641 | 0.9554 | 0.9788 |
| Breastw | 0.9938 | 0.9714 | 0.9937 | 0.9938 | 0.9719 | 0.9649 | 0.7605 | 0.9925 | 0.9909 | 0.9777 | 0.7327 | 0.9458 | 0.9725 | 0.9278 | 0.9955 | 0.9848 | 0.9968 |
| Campaign | 0.7626 | 0.7324 | 0.7061 | 0.7707 | 0.7269 | 0.7555 | 0.5785 | 0.5254 | 0.8011 | 0.7348 | 0.453 | 0.6982 | 0.7241 | 0.7595 | 0.8902 | 0.7915 | 0.7714 |
| Cardio | 0.9654 | 0.8979 | 0.9562 | 0.9655 | 0.922 | 0.637 | 0.6951 | 0.9313 | 0.567 | 0.8729 | 0.6812 | 0.7349 | 0.9514 | 0.8726 | 0.9603 | 0.9453 | 0.9562 |
| Cardiotocography | 0.7522 | 0.6925 | 0.6449 | 0.7889 | 0.7249 | 0.7889 | 0.6101 | 0.5149 | 0.5227 | 0.7639 | 0.4428 | 0.7177 | 0.6478 | 0.6013 | 0.8001 | 0.7187 | 0.84 |
| Census | 0.7158 | 0.6584 | 0.711 | 0.7075 | 0.6015 | 0.5772 | 0.6327 | 0.4506 | 0.7092 | 0.6678 | 0.5615 | 0.4685 | 0.6831 | 0.6962 | 0.7581 | 0.7715 | 0.7714 |
| Comm.and.crime | 0.7047 | 0.7269 | 0.72 | 0.7871 | 0.8044 | 0.517 | 0.5956 | 0.6941 | 0.7449 | 0.7682 | 0.8785 | 0.8517 | 0.8213 | 0.6134 | 0.7361 | 0.7621 | 0.8461 |
| Cover | 0.5003 | 0.8713 | 0.9121 | 0.8789 | 0.7185 | 0.8932 | 0.7659 | 0.5324 | 0.9277 | 0.7896 | 0.6276 | 0.7042 | 0.7393 | 0.9173 | 0.6095 | 0.5392 | 0.9804 |
| Fault | 0.5682 | 0.5795 | 0.4763 | 0.5587 | 0.5609 | 0.5037 | 0.5552 | 0.5342 | 0.6073 | 0.6019 | 0.5742 | 0.5753 | 0.5777 | 0.5864 | 0.6062 | 0.7621 | 0.6279 |
| Fraud | 0.9548 | 0.9235 | 0.9573 | 0.9445 | 0.963 | 0.8535 | 0.7271 | 0.9519 | 0.9335 | 0.9256 | 0.4808 | 0.8807 | 0.9107 | 0.9352 | 0.9357 | 0.9565 | 0.9314 |
| Glass | 0.548 | 0.6141 | 0.562 | 0.548 | 0.5771 | 0.6235 | 0.5982 | 0.5566 | 0.6088 | 0.6307 | 0.5741 | 0.6312 | 0.835 | 0.6964 | 0.7225 | 0.7875 | 0.6969 |
| Hepatitis | 0.4955 | 0.4682 | 0.5396 | 0.8122 | 0.7255 | 0.6991 | 0.6407 | 0.6621 | 0.5977 | 0.5701 | 0.5724 | 0.6584 | 0.5618 | 0.8022 | 0.5289 | 0.6629 | 0.8122 |
| Imgseg | 0.7414 | 0.8723 | 0.8407 | 0.6735 | 0.6859 | 0.6242 | 0.5304 | 0.5348 | 0.7991 | 0.7633 | 0.6824 | 0.8726 | 0.8717 | 0.7252 | 0.8134 | 0.8471 | 0.9093 |
| Ionosphere | 0.8765 | 0.9167 | 0.9454 | 0.8765 | 0.9683 | 0.9569 | 0.7041 | 0.8552 | 0.6231 | 0.8965 | 0.8993 | 0.9453 | 0.971 | 0.9542 | 0.9726 | 0.9735 | 0.984 |
| Lympho | 0.9812 | 0.987 | 0.9977 | 1 | 0.9945 | 0.9906 | 0.9343 | 0.9977 | 0.7856 | 0.9648 | 0.9757 | 0.9648 | 0.9546 | 0.9899 | 0.9257 | 0.9993 | 1 |
| Mammography | 0.9003 | 0.864 | 0.8922 | 0.8993 | 0.822 | 0.8251 | 0.7412 | 0.8879 | 0.8472 | 0.8504 | 0.7348 | 0.7449 | 0.6548 | 0.8862 | 0.9053 | 0.8873 | 0.9455 |
| Mnist | 0.5 | 0.9265 | 0.9511 | 0.9022 | 0.8623 | 0.7487 | 0.6119 | 0.6371 | 0.5151 | 0.9289 | 0.9201 | 0.9737 | 0.9646 | 0.8743 | 0.8743 | 0.9436 | 0.9712 |
| Musk | 0.5 | 0.9917 | 1 | 1 | 0.9521 | 0.9987 | 0.3649 | 1 | 1 | 1 | 0.9543 | 1 | 1 | 1 | 0.9752 | 1 | 1 |
| Optdigits | 0.6338 | 0.9862 | 0.9665 | 0.5817 | 0.8239 | 0.6145 | 0.4706 | 0.7603 | 0.6694 | 0.9122 | 0.5972 | 0.8471 | 0.787 | 0.8238 | 0.9947 | 0.9317 | 0.9972 |
| Parkinson | 0.6043 | 0.4365 | 0.4206 | 0.6927 | 0.768 | 0.515 | 0.4828 | 0.6601 | 0.6943 | 0.4915 | 0.4688 | 0.4205 | 0.4747 | 0.4672 | 0.3661 | 0.816 | 0.6732 |
| Pendigits | 0.9636 | 0.9906 | 0.9905 | 0.9437 | 0.9666 | 0.9295 | 0.3982 | 0.4563 | 0.9937 | 0.9764 | 0.2141 | 0.9859 | 0.9142 | 0.9761 | 0.9919 | 0.9987 | 0.9979 |
| Pima | 0.7133 | 0.6723 | 0.6913 | 0.7133 | 0.6737 | 0.5834 | 0.6108 | 0.7348 | 0.7163 | 0.6628 | 0.4338 | 0.617 | 0.6727 | 0.6788 | 0.7369 | 0.7962 | 0.7651 |
| Satellite | 0.6663 | 0.8139 | 0.7391 | 0.6663 | 0.8026 | 0.7884 | 0.7259 | 0.7659 | 0.7233 | 0.7894 | 0.7374 | 0.808 | 0.8549 | 0.7661 | 0.7962 | 0.806 | 0.832 |
| Satimage-2 | 0.9817 | 0.9891 | 0.9961 | 0.9817 | 0.9938 | 0.965 | 0.8994 | 0.9881 | 0.9979 | 0.9915 | 0.9929 | 0.9979 | 0.9792 | 0.9967 | 0.9992 | 0.9993 | 0.9968 |
| Shuttle | 0.9969 | 0.9893 | 0.9983 | 0.9936 | 0.9961 | 0.9978 | 0.9049 | 0.9952 | 0.9944 | 0.9982 | 0.9897 | 0.9994 | 0.9935 | 0.9993 | 0.9975 | 0.9931 | 0.9992 |
| SpamBase | 0.7682 | 0.745 | 0.613 | 0.814 | 0.858 | 0.6883 | 0.5395 | 0.7328 | 0.8212 | 0.7776 | 0.3842 | 0.3824 | 0.6042 | 0.8174 | 0.8019 | 0.8068 | 0.8289 |
| Speech | 0.3673 | 0.3561 | 0.3759 | 0.3638 | 0.3812 | 0.3596 | 0.5178 | 0.5071 | 0.3633 | 0.3824 | 0.5065 | 0.4809 | 0.4883 | 0.3817 | 0.4409 | 0.5925 | 0.5821 |
| Thyroid | 0.9855 | 0.9525 | 0.9856 | 0.9855 | 0.9271 | 0.8827 | 0.7149 | 0.9887 | 0.978 | 0.9442 | 0.8523 | 0.9701 | 0.9518 | 0.9863 | 0.9804 | 0.9787 | 0.9911 |
| Vertebral | 0.2654 | 0.1171 | 0.4317 | 0.1746 | 0.1444 | 0.4124 | 0.4128 | 0.2706 | 0.2375 | 0.3058 | 0.2262 | 0.3121 | 0.2821 | 0.543 | 0.3767 | 0.5347 | 0.6216 |
| Vowels | 0.7557 | 0.8174 | 0.8564 | 0.5229 | 0.5902 | 0.6147 | 0.5581 | 0.5734 | 0.7905 | 0.6311 | 0.7247 | 0.7239 | 0.702 | 0.8142 | 0.6529 | 0.9935 | 0.8511 |
| WDBC | 0.9637 | 0.9728 | 0.9978 | 0.9989 | 0.9983 | 0.9793 | 0.6661 | 0.9879 | 0.9976 | 0.9959 | 0.995 | 0.9568 | 0.9898 | 0.9852 | 0.9419 | 0.9953 | 1 |
| WPBC | 0.47 | 0.4854 | 0.477 | 0.4686 | 0.4975 | 0.4706 | 0.4847 | 0.4956 | 0.4942 | 0.5487 | 0.4591 | 0.4709 | 0.5482 | 0.5916 | 0.5105 | 0.5424 | 0.6198 |
| Wbc | 0.9667 | 0.9536 | 0.967 | 0.9667 | 0.9715 | 0.8747 | 0.7999 | 0.9633 | 0.9737 | 0.9299 | 0.4806 | 0.9133 | 0.908 | 0.8054 | 0.9814 | 0.9622 | 0.9963 |
| Wilt | 0.7923 | 0.7295 | 0.7668 | 0.2607 | 0.4595 | 0.3748 | 0.5011 | 0.342 | 0.4633 | 0.6053 | 0.5589 | 0.8328 | 0.5332 | 0.6291 | 0.3741 | 0.3505 | 0.9357 |
| Wine | 0.485 | 0.9917 | 0.4083 | 0.4467 | 0.6571 | 0.7433 | 0.8833 | 0.5067 | 0.5356 | 0.8672 | 0.5366 | 0.9753 | 0.915 | 0.9944 | 0.9538 | 0.9622 | 1 |
| Yeast | 0.4483 | 0.4366 | 0.4571 | 0.4324 | 0.4095 | 0.4464 | 0.5297 | 0.479 | 0.4503 | 0.4645 | 0.6 | 0.5488 | 0.5076 | 0.4458 | 0.4259 | 0.4691 | 0.512 |
| Average AUC-ROC | 0.7181 | 0.7729 | 0.7667 | 0.7427 | 0.7556 | 0.7141 | 0.6382 | 0.6982 | 0.7253 | 0.7637 | 0.6427 | 0.7591 | 0.7639 | 0.7747 | 0.7756 | 0.8185 | **0.8574** |
| Average ranking | 10.3 | 9.3 | 7.8 | 9.4 | 8.975 | 11.625 | 13.425 | 10.65 | 8.9125 | 8.665 | 12.5 | 8.9125 | 9.0125 | 8.1875 | 7.275 | 5.6125 | **2.45** |
| Win | 0 | 0 | 1 | 3 | 2 | 0 | 0 | 1 | 1 | 2 | 2 | 3 | 4 | 1 | 1 | 9 | **20** |
| Average std | 0 | 0 | 0 | 0 | 0.012 | 0 | 0.047 | 0.034 | 0.032 | 0.024 | 0.009 | 0.017 | 0.026 | 0.029 | 0.046 | 0.003 | 0.013 |

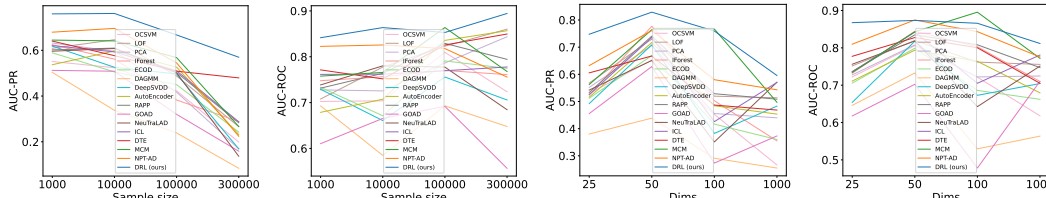

Figure 8: Model performance w.r.t. different sample size and dimensionality. We categorized all datasets into four intervals based on sample size: ≤ 1000, ≥ 1000 & ≤ 10000, ≥ 10000 & ≤ 100000, ≥ 100000 & ≤ 300000. For dimensionality, the intervals are defined as: ≤ 25, ≥ 25 & ≤ 50, ≥ 50 & ≤ 100, ≥ 100 & ≤ 1000. We provide the average performance for each interval across different datasets.

## A.5 Computational Cost

We provide the runtime in seconds of DRL for the training and inference phase on a single GTX 3090 GPU, as shown in Table 7.

Table 7: Runtime in seconds of DRL for the training and inference phase.

|  | Abalone | Amazon | Annthyroid | Arrhythmia | Backdoor | Breastw | Campaign | Cardio |
|---|---|---|---|---|---|---|---|---|
| Train | 2.4855 | 8.6840 | 6.5400 | 0.3842 | 77.9249 | 0.3972 | 32.1423 | 1.8538 |
| Inference | 0.0095 | 0.0225 | 0.0248 | 0.0027 | 0.1546 | 0.0030 | 0.0660 | 0.0091 |

|  | Cardiotocography | Census | Comm.and.crime | Cover | Fault | Fraud | Glass | Hepatitis |
|---|---|---|---|---|---|---|---|---|
| Train | 1.7553 | 247.3347 | 0.9703 | 236.5932 | 1.0365 | 233.1834 | 0.3029 | 0.2812 |
| Inference | 0.0085 | 0.5250 | 0.0106 | 0.3724 | 0.0109 | 0.3743 | 0.0022 | 0.0025 |

|  | Imgseg | Ionosphere | Lympho | Mammography | Mnist | Musk | Optdigits | Parkinson |
|---|---|---|---|---|---|---|---|---|
| Train | 1.6713 | 0.3324 | 0.2868 | 8.6523 | 5.2673 | 2.5212 | 3.8596 | 0.2775 |
| Inference | 0.0060 | 0.0025 | 0.0022 | 0.0256 | 0.0145 | 0.0060 | 0.0127 | 0.0029 |

|  | Pendigits | Pima | Satellite | Satimage-2 | Shuttle | SpamBase | Speech | Thyroid |
|---|---|---|---|---|---|---|---|---|
| Train | 5.8137 | 0.7752 | 3.6661 | 4.9489 | 40.0011 | 2.6029 | 2.8119 | 3.5806 |
| Inference | 0.0103 | 0.0067 | 0.0172 | 0.0163 | 0.0929 | 0.0175 | 0.0128 | 0.0094 |

|  | Vertebral | Vowels | WDBC | WPBC | Wbc | Wilt | Wine | Yeast |
|---|---|---|---|---|---|---|---|---|
| Train | 0.3003 | 1.5365 | 0.3250 | 0.2854 | 0.3436 | 4.0054 | 0.2858 | 0.9532 |
| Inference | 0.0023 | 0.0065 | 0.0024 | 0.0024 | 0.0024 | 0.0080 | 0.0022 | 0.0091 |

Table 8: Runtime in seconds of DRL and other baseline methods for the training and inference phase averaged over all datasets.

|  | OCSVM | KNN | LOF | PCA | IForest | ECOD | DAGMM | DeepSVDD | AutoEncoder | RAPP | GOAD | NeuTraL | ICL | MCM | NPT-AD | DRL |
|---|---|---|---|---|---|---|---|---|---|---|---|---|---|---|---|---|
| Train | 224.5780 | 16.7915 | 13.9255 | 0.4415 | 1.3064 | 0.2819 | 5.2457 | 45.5777 | 45.3946 | 20.5383 | 326.4836 | 107.5398 | 188.0606 | 78.2938 | 811.0800 | 23.6743 |
| Inference | 95.4769 | 236.1779 | 15.1572 | 0.4116 | 0.3713 | 0.6268 | 1.6679 | 0.3185 | 0.1404 | 0.0930 | 77.8951 | 4.2395 | 11.0595 | 1.8856 | 182.5385 | 0.0472 |

## A.6 Hyper Parameter Sensitivity Analysis

We incorporate the sensitivity analysis for the number of basis vectors, the loss weight $\lambda_1$, $\lambda_2$, the number of training epochs, and the number of batch size in Fig. 9.

## A.7 Representation Weight of Weight Learner Visualization

The weight visualization in Fig. 10 shows that the separation constraint effectively diversifies the weights of normal representations, while the weights for anomalous representations differ significantly from those of normal representations.

## A.8 Robustness to Anomaly Contamination

Our method is implemented in a one-class classification setting where the training set consists entirely of normal data. However, real-world anomaly detection applications frequently encounter contaminated training sets, requiring models to be robust against small levels of dataset contamination. To analyse the robustness of DRL w.r.t anomaly contamination, we conduct experiments in the case of anomaly contamination ratio of 0%, 1%, 2%, 3%, 4% and 5%, as shown in Fig. 11. The results indicate that DRL exhibits greater stability and consistently outperforms other models, demonstrating its superior robustness to anomaly contamination.

## A.9 Performance on Different Types of Anomalies

While extensive public datasets are available for benchmarking, they often contain a mix of different types of anomalies Existing works (Han et al., 2022) have summarized four common types of anomalies and proposed methods for generating them. Following this framework, we conducted experiments to generate synthetic anomalies based on the realistic Shuttle dataset and assess the

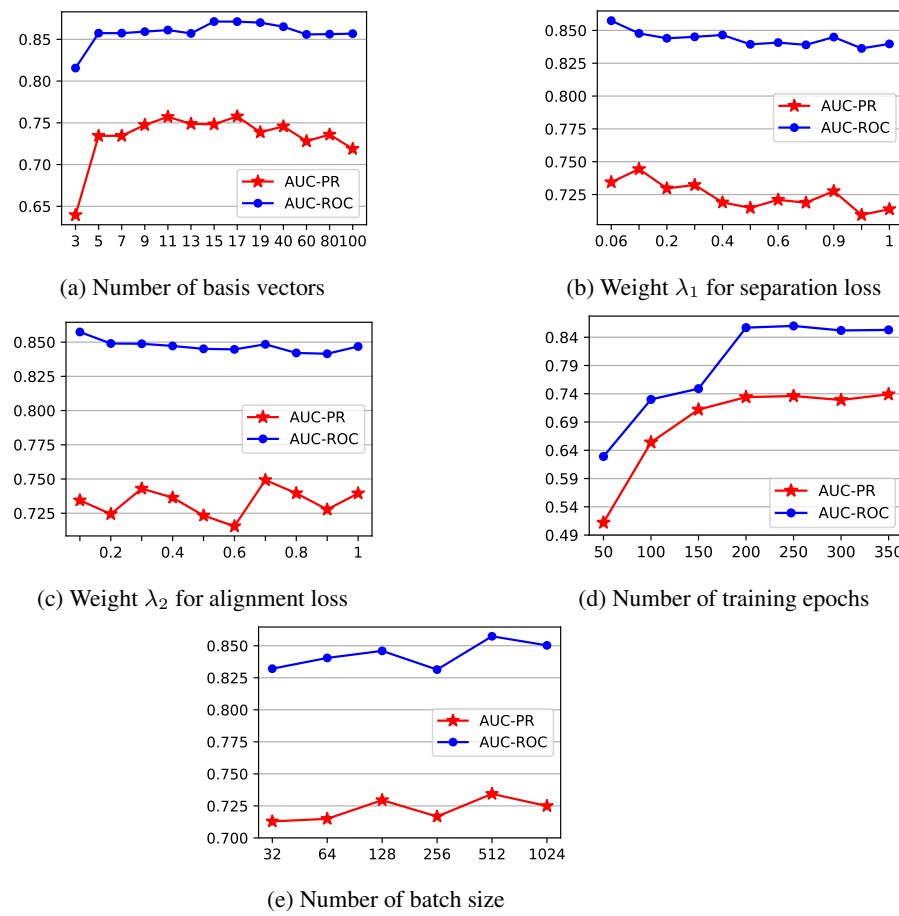

Figure 9: Sensitivity analysis. The results are averaged over all datasets.

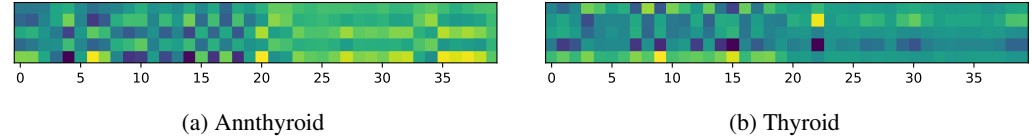

(a) Annthyroid       (b) Thyroid

Figure 10: Weight visualization of normal and anomalous representations. For each subfigure, the left 20 columns and right 20 columns correspond to the weights of normal and anomalous representations respectively. For each dataset, we randomly select 20 normal samples and 20 anomalous samples for visualization.

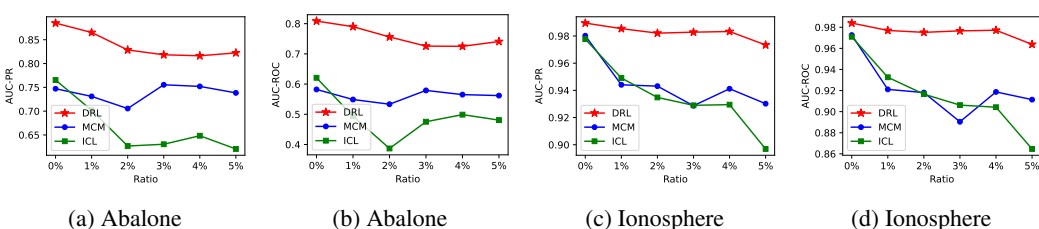

(a) Abalone  (b) Abalone  (c) Ionosphere  (d) Ionosphere

Figure 11: Model performance w.r.t. different contamination ratios (the percentage of anomalies in the training set).

performance of DRL on these specific anomaly types. For details on the generation process of the four types of anomalies, please refer to (Han et al., 2022). The four types of anomalies are as follows: *Local anomalies* refer to the anomalies that are deviant from their local neighborhoods; *Global anomalies* are samples scattering throughout the entire space, while being far from the normal data distribution; *Dependency anomalies* represent samples that do not adhere to the correlation structure among normal data; *Clustered anomalies* refer to groups of data with similar characteristics but significantly different from normal data. The results on these different anomaly types are provided in Table 9. The results show that DRL exhibits strong performance across all anomaly types.

Table 9: The AUC-PR results on different types of anomalies.

|  | Iforest | LOF | OCSVM | ECOD | NeuTralAD | ICL | MCM | NPT-AD | DRL |
|---|---|---|---|---|---|---|---|---|---|
| Local | 0.3494 | 0.5256 | 0.3847 | 0.4324 | 0.4247 | 0.3284 | 0.5103 | 0.5125 | **0.6742** |
| Cluster | 0.9888 | 0.9772 | 0.9909 | 0.9272 | 0.9995 | 1 | 1 | 1 | **1** |
| Dependency | 0.6333 | 0.325 | 0.5 | 0.625 | 0.1213 | 0.3333 | 0.6429 | 0.673 | **0.6913** |
| Global | 0.9992 | 0.9453 | 0.9847 | 0.9996 | 0.999 | 0.9992 | 1 | 1 | **1** |

## A.10 ADDITIONAL EXPERIMENT RESULTS

Table 10: The comparison between $\mathbb{E}\left[\|\mathbf{w}_a - \mu_{\mathbf{w}_n}\|_2^2\right]$ and $\mathbb{E}\left[\|\mathbf{w}_n - \mu_{\mathbf{w}_n}\|_2^2\right]$ under the influence of separation loss over all datasets. $\mathbf{w}_n$ and $\mathbf{w}_a$ denote the computed weights of normal and anomalous samples respectively. $\mu_{\mathbf{w}_n} = \mathbb{E}\left[\mathbf{w}_n\right]$, represents the the center of normal weights.

|  |  | abalone | amazon | annthyroid | arrhythmia | backdoor | breastw | campaign | cardio | Cardiotocography | census |
|---|---|---|---|---|---|---|---|---|---|---|---|
| w/o separation | $\mathbb{E}\left[\|\mathbf{w}_a - \mu_{\mathbf{w}_n}\|_2^2\right]$ | 0.2550 | 0.1126 | 0.1717 | 0.5577 | 0.1916 | 0.1738 | 0.0057 | 0.3347 | 0.2055 | 0.0270 |
|  | $\mathbb{E}\left[\|\mathbf{w}_n - \mu_{\mathbf{w}_n}\|_2^2\right]$ | 0.1411 | 0.1021 | 0.0744 | 0.3935 | 0.0995 | 0.0307 | 0.0041 | 0.1836 | 0.1439 | 0.0274 |
|  | Gap | 0.1139 | 0.0105 | 0.0973 | 0.1642 | 0.0921 | 0.1431 | 0.0016 | 0.1511 | 0.0616 | -0.0004 |
| w/ separation | $\mathbb{E}\left[\|\mathbf{w}_a - \mu_{\mathbf{w}_n}\|_2^2\right]$ | 0.5647 | 0.3276 | 0.3049 | 0.6160 | 0.2893 | 0.2080 | 0.8078 | 0.3935 | 0.2411 | 0.8362 |
|  | $\mathbb{E}\left[\|\mathbf{w}_n - \mu_{\mathbf{w}_n}\|_2^2\right]$ | 0.2316 | 0.1928 | 0.1859 | 0.4333 | 0.1678 | 0.0480 | 0.1964 | 0.2077 | 0.1739 | 0.1206 |
|  | Gap | 0.3331 | 0.1348 | 0.119 | 0.1827 | 0.1215 | 0.16 | 0.6114 | 0.1858 | 0.0672 | 0.7156 |

|  |  | comm.and.crime | cover | fault | fraud | glass | Hepatitis | imgseg | ionosphere | lympho | mammography |
|---|---|---|---|---|---|---|---|---|---|---|---|
| w/o separation | $\mathbb{E}\left[\|\mathbf{w}_a - \mu_{\mathbf{w}_n}\|_2^2\right]$ | 0.3816 | 0.2953 | 0.1154 | 0.1011 | 0.1766 | 0.0853 | 0.0645 | 0.1679 | 0.3146 | 0.2532 |
|  | $\mathbb{E}\left[\|\mathbf{w}_n - \mu_{\mathbf{w}_n}\|_2^2\right]$ | 0.0794 | 0.0537 | 0.1353 | 0.0233 | 0.1570 | 0.0737 | 0.0655 | 0.0676 | 0.0437 | 0.1768 |
|  | Gap | 0.3022 | 0.2416 | -0.0199 | 0.0778 | 0.0196 | 0.0116 | -0.001 | 0.1003 | 0.2709 | 0.0764 |
| w/ separation | $\mathbb{E}\left[\|\mathbf{w}_a - \mu_{\mathbf{w}_n}\|_2^2\right]$ | 0.7966 | 0.3140 | 0.1681 | 0.1914 | 0.3291 | 0.1919 | 0.3160 | 0.1949 | 0.4917 | 0.4266 |
|  | $\mathbb{E}\left[\|\mathbf{w}_n - \mu_{\mathbf{w}_n}\|_2^2\right]$ | 0.1136 | 0.0668 | 0.1574 | 0.1135 | 0.2207 | 0.1749 | 0.1422 | 0.0853 | 0.1512 | 0.2022 |
|  | Gap | 0.683 | 0.2472 | 0.0107 | 0.0779 | 0.1084 | 0.017 | 0.1738 | 0.1096 | 0.3405 | 0.2244 |

|  |  | mnist | musk | optdigits | Parkinson | pendigits | pima | satellite | satimage-2 | shuttle | SpamBase |
|---|---|---|---|---|---|---|---|---|---|---|---|
| w/o separation | $\mathbb{E}\left[\|\mathbf{w}_a - \mu_{\mathbf{w}_n}\|_2^2\right]$ | 0.7646 | 0.2715 | 0.0612 | 0.4004 | 0.0537 | 0.4218 | 0.1488 | 0.0981 | 0.3905 | 0.1306 |
|  | $\mathbb{E}\left[\|\mathbf{w}_n - \mu_{\mathbf{w}_n}\|_2^2\right]$ | 0.6229 | 0.1382 | 0.0553 | 0.2336 | 0.0399 | 0.3810 | 0.1460 | 0.0493 | 0.1179 | 0.0542 |
|  | Gap | 0.1417 | 0.1333 | 0.0059 | 0.1668 | 0.0138 | 0.0408 | 0.0028 | 0.0488 | 0.2726 | 0.0764 |
| w/ separation | $\mathbb{E}\left[\|\mathbf{w}_a - \mu_{\mathbf{w}_n}\|_2^2\right]$ | 0.7950 | 0.3669 | 0.0807 | 0.5102 | 0.1953 | 0.4723 | 0.1966 | 0.1601 | 0.4801 | 0.3845 |
|  | $\mathbb{E}\left[\|\mathbf{w}_n - \mu_{\mathbf{w}_n}\|_2^2\right]$ | 0.6390 | 0.1457 | 0.0653 | 0.2582 | 0.1527 | 0.4199 | 0.1672 | 0.0676 | 0.1548 | 0.1060 |
|  | Gap | 0.156 | 0.2212 | 0.0154 | 0.252 | 0.0426 | 0.0524 | 0.0294 | 0.0925 | 0.3253 | 0.2785 |

|  |  | speech | thyroid | vertebral | vowels | WDBC | WPBC | wbc | Wilt | wine | yeast |
|---|---|---|---|---|---|---|---|---|---|---|---|
| w/o separation | $\mathbb{E}\left[\|\mathbf{w}_a - \mu_{\mathbf{w}_i}\|_2^2\right]$ | 0.0422 | 0.2963 | 0.0336 | 0.0736 | 0.3867 | 0.0403 | 0.3776 | 0.0769 | 0.7894 | 0.0631 |
|  | $\mathbb{E}\left[\|\mathbf{w}_n - \mu_{\mathbf{w}_n}\|_2^2\right]$ | 0.0333 | 0.0958 | 0.0504 | 0.1161 | 0.0757 | 0.0432 | 0.0830 | 0.0808 | 0.5375 | 0.0672 |
|  | Gap | 0.0089 | 0.2005 | -0.0168 | -0.0425 | 0.311 | -0.0029 | 0.2946 | -0.0039 | 0.2519 | -0.0041 |
| w/ separation | $\mathbb{E}\left[\|\mathbf{w}_a - \mu_{\mathbf{w}_n}\|_2^2\right]$ | 0.2306 | 0.4663 | 0.0991 | 0.1912 | 0.4423 | 0.2614 | 0.4883 | 0.2243 | 0.8883 | 0.2191 |
|  | $\mathbb{E}\left[\|\mathbf{w}_n - \mu_{\mathbf{w}_n}\|_2^2\right]$ | 0.1303 | 0.1142 | 0.0645 | 0.1472 | 0.0893 | 0.0621 | 0.1037 | 0.1129 | 0.5874 | 0.1091 |
|  | Gap | 0.1003 | 0.3521 | 0.0346 | 0.044 | 0.353 | 0.1993 | 0.3846 | 0.1114 | 0.3009 | 0.11 |

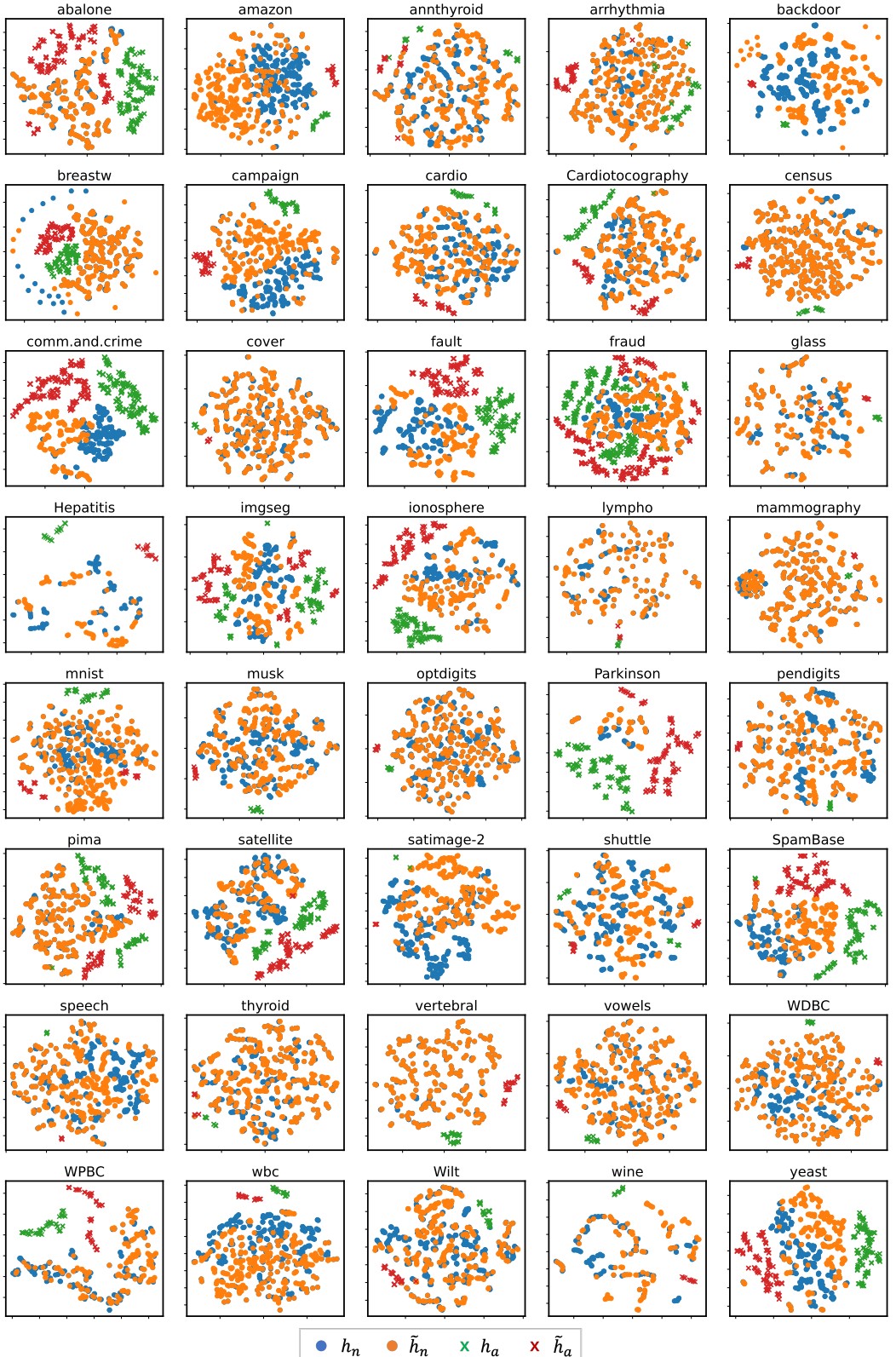

Figure 12: The T-SNE visualization of representation $\mathbf{h}$ by feature extractor $f$ and the reconstructed representation $\tilde{\mathbf{h}}$ by linear combination of basis vectors $\mathbf{w}\mathcal{B}$ over all datasets. $\mathbf{h}_n$ and $\mathbf{h}_a$ correspond to normal and anomalous representations respectively.

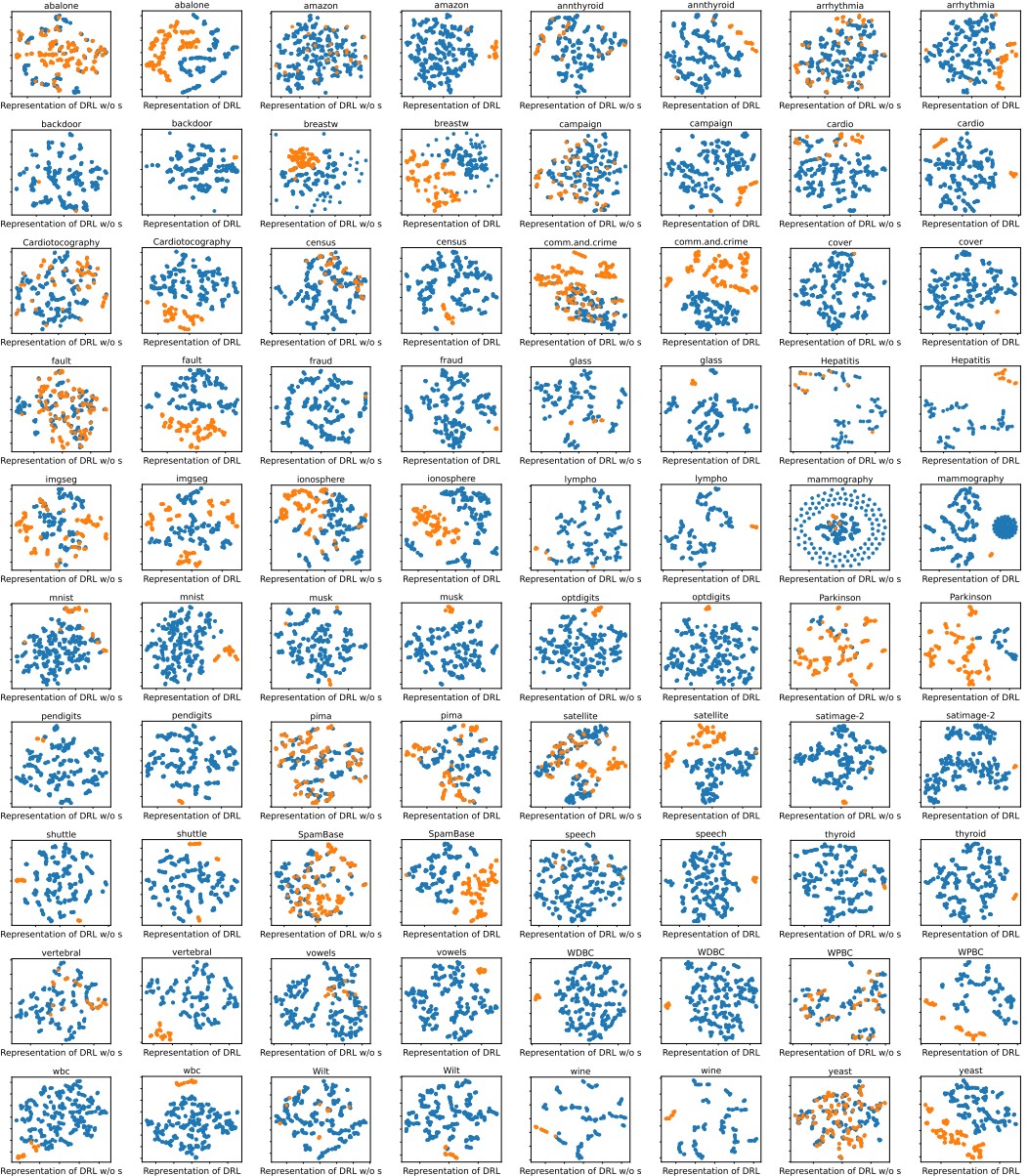

Figure 13: T-SNE visualization of normal and anomalous representations extracted by the feature extractor $f$, illustrating the impact of separation loss across all datasets. Each pair of subfigures corresponds to the same dataset; for instance, the first and second subfigures in the first row correspond to representations without and with separation loss, respectively.

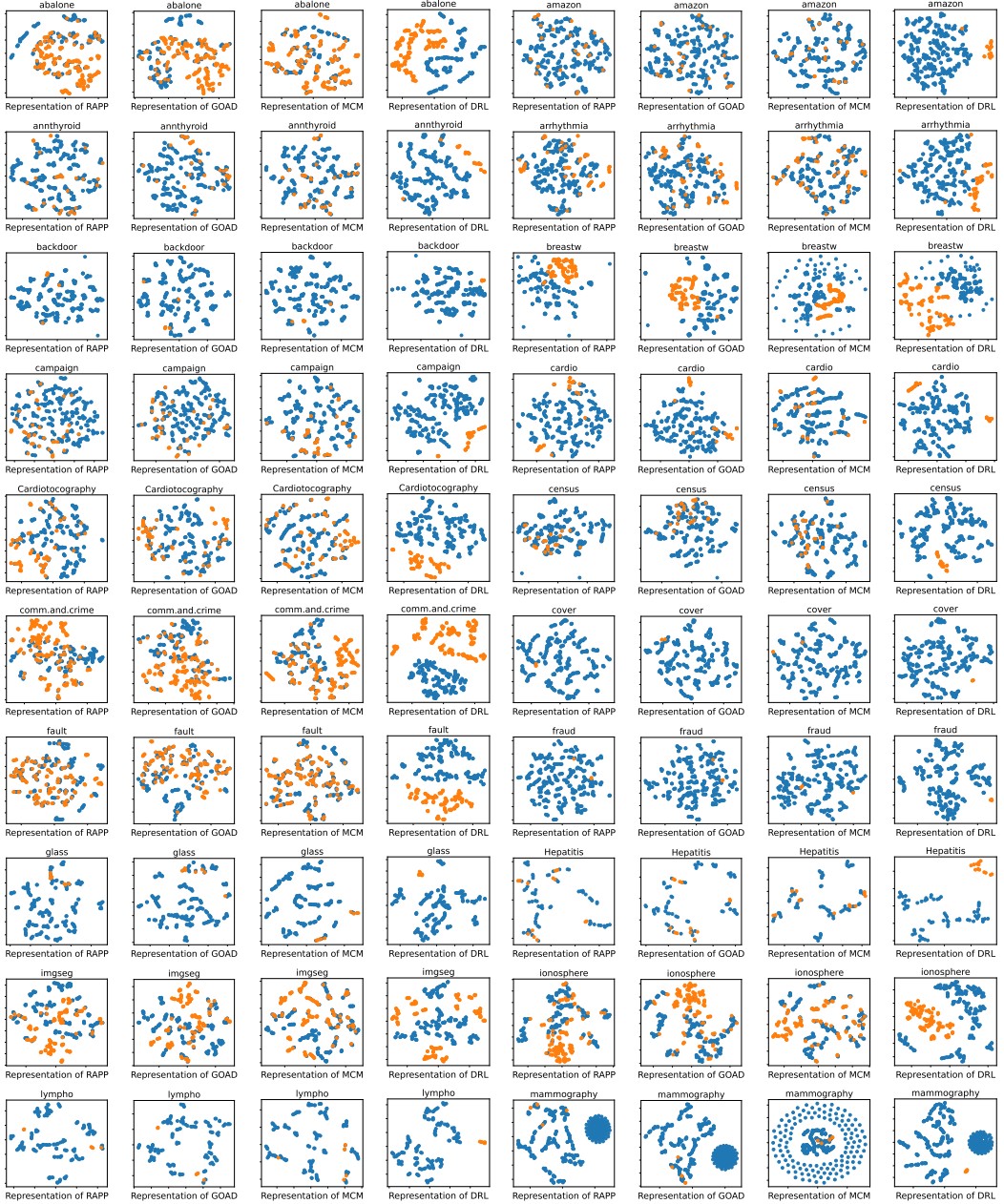

Figure 14: T-SNE Visualization of different deep TAD models' latent representations across all datasets (the former 20 datasets). This figure serves as an extension of Fig. 1, with each group of four subfigures corresponding to the same dataset.

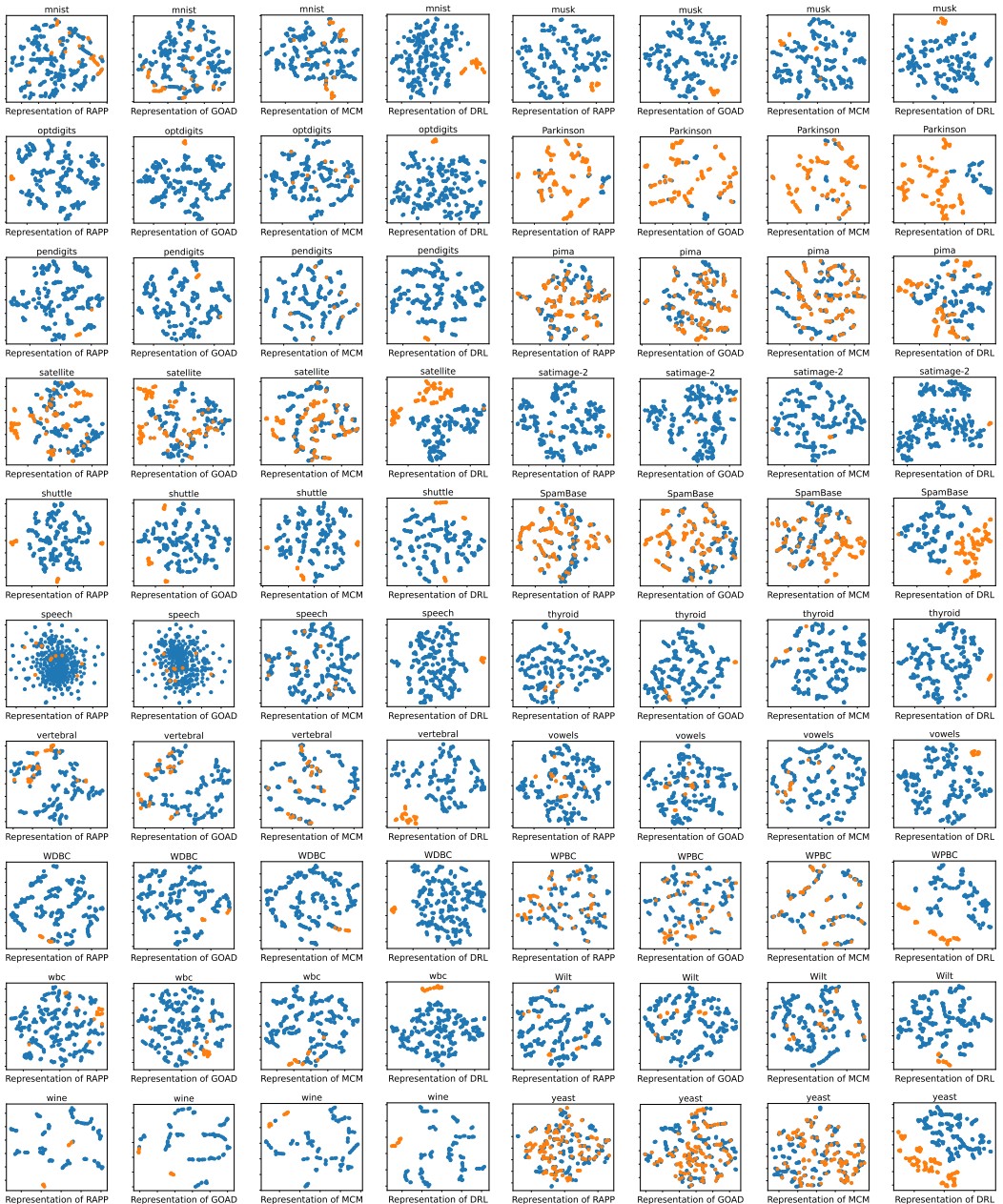

Figure 15: T-SNE Visualization of different deep TAD models' latent representations across all datasets (the latter 20 datasets). This figure serves as an extension of Fig. 1, with each group of four subfigures corresponding to the same dataset.

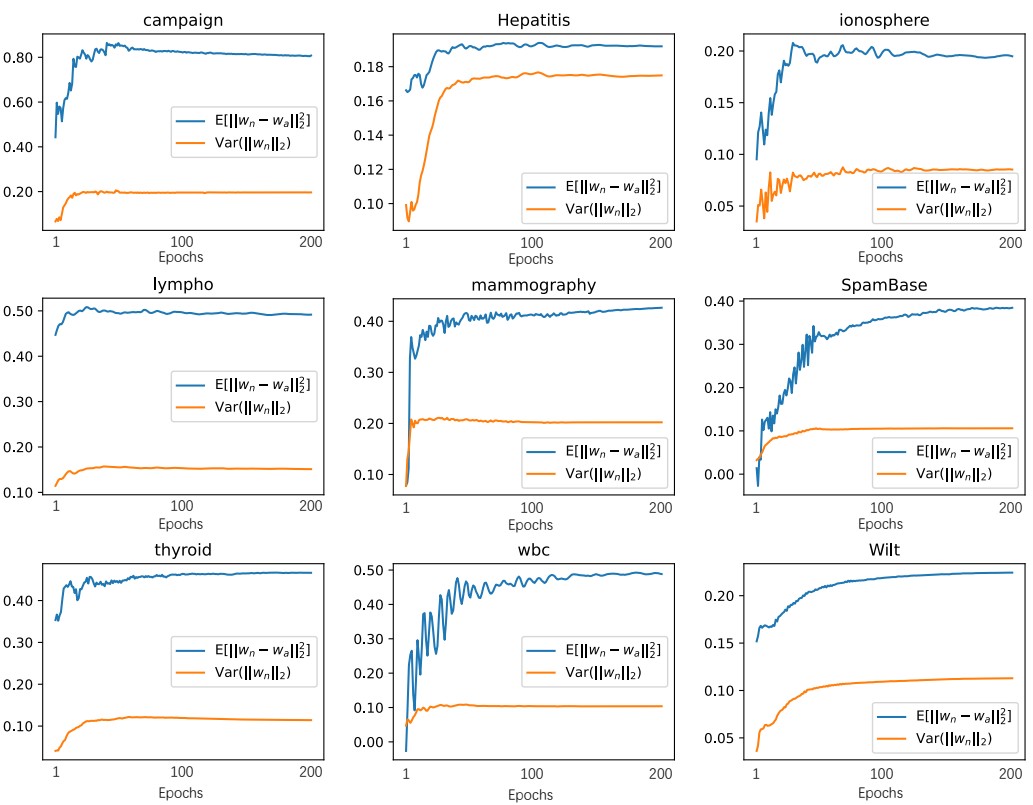

Figure 16: Progression of $\mathbb{E}\left[\|\mathbf{w}_n - \mathbf{w}_a\|_2^2\right]$ and $\mathrm{Var}(\|\mathbf{w}_n\|_2)$ during training. The results could verify Proposition 1 in Section 3.2.2. $\mathbf{w}_n$ and $\mathbf{w}_a$ denote the computed weights of normal and anomalous samples respectively.

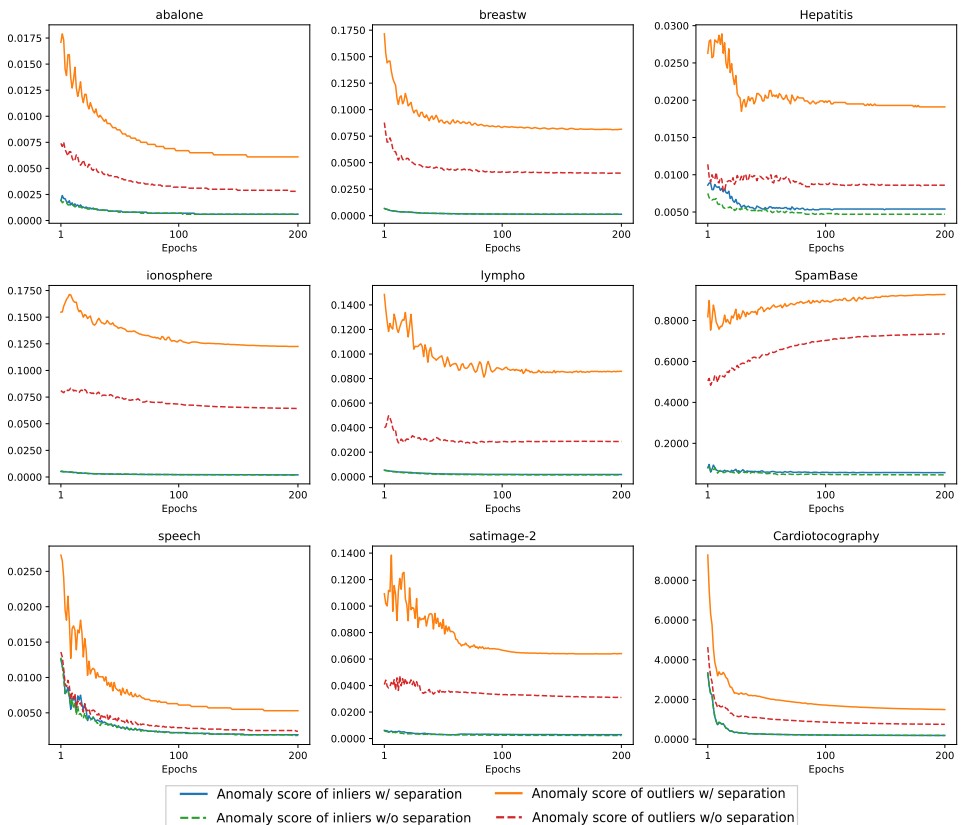

Figure 17: Visualization of test samples' anomaly score by DRL with and without separation loss as training progresses. We report the average anomaly score of normal and anomalous test samples respectively. The anomaly score is calculated by decomposition loss (Eq. 3).

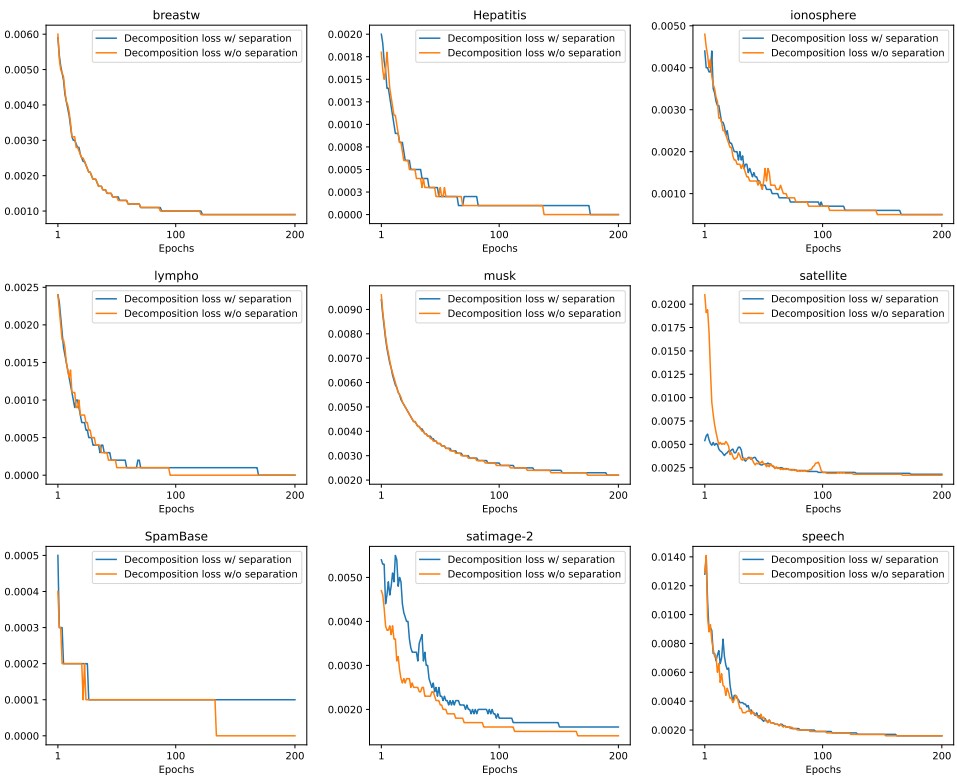

Figure 18: Effect of separation loss on decomposition loss (Eq. 3) convergence. We report the average decomposition loss of training samples. The dataset dimension is as follows, breastw: 9; Hepatitis: 19; ionosphere: 33; lympho: 18; musk: 166; satellite: 36; SpamBase: 57; satimage-2: 36; speech: 400.

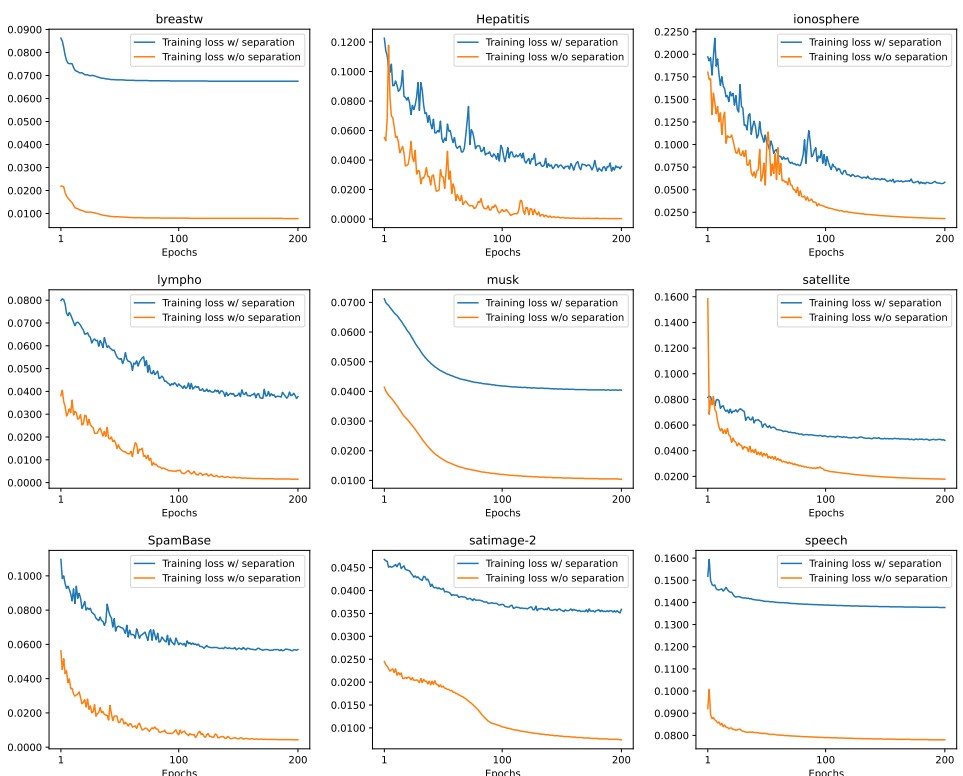

Figure 19: Effect of separation loss on overall training loss (Eq. 6) convergence. We report the average overall training loss of training samples. The dataset dimension is as follows, breastw: 9; Hepatitis: 19; ionosphere: 33; lympho: 18; musk: 166; satellite: 36; SpamBase: 57; satimage-2: 36; speech: 400.

Table 11: Full analysis on the effects of different components in DRL in terms of AUC-PR. This table serves as an extension of Table 2.

| | A | B | C | D | E | F | G | H | I | J | DRL (ours) |
|---|---|---|---|---|---|---|---|---|---|---|---|
| Abalone | 0.8764 | 0.8786 | 0.8804 | 0.7281 | 0.6711 | 0.8827 | 0.7974 | 0.8453 | 0.8573 | 0.7621 | 0.885 |
| Amazon | 0.1067 | 0.1079 | 0.1119 | 0.0139 | 0.1073 | 0.1125 | 0.1027 | 0.1160 | 0.1215 | 0.1096 | 0.1206 |
| Annthyroid | 0.4534 | 0.5666 | 0.6775 | 0.3213 | 0.2802 | 0.6702 | 0.4729 | 0.5326 | 0.5629 | 0.6260 | 0.6761 |
| Arrhythmia | 0.6261 | 0.6266 | 0.6267 | 0.4174 | 0.4257 | 0.5967 | 0.5413 | 0.5952 | 0.627 | 0.5827 | 0.627 |
| Backdoor | 0.7504 | 0.8742 | 0.8845 | 0.5214 | 0.1093 | 0.841 | 0.5174 | 0.7487 | 0.8941 | 0.8787 | 0.8915 |
| Breastw | 0.9974 | 0.9975 | 0.9979 | 0.8959 | 0.9828 | 0.9981 | 0.9388 | 0.9939 | 0.9965 | 0.8956 | 0.9966 |
| Campaign | 0.4772 | 0.4995 | 0.5209 | 0.3968 | 0.1423 | 0.4875 | 0.4120 | 0.4742 | 0.4863 | 0.4264 | 0.5013 |
| Cardio | 0.8261 | 0.8671 | 0.8676 | 0.6247 | 0.8732 | 0.809 | 0.7663 | 0.8275 | 0.8322 | 0.7026 | 0.8325 |
| Cardiotocography | 0.7543 | 0.7543 | 0.755 | 0.6165 | 0.6128 | 0.7197 | 0.6678 | 0.7103 | 0.754 | 0.7147 | 0.754 |
| Census | 0.1667 | 0.2339 | 0.2731 | 0.1918 | 0.4062 | 0.1717 | 0.1927 | 0.2474 | 0.2642 | 0.1469 | 0.2649 |
| Comm.and.crime | 0.917 | 0.9179 | 0.9183 | 0.7525 | 0.7524 | 0.89 | 0.8415 | 0.8957 | 0.917 | 0.8111 | 0.9164 |
| Cover | 0.8044 | 0.8264 | 0.8602 | 0.286 | 0.0107 | 0.8821 | 0.7110 | 0.7692 | 0.783 | 0.6998 | 0.783 |
| Fault | 0.6245 | 0.6372 | 0.6454 | 0.5143 | 0.5939 | 0.6425 | 0.5667 | 0.6227 | 0.6563 | 0.6059 | 0.6649 |
| Fraud | 0.0703 | 0.4723 | 0.6597 | 0.3105 | 0.2165 | 0.5905 | 0.3987 | 0.6608 | 0.7586 | 0.7773 | 0.6614 |
| Glass | 0.0959 | 0.1052 | 0.1242 | 0.1076 | 0.0972 | 0.0969 | 0.1068 | 0.1550 | 0.167 | 0.0792 | 0.167 |
| Hepatitis | 0.5056 | 0.5138 | 0.5533 | 0.4155 | 0.6289 | 0.6467 | 0.0159 | 0.6248 | 0.6186 | 0.6421 | 0.6627 |
| Imgseg | 0.9088 | 0.9153 | 0.9306 | 0.6685 | 0.6954 | 0.9161 | 0.6947 | 0.9079 | 0.9098 | 0.8318 | 0.9238 |
| Ionosphere | 0.9804 | 0.9814 | 0.9827 | 0.9748 | 0.9039 | 0.9779 | 0.9225 | 0.9889 | 0.9909 | 0.8135 | 0.9895 |
| Lympho | 0.8179 | 0.8734 | 0.9762 | 0.8151 | 1 | 0.9151 | 0.9686 | 0.9970 | 1 | 0.8278 | 1 |
| Mammography | 0.3237 | 0.5114 | 0.5135 | 0.1945 | 0.5104 | 0.8107 | 0.2743 | 0.3196 | 0.3344 | 0.6452 | 0.8406 |
| Mnist | 0.882 | 0.8894 | 0.8894 | 0.6568 | 0.1092 | 0.7681 | 0.8167 | 0.8861 | 0.887 | 0.6484 | 0.887 |
| Musk | 0.9274 | 0.9533 | 1 | 0.8983 | 1 | 1 | 0.9637 | 0.9885 | 1 | 0.0327 | 1 |
| Optdigits | 0.6387 | 0.6405 | 0.6578 | 0.1194 | 0.0521 | 0.854 | 0.6032 | 0.6429 | 0.6459 | 0.7712 | 0.9356 |
| Parkinson | 0.92 | 0.9209 | 0.921 | 0.8221 | 0.9311 | 0.9199 | 0.7982 | 0.8347 | 0.921 | 0.9318 | 0.921 |
| Pendigits | 0.5909 | 0.6647 | 0.815 | 0.2225 | 0.4575 | 0.7704 | 0.7594 | 0.8010 | 0.7102 | 0.7157 | 0.936 |
| Pima | 0.7448 | 0.7449 | 0.745 | 0.6105 | 0.6905 | 0.6904 | 0.6602 | 0.7049 | 0.7449 | 0.6053 | 0.7449 |
| Satellite | 0.8428 | 0.8521 | 0.8652 | 0.7484 | 0.7711 | 0.8489 | 0.7857 | 0.8061 | 0.8634 | 0.7856 | 0.8692 |
| Satimage-2 | 0.2661 | 0.9621 | 0.9748 | 0.8812 | 0.9154 | 0.9293 | 0.9295 | 0.8725 | 0.9668 | 0.8340 | 0.9703 |
| Shuttle | 0.974 | 0.9796 | 0.9849 | 0.7824 | 0.9656 | 0.965 | 0.9113 | 0.9452 | 0.9823 | 0.8714 | 0.9819 |
| SpamBase | 0.8437 | 0.8486 | 0.8489 | 0.7133 | 0.8186 | 0.8157 | 0.7918 | 0.7865 | 0.8342 | 0.8401 | 0.8413 |
| Speech | 0.0482 | 0.055 | 0.0608 | 0.0321 | 0.0304 | 0.0466 | 0.0435 | 0.0455 | 0.0525 | 0.0245 | 0.0584 |
| Thyroid | 0.7967 | 0.7976 | 0.8364 | 0.5174 | 0.4413 | 0.839 | 0.4143 | 0.8063 | 0.8236 | 0.7357 | 0.8626 |
| Vertebral | 0.2216 | 0.2224 | 0.2242 | 0.1469 | 0.18 | 0.1726 | 0.2322 | 0.2460 | 0.2457 | 0.1689 | 0.2854 |
| Vowels | 0.3518 | 0.3993 | 0.4007 | 0.3411 | 0.1422 | 0.442 | 0.3429 | 0.3819 | 0.1964 | 0.3160 | 0.4506 |
| WDBC | 1 | 1 | 1 | 0.9 | 0.9909 | 0.9909 | 0.9894 | 0.9808 | 1 | 0.1167 | 1 |
| WPBC | 0.4126 | 0.4345 | 0.4696 | 0.338 | 0.4126 | 0.4068 | 0.4173 | 0.4439 | 0.3747 | 0.4574 | 0.5017 |
| Wbc | 0.9405 | 0.9522 | 0.9527 | 0.8159 | 0.8531 | 0.8119 | 0.9256 | 0.9408 | 0.9742 | 0.9111 | 0.9742 |
| Wilt | 0.4074 | 0.412 | 0.4226 | 0.195 | 0.3372 | 0.4207 | 0.4100 | 0.4295 | 0.4474 | 0.5401 | 0.4543 |
| Wine | 1 | 1 | 1 | 0.9909 | 1 | 1 | 0.9696 | 0.9184 | 1 | 0.9898 | 1 |
| Yeast | 0.4847 | 0.4864 | 0.4952 | 0.3882 | 0.5083 | 0.5073 | 0.3765 | 0.5064 | 0.5099 | 0.5240 | 0.5416 |
| Average | 0.6344 | 0.6844 | 0.7081 | 0.5222 | 0.5407 | 0.6964 | 0.6013 | 0.6750 | 0.6928 | 0.6100 | 0.7344 |

Table 12: Full analysis on the effects of different components in DRL in terms of AUC-ROC. This table serves as an extension of Table 2.

| | A | B | C | D | E | F | G | H | I | J | DRL (ours) |
|---|---|---|---|---|---|---|---|---|---|---|---|
| Abalone | 0.7575 | 0.7633 | 0.7638 | 0.6373 | 0.5582 | 0.7963 | 0.6050 | 0.5976 | 0.7833 | 0.7531 | 0.8086 |
| Amazon | 0.4891 | 0.5075 | 0.5171 | 0.4861 | 0.5497 | 0.5465 | 0.5466 | 0.5806 | 0.5227 | 0.5386 | 0.5699 |
| Annthyroid | 0.7381 | 0.8321 | 0.8927 | 0.7371 | 0.7211 | 0.8047 | 0.6219 | 0.6959 | 0.8805 | 0.7925 | 0.9239 |
| Arrhythmia | 0.7339 | 0.734 | 0.7341 | 0.6754 | 0.5584 | 0.7342 | 0.8022 | 0.8093 | 0.7541 | 0.5867 | 0.7742 |
| Backdoor | 0.929 | 0.9291 | 0.9425 | 0.8507 | 0.718 | 0.8995 | 0.9185 | 0.9646 | 0.9787 | 0.9759 | 0.9788 |
| Breastw | 0.9574 | 0.9574 | 0.9579 | 0.8957 | 0.9818 | 0.998 | 0.9721 | 0.9665 | 0.9967 | 0.8413 | 0.9968 |
| Campaign | 0.7176 | 0.7637 | 0.7684 | 0.7114 | 0.7874 | 0.7336 | 0.6750 | 0.9199 | 0.7793 | 0.7247 | 0.7714 |
| Cardio | 0.9135 | 0.9263 | 0.9263 | 0.8112 | 0.9712 | 0.9659 | 0.8734 | 0.9404 | 0.9561 | 0.8676 | 0.9562 |
| Cardiotocography | 0.8 | 0.8001 | 0.8005 | 0.7529 | 0.6859 | 0.8167 | 0.6358 | 0.8225 | 0.82 | 0.8195 | 0.8400 |
| Census | 0.6084 | 0.6897 | 0.7691 | 0.5964 | 0.837 | 0.5856 | 0.6709 | 0.7615 | 0.7509 | 0.6210 | 0.7714 |
| Comm.and.crime | 0.8047 | 0.8055 | 0.8075 | 0.6332 | 0.5696 | 0.8028 | 0.8330 | 0.7512 | 0.8192 | 0.7436 | 0.8461 |
| Cover | 0.9479 | 0.9501 | 0.9502 | 0.8694 | 0.1528 | 0.9908 | 0.6772 | 0.6223 | 0.9604 | 0.9627 | 0.9804 |
| Fault | 0.5459 | 0.5505 | 0.5728 | 0.4811 | 0.5475 | 0.6273 | 0.5314 | 0.6017 | 0.6239 | 0.6079 | 0.6279 |
| Fraud | 0.8534 | 0.8545 | 0.8996 | 0.7343 | 0.9403 | 0.944 | 0.9031 | 0.9258 | 0.9102 | 0.9149 | 0.9314 |
| Glass | 0.5382 | 0.5425 | 0.5921 | 0.5192 | 0.5858 | 0.5836 | 0.7161 | 0.6846 | 0.6269 | 0.5790 | 0.6969 |
| Hepatitis | 0.6885 | 0.693 | 0.7881 | 0.6919 | 0.81 | 0.7059 | 0.5361 | 0.5353 | 0.7466 | 0.7276 | 0.8122 |
| Imgseg | 0.8516 | 0.8718 | 0.8722 | 0.7681 | 0.5608 | 0.8751 | 0.7869 | 0.8323 | 0.889 | 0.8636 | 0.9093 |
| Ionosphere | 0.9306 | 0.932 | 0.9353 | 0.8685 | 0.8838 | 0.9666 | 0.9506 | 0.9743 | 0.9274 | 0.8322 | 0.9840 |
| Lympho | 0.9483 | 0.9506 | 0.9577 | 0.893 | 1 | 0.993 | 0.9176 | 0.9470 | 1 | 0.9507 | 1.0000 |
| Mammography | 0.8124 | 0.8605 | 0.8754 | 0.6967 | 0.8574 | 0.8999 | 0.8942 | 0.9034 | 0.8621 | 0.7484 | 0.9455 |
| Mnist | 0.9295 | 0.9321 | 0.9321 | 0.8219 | 0.2673 | 0.875 | 0.6193 | 0.9161 | 0.9312 | 0.9164 | 0.9712 |
| Musk | 0.9546 | 0.9573 | 0.96 | 0.8999 | 1 | 1 | 0.8621 | 0.9818 | 1 | 0.6706 | 1.0000 |
| Optdigits | 0.9254 | 0.9302 | 0.9305 | 0.7737 | 0.4667 | 0.9487 | 0.7806 | 0.9948 | 0.9653 | 0.8000 | 0.9972 |
| Parkinson | 0.6247 | 0.6315 | 0.6318 | 0.5715 | 0.6976 | 0.6658 | 0.3728 | 0.4927 | 0.6532 | 0.7035 | 0.6732 |
| Pendigits | 0.9372 | 0.9392 | 0.9534 | 0.8031 | 0.9449 | 0.8833 | 0.9430 | 0.9694 | 0.9903 | 0.9577 | 0.9979 |
| Pima | 0.725 | 0.7251 | 0.7253 | 0.6204 | 0.6857 | 0.7243 | 0.6671 | 0.7506 | 0.7451 | 0.6231 | 0.7651 |
| Satellite | 0.7575 | 0.7653 | 0.7858 | 0.7064 | 0.6594 | 0.824 | 0.7486 | 0.8099 | 0.8173 | 0.7569 | 0.8320 |
| Satimage-2 | 0.9038 | 0.9573 | 0.9573 | 0.8992 | 0.9789 | 0.9953 | 0.9629 | 0.9865 | 0.9955 | 0.9247 | 0.9968 |
| Shuttle | 0.9588 | 0.959 | 0.9594 | 0.8992 | 0.9919 | 0.9965 | 0.9951 | 0.9846 | 0.9993 | 0.9885 | 0.9992 |
| SpamBase | 0.7967 | 0.8035 | 0.8039 | 0.7543 | 0.8148 | 0.8066 | 0.7621 | 0.8049 | 0.8118 | 0.8071 | 0.8289 |
| Speech | 0.5775 | 0.5804 | 0.5848 | 0.3215 | 0.4268 | 0.5867 | 0.4622 | 0.4830 | 0.5631 | 0.4541 | 0.5821 |
| Thyroid | 0.942 | 0.9425 | 0.9477 | 0.862 | 0.8319 | 0.9876 | 0.9522 | 0.9831 | 0.9832 | 0.9288 | 0.9911 |
| Vertebral | 0.4616 | 0.4708 | 0.4781 | 0.1381 | 0.4032 | 0.3657 | 0.2511 | 0.4726 | 0.5254 | 0.5540 | 0.6216 |
| Vowels | 0.8097 | 0.8131 | 0.8355 | 0.7871 | 0.6506 | 0.934 | 0.6313 | 0.7652 | 0.7504 | 0.7562 | 0.8511 |
| WDBC | 0.96 | 0.96 | 0.96 | 0.9 | 0.9994 | 0.9994 | 0.9380 | 0.9466 | 1 | 0.8832 | 1.0000 |
| WPBC | 0.4956 | 0.4998 | 0.5023 | 0.4235 | 0.5451 | 0.5599 | 0.5022 | 0.5167 | 0.5034 | 0.5930 | 0.6198 |
| Wbc | 0.9504 | 0.9528 | 0.9531 | 0.8606 | 0.9713 | 0.9511 | 0.7803 | 0.9821 | 0.9763 | 0.8174 | 0.9963 |
| Wilt | 0.8794 | 0.8861 | 0.8866 | 0.6467 | 0.8752 | 0.9357 | 0.4442 | 0.3820 | 0.8736 | 0.8827 | 0.9357 |
| Wine | 0.96 | 0.96 | 0.96 | 0.8983 | 1 | 1 | 0.9041 | 0.9597 | 1 | 0.8717 | 1.0000 |
| Yeast | 0.4109 | 0.4163 | 0.4206 | 0.3655 | 0.4712 | 0.5084 | 0.4451 | 0.3294 | 0.4706 | 0.5379 | 0.5120 |
| Average | 0.7882 | 0.7999 | 0.8123 | 0.7066 | 0.7240 | 0.8205 | 0.7273 | 0.7837 | 0.8286 | 0.7720 | 0.8574 |

Table 13: Full analysis on the effects of different distance metrics in DRL in terms of AUC-PR. This table serves as an extension of Table 3.

| | Decomposition w/ Cosine distance | Separation w/ L1 distance | Alignment w/ L1 distance | Separation w/ L2 distance | Alignment w/ L2 distance | DRL (ours) |
|---|---|---|---|---|---|---|
| Abalone | 0.8350 | 0.8792 | 0.8535 | 0.876 | 0.8631 | 0.885 |
| Amazon | 0.0984 | 0.1091 | 0.1094 | 0.1006 | 0.1028 | 0.1206 |
| Annthyroid | 0.6824 | 0.6551 | 0.6484 | 0.6494 | 0.6497 | 0.6761 |
| Arrhythmia | 0.6165 | 0.5900 | 0.5852 | 0.5697 | 0.5924 | 0.627 |
| Backdoor | 0.8808 | 0.8784 | 0.8868 | 0.8786 | 0.8886 | 0.8915 |
| Breastw | 0.8851 | 0.9961 | 0.9957 | 0.9958 | 0.9943 | 0.9966 |
| Campaign | 0.4972 | 0.4344 | 0.4678 | 0.4609 | 0.4811 | 0.5013 |
| Cardio | 0.8253 | 0.8158 | 0.7921 | 0.8123 | 0.8209 | 0.8325 |
| Cardiotocography | 0.7318 | 0.7430 | 0.7458 | 0.7316 | 0.7415 | 0.754 |
| Census | 0.2109 | 0.2358 | 0.2421 | 0.2204 | 0.2333 | 0.2649 |
| Comm.and.crime | 0.8903 | 0.8538 | 0.9079 | 0.8692 | 0.903 | 0.9164 |
| Cover | 0.7663 | 0.7134 | 0.7793 | 0.7423 | 0.7613 | 0.783 |
| Fault | 0.6534 | 0.6391 | 0.6433 | 0.6444 | 0.6576 | 0.6649 |
| Fraud | 0.6097 | 0.6739 | 0.6551 | 0.6297 | 0.6444 | 0.6614 |
| Glass | 0.2792 | 0.1134 | 0.1187 | 0.1449 | 0.097 | 0.167 |
| Hepatitis | 0.6486 | 0.5816 | 0.6453 | 0.6179 | 0.6574 | 0.6627 |
| Imgseg | 0.9036 | 0.9185 | 0.9004 | 0.8998 | 0.9125 | 0.9238 |
| Ionosphere | 0.9526 | 0.9092 | 0.9549 | 0.9539 | 0.969 | 0.9895 |
| Lympho | 0.9868 | 0.8391 | 0.9762 | 0.8901 | 1 | 1 |
| Mammography | 0.8145 | 0.7645 | 0.8211 | 0.7972 | 0.8041 | 0.8406 |
| Mnist | 0.8579 | 0.8893 | 0.7912 | 0.8243 | 0.835 | 0.887 |
| Musk | 0.9705 | 0.8125 | 0.8484 | 0.9335 | 0.951 | 1 |
| Optdigits | 0.9035 | 0.9358 | 0.9350 | 0.9493 | 0.947 | 0.9356 |
| Parkinson | 0.9175 | 0.9442 | 0.9237 | 0.9476 | 0.9314 | 0.921 |
| Pendigits | 0.9029 | 0.8820 | 0.9218 | 0.8735 | 0.909 | 0.936 |
| Pima | 0.6978 | 0.7398 | 0.7440 | 0.7486 | 0.7584 | 0.7449 |
| Satellite | 0.8426 | 0.8364 | 0.8212 | 0.8499 | 0.8099 | 0.8692 |
| Satimage-2 | 0.9185 | 0.9174 | 0.9648 | 0.9345 | 0.9491 | 0.9703 |
| Shuttle | 0.9622 | 0.9318 | 0.9718 | 0.9674 | 0.9671 | 0.9819 |
| SpamBase | 0.8327 | 0.8629 | 0.8113 | 0.8852 | 0.8205 | 0.8413 |
| Speech | 0.0545 | 0.0310 | 0.0342 | 0.0398 | 0.0408 | 0.0584 |
| Thyroid | 0.8561 | 0.8113 | 0.8210 | 0.8209 | 0.8299 | 0.8626 |
| Vertebral | 0.2312 | 0.2681 | 0.2542 | 0.2686 | 0.2781 | 0.2854 |
| Vowels | 0.4307 | 0.4400 | 0.3696 | 0.4635 | 0.4425 | 0.4506 |
| WDBC | 0.9954 | 0.9928 | 1.0000 | 0.93 | 0.9319 | 1 |
| WPBC | 0.4896 | 0.4911 | 0.4843 | 0.4984 | 0.4943 | 0.5017 |
| Wbc | 0.9417 | 0.9589 | 0.9401 | 0.9655 | 0.959 | 0.9742 |
| Wilt | 0.4146 | 0.4384 | 0.4105 | 0.413 | 0.4238 | 0.4543 |
| Wine | 1.0000 | 1.0000 | 0.9714 | 1 | 1 | 1 |
| Yeast | 0.5308 | 0.5010 | 0.5049 | 0.5206 | 0.4817 | 0.5416 |
| Average | 0.7130 | 0.7007 | 0.7063 | 0.708 | 0.7134 | 0.7344 |

Table 14: Full analysis on the effects of different distance metrics in DRL in terms of AUC-ROC. This table serves as an extension of Table 3.

| | Decomposition w/ Cosine distance | Separation w/ L1 distance | Alignment w/ L1 distance | Separation w/ L2 distance | Alignment w/ L2 distance | DRL (ours) |
|---|---|---|---|---|---|---|
| Abalone | 0.7934 | 0.7744 | 0.7610 | 0.7691 | 0.7748 | 0.8086 |
| Amazon | 0.5525 | 0.5413 | 0.5649 | 0.5109 | 0.5258 | 0.5699 |
| Annthyroid | 0.8828 | 0.8342 | 0.8931 | 0.8672 | 0.9031 | 0.9239 |
| Arrhythmia | 0.7667 | 0.7309 | 0.7376 | 0.7296 | 0.742 | 0.7742 |
| Backdoor | 0.9607 | 0.9036 | 0.9248 | 0.9339 | 0.9372 | 0.9788 |
| Breastw | 0.9888 | 0.9932 | 0.9958 | 0.9961 | 0.9944 | 0.9968 |
| Campaign | 0.7646 | 0.7423 | 0.7188 | 0.7229 | 0.736 | 0.7714 |
| Cardio | 0.9327 | 0.9110 | 0.9586 | 0.9139 | 0.9236 | 0.9562 |
| Cardiotocography | 0.8134 | 0.7912 | 0.7276 | 0.7805 | 0.6999 | 0.84 |
| Census | 0.7659 | 0.6987 | 0.7083 | 0.7169 | 0.7354 | 0.7714 |
| Comm.and.crime | 0.8346 | 0.7802 | 0.8075 | 0.7976 | 0.8155 | 0.8461 |
| Cover | 0.9742 | 0.9417 | 0.9297 | 0.9394 | 0.952 | 0.9804 |
| Fault | 0.6495 | 0.5632 | 0.6057 | 0.5749 | 0.6028 | 0.6279 |
| Fraud | 0.9248 | 0.9243 | 0.9330 | 0.9362 | 0.9417 | 0.9314 |
| Glass | 0.6785 | 0.6729 | 0.6724 | 0.6887 | 0.6736 | 0.6969 |
| Hepatitis | 0.7936 | 0.7163 | 0.8032 | 0.733 | 0.8281 | 0.8122 |
| Imgseg | 0.9010 | 0.6974 | 0.7933 | 0.8724 | 0.882 | 0.9093 |
| Ionosphere | 0.9794 | 0.9685 | 0.9562 | 0.9501 | 0.9065 | 0.984 |
| Lympho | 0.9965 | 0.9507 | 0.9977 | 0.9005 | 1 | 1 |
| Mammography | 0.9473 | 0.8969 | 0.8956 | 0.8891 | 0.8753 | 0.9455 |
| Mnist | 0.9709 | 0.9686 | 0.9417 | 0.9352 | 0.9464 | 0.9712 |
| Musk | 0.9921 | 0.9006 | 0.9908 | 0.9295 | 0.9619 | 1 |
| Optdigits | 0.9631 | 0.8779 | 0.8706 | 0.8757 | 0.8908 | 0.9972 |
| Parkinson | 0.6688 | 0.7287 | 0.6738 | 0.7423 | 0.6984 | 0.6732 |
| Pendigits | 0.9712 | 0.8971 | 0.9321 | 0.9136 | 0.9275 | 0.9979 |
| Pima | 0.7607 | 0.7524 | 0.7505 | 0.7495 | 0.7773 | 0.7651 |
| Satellite | 0.8596 | 0.7916 | 0.7999 | 0.7952 | 0.8021 | 0.832 |
| Satimage-2 | 0.9335 | 0.8948 | 0.9696 | 0.9191 | 0.988 | 0.9968 |
| Shuttle | 0.9779 | 0.9860 | 0.9976 | 0.8803 | 0.9973 | 0.9992 |
| SpamBase | 0.8496 | 0.8717 | 0.7980 | 0.8696 | 0.818 | 0.8289 |
| Speech | 0.5123 | 0.4773 | 0.5173 | 0.5078 | 0.4901 | 0.5821 |
| Thyroid | 0.8935 | 0.9424 | 0.9856 | 0.9619 | 0.9879 | 0.9911 |
| Vertebral | 0.6276 | 0.5505 | 0.5848 | 0.5546 | 0.5857 | 0.6216 |
| Vowels | 0.8009 | 0.7824 | 0.8255 | 0.7909 | 0.7999 | 0.8511 |
| WDBC | 0.9749 | 0.9732 | 1.0000 | 0.9542 | 0.9955 | 1 |
| WPBC | 0.6096 | 0.5325 | 0.5702 | 0.5438 | 0.5796 | 0.6198 |
| Wbc | 0.9925 | 0.8801 | 0.9636 | 0.9483 | 0.9702 | 0.9963 |
| Wilt | 0.5033 | 0.7463 | 0.8637 | 0.8795 | 0.8407 | 0.9357 |
| Wine | 1.0000 | 1.0000 | 0.9933 | 1 | 1 | 1 |
| Yeast | 0.5078 | 0.4622 | 0.4857 | 0.4602 | 0.4668 | 0.512 |
| Average | 0.8318 | 0.8012 | 0.8225 | 0.8109 | 0.8243 | 0.8574 |

## A.11 THEORETICAL ANALYSIS

**Proposition 1** *Let $\mathbf{w}_n \in \mathcal{N}$ and $\mathbf{w}_a \in \mathcal{A}$ denote the computed weights of normal and anomalous samples, where $\mathcal{N}$ and $\mathcal{A}$ denote the weight sets of normal and anomalous samples, respectively. Given $\mathbf{w} \in \sum^K$, where $\sum^K$ represents the probability simplex in $\mathbb{R}^K$, the expected discrepancy between normal and anomalous weights, $\mathbb{E}_{\mathbf{w}_n \in \mathcal{N}, \mathbf{w}_a \in \mathcal{A}} \left[ \|\mathbf{w}_n - \mathbf{w}_a\|_2^2 \right]$, can be amplified by increasing the variance of $\|\mathbf{w}_n\|_2$ among $\mathcal{N}$.*

*Proof.* The expected discrepancy between normal and anomalous weights is formulated as:

$$
\begin{aligned}
\mathbb{E}\left[\|\mathbf{w}_n - \mathbf{w}_a\|_2^2\right] =& \mathbb{E}\left[(\mathbf{w}_n - \mathbf{w}_a)^{\mathrm{T}}(\mathbf{w}_n - \mathbf{w}_a)\right] = \mathbb{E}\left[\|\mathbf{w}_n\|_2^2\right] + \mathbb{E}\left[\|\mathbf{w}_a\|_2^2\right] - 2\mathbb{E}\left[\mathbf{w}_n^{\mathrm{T}}\mathbf{w}_a\right] \\
=& \mathrm{Var}(\|\mathbf{w}_n\|_2) + \mathbb{E}^2\left[\|\mathbf{w}_n\|_2\right] + \mathbb{E}\left[\|\mathbf{w}_a\|_2^2\right] - 2\mathbb{E}\left[\mathbf{w}_n^{\mathrm{T}}\mathbf{w}_a\right] \\
\geq& \mathrm{Var}(\|\mathbf{w}_n\|_2) + \mathbb{E}^2\left[\frac{1}{\sqrt{K}}\right] + \mathbb{E}\left[(\frac{1}{\sqrt{K}})^2\right] - 2\mathbb{E}\left[\|\mathbf{w}_n\|_2\|\mathbf{w}_a\|_2\right] \qquad (7) \\
\geq& \mathrm{Var}(\|\mathbf{w}_n\|_2) + \frac{2}{K} - 2\mathbb{E}\left[1*1\right] = \mathrm{Var}(\|\mathbf{w}_n\|_2) + \frac{2}{K} - 2,
\end{aligned}
$$

where $\mathrm{Var}(\cdot)$ denotes the variance, $\mathbf{w}$ is a probability simplex as discussed above and satisfies $\sum_{k=1}^{K} w^k = 1$ and $0 \leq w^k \leq 1$, $\|\mathbf{w}\|_2 = \sqrt{\sum_{k=1}^{K}(w^k)^2} \geq \frac{\sum_{k=1}^{K} w^k}{\sqrt{K}} = \frac{1}{\sqrt{K}}$ is given by Arithmetic Mean-Geometric Mean Inequality (Bhatia & Kittaneh, 2000), $\mathbb{E}\left[\mathbf{w}_n^{\mathrm{T}}\mathbf{w}_a\right] \leq \mathbb{E}\left[|\mathbf{w}_n^{\mathrm{T}}\mathbf{w}_a|\right] \leq \mathbb{E}\left[\|\mathbf{w}_n\|_2\|\mathbf{w}_a\|_2\right]$ is given by Cauchy–Schwarz inequality (Bhatia & Davis, 1995), and $\|\mathbf{w}\|_2 \leq 1$ arises from the condition about the simplex. The lower bound of $\mathbb{E}\left[\|\mathbf{w}_n - \mathbf{w}_a\|_2^2\right]$ is given by $\mathrm{Var}(\|\mathbf{w}_n\|_2) + \frac{2}{K} - 2$. To increase this lower bound, we can enhance $\mathrm{Var}(\|\mathbf{w}_n\|_2)$.

**Proposition 2** *Let $\mathbf{w}_n \in \mathcal{N}$ and $\mathbf{w}_a \in \mathcal{A}$ denote the computed weights of normal and anomalous samples, where $\mathcal{N}$ and $\mathcal{A}$ denote the weight sets of normal and anomalous samples, respectively. Given $\mathbf{w} \in \sum^K$, where $\sum^K$ represents the probability simplex in $\mathbb{R}^K$, denoting $\mu_{\mathbf{w}_n} = \mathbb{E}_{\mathbf{w}_n \in \mathcal{N}}\left[\mathbf{w}_n\right]$ as the centroid of the normal samples' weights, $\mathbb{E}_{\mathbf{w}_n \in \mathcal{N}, \mathbf{w}_a \in \mathcal{A}}\left[\|\mathbf{w}_a - \mu_{\mathbf{w}_n}\|_2^2\right] - \mathbb{E}_{\mathbf{w}_n \in \mathcal{N}}\left[\|\mathbf{w}_n - \mu_{\mathbf{w}_n}\|_2^2\right]$, can be amplified by increasing the variance of $\|\mathbf{w}_n\|_2$ among $\mathcal{N}$.*

*Proof.* $\mathbb{E}_{\mathbf{w}_n \in \mathcal{N}, \mathbf{w}_a \in \mathcal{A}}\left[\|\mathbf{w}_a - \mu_{\mathbf{w}_n}\|_2^2\right] - \mathbb{E}_{\mathbf{w}_n \in \mathcal{N}}\left[\|\mathbf{w}_n - \mu_{\mathbf{w}_n}\|_2^2\right]$ is formulated as:

$$
\begin{aligned}
& \mathbb{E}\left[\|\mathbf{w}_a - \mu_{\mathbf{w}_n}\|_2^2\right] - \mathbb{E}\left[\|\mathbf{w}_n - \mu_{\mathbf{w}_n}\|_2^2\right] \\
=& \mathbb{E}\left[\|\mathbf{w}_a - \mu_{\mathbf{w}_n}\|_2^2 - \|\mathbf{w}_n - \mu_{\mathbf{w}_n}\|_2^2\right] \\
=& \mathbb{E}\left[(\mathbf{w}_a - \mu_{\mathbf{w}_n})^{\mathrm{T}}(\mathbf{w}_a - \mu_{\mathbf{w}_n}) - (\mathbf{w}_n - \mu_{\mathbf{w}_n})^{\mathrm{T}}(\mathbf{w}_n - \mu_{\mathbf{w}_n})\right] \\
=& \mathbb{E}\left[\mathbf{w}_a^{\mathrm{T}}\mathbf{w}_a - 2\mathbf{w}_a^{\mathrm{T}}\mu_{\mathbf{w}_n} + \mu_{\mathbf{w}_n}^{\mathrm{T}}\mu_{\mathbf{w}_n} - (\mathbf{w}_n^{\mathrm{T}}\mathbf{w}_n - 2\mathbf{w}_n^{\mathrm{T}}\mu_{\mathbf{w}_n} + \mu_{\mathbf{w}_n}^{\mathrm{T}}\mu_{\mathbf{w}_n})\right] \\
=& \mathbb{E}\left[\mathbf{w}_a^{\mathrm{T}}\mathbf{w}_a + \mathbf{w}_n^{\mathrm{T}}\mathbf{w}_n - 2\mathbf{w}_n^{\mathrm{T}}\mathbf{w}_n + 2\mathbf{w}_n^{\mathrm{T}}\mu_{\mathbf{w}_n} - 2\mathbf{w}_a^{\mathrm{T}}\mu_{\mathbf{w}_n}\right] \\
=& \mathbb{E}\left[\|\mathbf{w}_a\|_2^2\right] + \mathbb{E}\left[\|\mathbf{w}_n\|_2^2\right] - 2\mathbb{E}\left[\|\mathbf{w}_n\|_2^2\right] + 2\mathbb{E}\left[\mathbf{w}_n^{\mathrm{T}}\mu_{\mathbf{w}_n} - \mathbf{w}_a^{\mathrm{T}}\mu_{\mathbf{w}_n}\right] \\
=& \mathbb{E}\left[\|\mathbf{w}_a\|_2^2\right] + \mathrm{Var}(\|\mathbf{w}_n\|_2) + \mathbb{E}^2\left[\|\mathbf{w}_n\|_2\right] - 2\mathbb{E}\left[\|\mathbf{w}_n\|_2^2\right] + 2\mathbb{E}\left[\mathbf{w}_n^{\mathrm{T}}\mu_{\mathbf{w}_n}\right] - 2\mathbb{E}\left[\mathbf{w}_a^{\mathrm{T}}\mu_{\mathbf{w}_n}\right] \\
=& \mathrm{Var}(\|\mathbf{w}_n\|_2) + \mathbb{E}\left[\|\mathbf{w}_a\|_2^2\right] + \mathbb{E}^2\left[\|\mathbf{w}_n\|_2\right] - 2\mathbb{E}\left[\|\mathbf{w}_n\|_2^2\right] + 2\mathbb{E}\left[\mathbf{w}_n^{\mathrm{T}}\mu_{\mathbf{w}_n}\right] - 2\mathbb{E}\left[\mathbf{w}_a^{\mathrm{T}}\mu_{\mathbf{w}_n}\right] \qquad (8) \\
\geq& \mathrm{Var}(\|\mathbf{w}_n\|_2) + \mathbb{E}\left[(\frac{1}{\sqrt{K}})^2\right] + \mathbb{E}^2\left[\frac{1}{\sqrt{K}}\right] - 2\mathbb{E}\left[\|\mathbf{w}_n\|_2\|\mathbf{w}_n\|_2\right] - 2\mathbb{E}\left[\|\mathbf{w}_a\|_2\|\mu_{\mathbf{w}_n}\|_2\right] \\
\geq& \mathrm{Var}(\|\mathbf{w}_n\|_2) + \mathbb{E}\left[(\frac{1}{\sqrt{K}})^2\right] + \mathbb{E}^2\left[\frac{1}{\sqrt{K}}\right] - 2\mathbb{E}\left[1*1\right] - 2\mathbb{E}\left[1*1\right] \\
=& \mathrm{Var}(\|\mathbf{w}_n\|_2) + \frac{2}{K} - 4,
\end{aligned}
$$

where $\mathrm{Var}(\cdot)$ denotes the variance, $\|\mathbf{w}\|_2 = \sqrt{\sum_{k=1}^{K}(w^k)^2} \geq \frac{\sum_{k=1}^{K} w^k}{\sqrt{K}} = \frac{1}{\sqrt{K}}$ is given by Arithmetic Mean-Geometric Mean Inequality (Bhatia & Kittaneh, 2000), $\mathbb{E}\left[\mathbf{w}_a^{\mathrm{T}}\mu_{\mathbf{w}_n}\right] \leq \mathbb{E}\left[|\mathbf{w}_a^{\mathrm{T}}\mu_{\mathbf{w}_n}|\right] \leq \mathbb{E}\left[\|\mathbf{w}_a\|_2\|\mu_{\mathbf{w}_n}\|_2\right]$ is given by Cauchy–Schwarz Inequality (Bhatia & Davis, 1995), $\mu_{\mathbf{w}_n} \in \sum^K$ is given by $\mathbf{w}_n \in \sum^K$, and $\|\mathbf{w}\|_2 \leq 1$ arises from the conditions $\sum_{k=1}^{K} w^k = 1$ and $0 \leq w^k \leq 1$.

Therefore the lower bound of $\mathbb{E}_{\mathbf{w}_n \in \mathcal{N}, \mathbf{w}_a \in \mathcal{A}} \left[ \|\mathbf{w}_a - \mu_{\mathbf{w}_n}\|_2^2 \right] - \mathbb{E}_{\mathbf{w}_n \in \mathcal{N}} \left[ \|\mathbf{w}_n - \mu_{\mathbf{w}_n}\|_2^2 \right]$ is given by $\mathrm{Var}(\|\mathbf{w}_n\|_2) + \frac{2}{K} - 4$. To increase this lower bound, we can enhance $\mathrm{Var}(\|\mathbf{w}_n\|_2)$.

## A.12 LIMITATION

The primary limitation of the current DRL method is that it is designed specifically for tabular anomaly detection. Although our method can be used to different data types for the anomaly detection task by introducing representation decomposition, we may need to design the specific architecture for the weight learner and alignment learner due to the difference between data types. In the future, we plan to explore its application to other data types, where incorporating prior structural knowledge from these data types might be a possible solution.

