# OpenReview forum: "DRL: Decomposed Representation Learning for Tabular Anomaly Detection"
_ICLR.cc/2025/Conference — ICLR 2025 Poster_

### Official Review · Reviewer_TJDb · 2024-10-28

**Soundness:** 3
**Presentation:** 3
**Contribution:** 2
**Rating:** 6
**Confidence:** 5

**Summary:**

The authors present a Novel reconstruction-based outlier detection (R-OD) method for tabular data called DRL. DRL's novelty among other R-OD methods is that it focuses on representing the data $\mathbf{x} \in \mathbb{R}^d$ as a linear combination of a randomly selected basis $\mathcal{B}$ of $\mathbb{R}^D$, with $d>D$. To this end, the network learns an encoder $f_{\theta_f}$ from $\mathbb{R}^d$ to $\mathbb{R}^D$, a decoder $g_{\theta_g}$ and a weight learning function $\phi_{\theta_\phi}$ that is in charge of learning such weight representation. The authors also include a novel loss function to train DRL, focusing on 3 different aspects. (i) It learns an embedding function $f$ that agrees with a linear combination representation of the embedding $\phi$  in the $L_{decomposition}$ loss, (ii) it focuses on separating the normal samples between themselves in the $L_\text{separation}$ loss and (iii) it reconstructs the embedding by $f$ with $g$ in the $L_{aligment}$ loss. Furthermore, the authors include a theoretical result that motivates the use of $L_\text{separation}$, by proving that an increase in the total $L_2$ distances of the representations of inlier data during training, leads to a greater expected distance between an inlier and an outlier (Proposition 1).

Additionally, the authors include an extensive list of real-world experiments in the main text, with 40 real-world datasets and 15 relevant competitors. In particular, the authors tested DRL's One-class classification (OCC) performance, provided an ablation study, and studied different types of distances and base selection strategies. The appendix includes further experiments with different types of synthetic outliers, sensibility to the parameters, robustness study and computational cost analysis among others.

**Strengths:**

1. The paper has a nice flow in the presentation.
2. The idea of using a linear combination of a projected random basis to represent the data is an interesting idea with big explainability potential.
3. The paper is well motivated, as the problem of OCC in tabular data is an important task in ML.
4. There is a large list of different experiments.
5. The authors use a statistical comparison test to provide a statistical significance analysis of the main experimental results.
6. The authors include the pseudo-code of the method in the appendix.

**Weaknesses:**

There is not enough evidence to support the claims that the authors make in the paper, both theoretically and experimentally.
Particularly:

### Theory
T.1. The authors claim that DRL " (...) assumes that the representation of each normal sample in the latent space can be effectively modeled as a mixture of fixed basis vectors with specific mixture proportions" (L160-161). This assumption is central to the method's idea, and it goes unsupported both in theory (no theoretical example of data behaving as such) and practice (no example proving that given such a theoretical example, one could extract exactly such representation). As an example, consider the manifold learning (ManL) literature. Assuming that data comes from a lower dimensional manifold $\mathcal{M}$ is a big assumption, however, there are examples of such data being generated by synthetic means, and also examples of ManL methods properly learning the representations ---see figure 5 in [Meila & Zhang].

T.2. The authors further claim that Proposition 1 proves that the outliers are going to be far away from the inliers (i.e., separated). While technically true on average, Proposition 1 does not "(...) amplify the discrepancy between the two (inlier and outlier) patterns" L224. What proposition 1 shows is that, if one increases the variance of the learned weights $w_i$, the process of measuring the distance between an inlier's representation and an outlier's representation will be, on average, higher. This, however, does not imply that, necessarily, you will increase the distance of the outliers with the total set of inliers. For example, one could obtain a representation that places a large set of outliers in the centroid of the inliers, with the remaining inliers being sparse around the centroid.

T.3. DRL's performance is not explained in the theoretical derivations. The authors focus on proving how the representation that they learn can "separate" outlier from inlier, but it is not clear how an increase in distance can affect the final scoring function (they use $L_\text{decomposition}$ as a final score). Particularly, it is not clear to me how it can affect the encoder $f$.


### Experiments
While this paper contains a large collection of experiments, they do not focus on verifying the theoretical claims of the method.

E.1. The authors include experiments to try to prove the claim mentioned in weakness number T.2 in Figure 1. However, out of the total list of datasets (40) and competitors (15), they only use 2 datasets and 1 competitor without any clear reason. This does not contain sufficient evidence that the outliers and the inliers can be separated by Proposition 1.

E.2. Weakness T.1 addresses that the random basis reconstruction assumption has no example in theory. In practice, this assumption seems to not be properly explored. The authors include an ablation study in Table 2 where they compare different versions (including one employing only the basis reconstruction assumption) of DRL. However, they only included 7 out of the 40 datasets introduced earlier.

Additionally, the authors include in Figure 5 a comparison between different variants of DRL. Particularly, variant B "applies the decomposition loss to observations, assuming that each normal sample can be decomposed into a set of orthogonal basis vectors (...)" L472-473. This variant ranked third to last among 8 different versions of the method. Furthermore, it is not possible to compare this performance to the other detectors as the authors do not say which methods they use, and only report an average PR-AUC.

E.3. At the time of this review, there is no code available for DRL, making it not reproducible.

**Questions:**

I kindly ask the authors to address the following questions & concerns in order to improve the manuscript:

1. The performance of the method is not properly verified (see T.1,T.2,T.3). It will greatly improve the paper to address, particularly T.1 and T.2. For T.2, I suggest to prove that $\|w_a - \mu_{w_i} \|$ has a lower bound greater than the increase in $\|w_i\|$, where $\mu_{w_i}$ is the centroid of the inlier's representation in basis $\mathcal{B}$.

2. The authors should consider addressing concern E.1, to at least experimentally verify T.2. Is there any particular reason why not all datasets and competitors were considered (at the very least as an image in the appendix)?

3. The authors should consider rewriting section 5.2. It is unclear to me the difference between the ablation study presented in the section and the *Comparison of DRL and Variants in the Observation Space*. For instance, why are they separate sections if they seem to try to study the same thing? Why is there no reference to the datasets used in *Comparison of DRL and Variants in the Observation Space*, and why only the average PRAUC and not the full table of results? Why did the authors consider only 7 datasets out of the total 40 in the ablation study?

4. The author should strongly consider releasing the code for the method. Reproducibility is a crucial part of the scientific process, and without the code, this paper cannot be reproduced. Is there any strong reason as to why the code was not available at the time of the review?

I will consider changing my rating upon successfully addressing these questions during the rebuttal.



### Additional remarks that did not influence the score
A couple of additional remarks that might improve the manuscript:

R1. P-values are not a scalar metric. This means that a difference between .20 to .90 is the same as a difference between .0501 and .0502 (when selecting a critical value of 0.05) [Lavine]. Thus, authors should consider removing Figure 4 for something more meaningful (see [Campos et al] results section).

R2. The Wilcoxon signed-rank test is a single-population comparison test. This means that it is strictly designed to work in single comparisons. The setting presented in the experiments is a multiple comparison setting, thus a multiple comparison test should be used in this regard (see, for example, the Conover-Iman test [Conover]).


### References

[Melia & Zhang] Meilă, M., & Zhang, H. (2023). Manifold learning: what, how, and why. https://arxiv.org/abs/2311.03757

[Lavine] M. Lavine, “Introduction to Statistical Thought”

[Campos et al] Campos, G.O., Zimek, A., Sander, J. et al. On the evaluation of unsupervised outlier detection: measures, datasets, and an empirical study. Data Min Knowl Disc 30, 891–927 (2016). https://doi.org/10.1007/s10618-015-0444-8

[Conover] WJ Conover (1979), Multiple-Comparisons Procedures.



### Update

After the discussion with the authors, I have decided to increase my score from 3 to 6. See the comments below for more information.

---

> ### Author Response · Authors · 2024-11-24
> **Response (Part 1/3)**
>
> Q1: Authors need to solve T1, T2 and T3.
>
>
> R to Q1: Thanks for your insightful comment! We would like to address T1, T2, and T3 respectively.
>
> T1: The assumption that the representation of each normal sample in the latent space can be effectively modeled as a mixture of fixed basis vectors with specific mixture proportions needs to be verified.
>
> R to T1: Thanks for your valuable comment!
>
> (1) It is worth noting that for anomaly detection, the key insight is that a higher anomaly score indicates higher confidence in a sample being anomalous. Therefore, even if the decomposition does not perfectly capture the normal sample distribution, as long as it better represents normal samples than anomalous ones, the model can still achieve accurate anomaly detection. Below, we will explain that decomposition is sufficient to express the normal samples and distinguish anomaly from normal samples.
>
> (2) To accurately capture the statistical characteristics of normal samples and distinguish them from anomalies, we introduce a set of fixed and shared basis vectors to represent the global structure of the normal data, where the orthogonal basis vectors $\mathcal{B}$ are introduced to enforce the diversity and dependencies among them. Given basis vectors $\mathcal{B}$, each representation $\mathbf{h}$ in this space can be expressed as $\mathbf{w}\mathcal{B}$, where $\mathbf{w}$ is the weight vector (denoting the coordinates) and specific to each representation. We optimize the decomposition loss (Eq.3 in original paper) to expect that all normal samples' representations can reside in such subspace, where we show the convergence of the decomposition loss on the training set in Fig. 18 and 19 in Appendix 10 in revised version.
>
> (3) Besides, as shown in Fig. 12 in Appendix 10 of revised version, we provide the T-SNE visualization of representation $\mathbf{h}$ by feature extractor $f$ and the reconstructed representation $\tilde{\mathbf{h}}$ by linear combination of basis vectors $\mathbf{w}\mathcal{B}$ over all datasets. We can observe a notable overlap between $\mathbf{h}$ and $\tilde{\mathbf{h}}$ of normal samples, and a significant distinction between $\mathbf{h}$ and $\tilde{\mathbf{h}}$ of anomalous samples. This demonstrates that compared to anomalous samples, normal sample representations can be better modeled as a mixture of fixed basis vectors, making it reasonable to distinguish anomaly from normal samples with the decomposition loss (i.e., anomaly score).
>
> (4) To further verify the failure of modeling anomalous representations as a mixture of fixed basis vectors, we provide experiment results in Fig. 17 of Appendix 10 in revised version. We can find that the anomaly score of anomalous samples is significantly larger than that of normal samples, especially when we introduce the separation loss (minimizing the similarity between weight vectors of normal samples).

---

> > ### Author Response · Authors · 2024-11-24
> > **Response (Part 3/3)**
> >
> > Q2: The authors should consider addressing concern E.1, to at least experimentally verify T.2 by extend the experiment results of Figure 1 in original paper.
> >
> > R to Q2: Thanks for your valuable suggestion! Following your suggestion, we extend the experiment results of Figure 1, and provide the extensive visualizations in  Fig.14 and Fig.15 of Appendix 10 in revised version.
> > We can observe that ignoring observation entanglement in tabular anomaly detection under the one-class classification setting can lead to diminished discriminative power between learned normal and anomalous patterns, as the overlap between normal and anomalous representations within the latent space of deep models may obscure the distinction between them.
> > Our proposed method can amplify the discrepancy between the two (inlier and outlier) patterns.
> >
> >
> >
> > Q3: Unclear description of Section 5.2.
> >
> > R to Q3: Thanks for your insightful suggestion! We apologize for any confusion and we have made improvements to the clarity of Section 5.2 in the revised version for better illustration.
> > Following your suggestion, we merge the ablation study and the comparison of DRL and variants in the observation space into the same section.
> > Due to the limited space, we summarize the results over 40 datasets in Table 2 in the revised version. We also provide the full results with the reference of data in Table 11 to Table 14 in Appendix 10 of revised version.
> > We also make the description of variant B (currently named variant E) clear in Section 5.2 as mentioned in E.2.
> >
> >
> > Q4: Code needs to be released.
> >
> > R to Q4:
> > Thanks for your valuable feedback!
> > Due to time constraints at the time of submission, the code was not available. However, we have now released the source code to facilitate reproducibility. We hope this addresses your concern.
> >
> >
> > Additional remarks:
> >
> > R1: P-values are not a scalar metric.
> >
> > R to R1:
> > Thanks for your valuable suggestion! We revised the Fig.4 in the revised paper.
> >
> > R2: Considering using a multiple comparison test.
> >
> > R to R2:
> > Thanks for your insightful suggestion! We would use a multiple comparison, for example, the Conover-Iman test, in the final revision.

---

> > ### Author Response · Authors · 2024-11-26
> > **A Gentle Reminder**
> >
> > Dear Reviewer TJDb,
> >
> > Thank you very much again for your time and efforts in reviewing our paper.
> >
> > We kindly remind that our discussion period is closing soon.
> >
> > We just wonder whether there is any further concern and hope to have a chance to respond before the discussion phase ends.
> >
> > Many thanks, Authors

---

> > > ### Comment · Reviewer_TJDb · 2024-11-28
> > > **Reviewer's Response**
> > >
> > > I want to thanks the authors for taking the time to answer all my questions.  I will structure my review as follows. I will first respond to all of your answers. Then, I will summarize my re-review. Lastly, I will give my revised and final score of the manuscript.
> > >
> > > ## Questions
> > > **Q1**:
> > >
> > > - _*T1*_:
> > >   - (1) That it will fail with anomalous samples is not a guarantee. That is why it is important to verify when this will happen.
> > >   - (3) Figure 12 does, in fact, exemplify a counterargument I made in my original Review. Particularly at the end of weakness T.2 --- see **breastw** and **ionosphere**, for example.  My particular worry is that pure distance in the embedding space does not have to translate to actual performance, as the proposed score is a reconstruction error and not a hyperplane in the embedding space.
> > >   - (4) It does show examples of it in cases where it works, but as it does not contain all of the datasets I cannot judge it.
> > >
> > > - _*T2*_: I want to thank the authors for their efforts. This is, indeed, what I asked for.
> > >
> > > - _*T3*_:  I still disagree regarding the reconstruction as a score. As I already mentioned, an increase in absolute distance does not have to carry over as an increase in reconstruction error. Additionally, the experiment in Fig. 17 are insufficient to say that, empirically, it works.
> > >
> > >
> > > **Q2**: I do agree that Figure 14 and 15 demonstrate a good separation power. However, i cannot be used as a measure of performance. For instance, the images are T-SNE representations of the embedding space, thus an overlap in the image does not have to mean that it will overlap in the actual embedding of each method. I do agree, however, with the last claim, and believe that the authors successfully proved that in the rebuttal.
> > >
> > > **Q3**: Thank you for taking care of this. I found Table 2 especially helpful.
> > >
> > > ## Summary
> > >
> > > My review raised multiple issues with the manuscript. Specifically, about (1) the theoretical guarantees of separating anomalies from inliers, (2) the increase in performance that this will take, and (3) that normal samples are better represented with the randomized basis in the embedding space than anomalies.
> > >
> > > *I believe that the authors successfully proved, both theoretically and experimentally (1). However, (2) was left unproven theoretically and addressed partially empirically. (3) still seems to be more of a hypothesis made about the data rather than an actual fact. However, based on the performance of the method in practice, I believe that there are some grounds for this being a general enough occurrence to be used in practice.*
> > >
> > > Additionally, the authors gave the code for the experiments after asking for it. However, this was past the deadline for the appendix (which this counts as), and the authors specifically stated that they did not share it because of missing the deadline.
> > >
> > > ## Final score
> > >
> > > I will increase my score in soundness significantly, as the authors partially addressed the separation question. While the authors did not provide a theoretical proof as to why the method performed well in general, I believe there is an understanding as to why it does perform well when anomalies cannot be represented with the same base as the inliers. While it is not enough for a general outlier detection method, due to its overall performance, and its good separation power of its representation, it has merit as an embedding technique for other outlier detection methods to utilize. That is why I chose to increase my final score of the manuscript.

---

> > > > ### Author Response · Authors · 2024-11-29
> > > > **Response**
> > > >
> > > > Thank you for taking the time to review our response and for your positive feedback! We would like to further explain the remaining issues in T1 and T3.
> > > >
> > > > To improve anomaly detection performance, we actually aim to learn less entangled latent representations without seeing anomaly samples. The way we achieve this includes two aspects. Firstly, we enforce the latent representations of normal observations to a constrained latent space defined by a set of orthogonal and fixed basis vectors, where each representation is decomposed into a weighted combination of the basis vectors (the weight vector is on a probability simplex). Besides, we enforce normal representations to span this constrained subspace by introducing the separation loss (minimizing the similarity between weight vectors of normal samples). Intuitively, the representation of an anomaly sample will be 'squeezed out' from the subspace. That is why our learned encoder can distinguish anomaly samples from normal samples better with separation loss. We will consider providing a theoretical proof to further support this intuition.
> > > >
> > > > We now extend the empirical evidence in Fig.17 in revised version to illustrate the impact of the separation loss on performance, which is available at https://anonymous.4open.science/r/DRL1-A5BB. We can find that the anomaly score (calculated by decomposition loss) of anomalous samples is larger than that of normal samples, especially when we introduce the separation loss (minimizing the similarity between weight vectors of normal samples). We will make it more clearer in our camera ready.

---

> ### Author Response · Authors · 2024-11-24
> **Response (Part 2/3)**
>
> T2: The authors need to further prove that the distance of the outliers from the total set of inliers could be increased with separation loss.
>
> R to T2: Thanks for your insightful suggestion! Let $\mathbf{w}_n$ and $\mathbf{w}_a$ denote the computed weights of normal and anomalous samples respectively. To empirically prove that $|\mathbf{w}_a  - \mu _ {\mathbf{w}_n}|$ has a lower bound greater than the increase in the variance of $|\mathbf{w}_n|$ when introducing the separation loss, we calculate the $\text{E}\left[\|\mathbf{w}_a  - \mu _ {\mathbf{w}_n}\|_2^2\right]$ and $\text{E}\left[\|\mathbf{w}_n  - \mu _ {\mathbf{w}_n}\|_2^2\right]$, where we consider two variants, without separation loss and with separation loss. We performed the statistics on all of the datasets and add the results in Table 10 in Appendix 10 of our revised version. Due to the limited space, we provide partial experiment results as follows. We can find that, introducing the separation loss can indeed increase the variance of normal samples. However, the $\text{E}\left[\|\mathbf{w}_a  - \mu _ {\mathbf{w}_n}\|_2^2\right]$ has a more greater growth.
> Additionally, we also theoretically prove that $\text{E}\left[\|\mathbf{w}_a - \mu _ {\mathbf{w}_n}\|_2^2\right] - \text{E}\left[\|\mathbf{w}_n - \mu _ {\mathbf{w}_n}\|_2^2\right]$, can be amplified by increasing the variance of $\|\mathbf{w}_n\|_2$, as illustrated in Proposition 2 in Appendix 10 of revised version.
>
>
> |                                 |                                                                   | abalone | amazon | annthyroid | arrhythmia | backdoor | breastw | campaign | cardio | Cardiotocography | census  | Average of 40 data |
> |---------------------------------|-------------------------------------------------------------------|---------|--------|------------|------------|----------|---------|----------|--------|------------------|---------|--------------------|
> |  w/o separation                 | $\text{E}\left[\|\mathbf{w}_a  - \mu _ {\mathbf{w}_n}\|_2^2\right]$ | 0.2550  | 0.1126 | 0.1717     | 0.5577     | 0.1916   | 0.1738  | 0.0057   | 0.3347 | 0.2055           | 0.0270  | 0.2228             |
> |  w/o separation                 | $\text{E}\left[\|\mathbf{w}_n  - \mu _ {\mathbf{w}_n}\|_2^2\right]$ | 0.1411  | 0.1021 | 0.0744     | 0.3935     | 0.0995   | 0.0307  | 0.0041   | 0.1836 | 0.1439           | 0.0274  | 0.1274             |
> |  w/o separation                 | Gap                                                               | 0.1139  | 0.0105 | 0.0973     | 0.1642     | 0.0921   | 0.1431  | 0.0016   | 0.1511 | 0.0616           | -0.0004 | 0.0954             |
> |          w/ separation          | $\text{E}\left[\|\mathbf{w}_a  - \mu _ {\mathbf{w}_n}\|_2^2\right]$ | 0.5647  | 0.3276 | 0.3049     | 0.6160     | 0.2893   | 0.2080  | 0.8078   | 0.3935 | 0.2411           | 0.8362  | 0.3791             |
> |          w/ separation          | $\text{E}\left[\|\mathbf{w}_n  - \mu _ {\mathbf{w}_n}\|_2^2\right]$ | 0.2316  | 0.1928 | 0.1859     | 0.4333     | 0.1678   | 0.0480  | 0.1964   | 0.2077 | 0.1739           | 0.1206  | 0.1771             |
> |          w/ separation          | Gap                                                               | 0.3331  | 0.1348 | 0.119      | 0.1827     | 0.1215   | 0.16    | 0.6114   | 0.1858 | 0.0672           | 0.7156  | 0.2020             |
>
>
> T3: How separation loss can affect the encoder $f$ and how an increase in distance can affect the final scoring function.
>
> R to T3: Thanks for your valuable comment! We are sorry for any confusion. Based on the responses to T1 and T2, we have proved that the learned representations can "separate" outliers from inliers. We further visualize the T-SNE of the learned representations of encoder w/o and w/ separation loss over all datasets in Fig.13 in Appendix 10 of revised version to verify the representation separation. We can observe that with separation loss, the discriminative distinction between normal and anomalous patterns within the latent space can be enhanced. This constraint can facilitate the capturing of shared information within normal patterns by encoder $f$.
>
> For the concern about how an increase in distance can affect the final scoring function, please refer to the part (3) and (4) of response to T1.
> In addition, the ablation results provided in Table 2 of original paper also verifies that the separation loss is crucial to the model performance.

---

### Official Review · Reviewer_Sgt1 · 2024-11-03

**Soundness:** 2
**Presentation:** 2
**Contribution:** 3
**Rating:** 6
**Confidence:** 4

**Summary:**

This paper introduces Decomposed Representation Learning (DRL), a new framework for anomaly detection in tabular data. This approach aims to overcome the limitations of traditional reconstruction-based methods, which often struggle with data entanglement, especially in tabular settings where feature heterogeneity can obscure the separation between normal and anomalous samples. DRL remaps data into a latent space, enforcing each normal sample's representation as a linear combination of orthogonal basis vectors that are both data-free and training-free. Furthermore, DRL introduces a constraint to amplify discrepancies between normal and anomalous patterns in the latent space.

**Strengths:**

1. This paper is well-motivated, and the technical details are illustrated properly.
2. The proposed method is supported by a theoretical analysis of the constraint to maximize the discrepancy between normal and anomalous samples.
3. The authors conducted comprehensive experiments to demonstrate the effectiveness of the proposed DRL.

**Weaknesses:**

1. DRL relies on decomposing the representation of normal samples into linear combinations of orthogonal basis vectors, and this decomposition assumes that normal samples are well described by fixed orthogonal bases in the potential space. However, the true distribution of normal samples in complex, high-dimensional feature spaces may not be captured simply by a small number of basis vectors. The reviewer is concerned about whether the decomposition may not be sufficient to completely express the normal sample features if the distribution of high-dimensional data is complex or contains nonlinear feature associations, which in turn leads to inaccurate anomaly detection.

2.  The decomposition and separation constraints of this method require the computation of unique weight vectors for each sample, which may incur significant computational costs on high-dimensional data and large-scale datasets. Also, the introduction of the weight learner increases the model complexity, especially when the number of orthogonal basis vectors is large. The reviewer is concerned about the efficiency of the proposed method. Although authors have shown the runtime in A.5, more comparison with other baselines in terms of runtime will improve the persuasiveness of this aspect.

3. The reproducibility of this work is limited. The reviewer could not validate this work as the source code was not released.

**Questions:**

Please refer to weakness.

---

> ### Author Response · Authors · 2024-11-24
> **Response (Part 1/2)**
>
> W1: Whether the linear decomposition of representations could capture the complex normal distribution.
>
> R to W1: Thanks for your valuable feedback! We would address your concern from the following perspectives.
>
> (1) It is worth noting that for anomaly detection, the key insight is that a higher anomaly score indicates higher confidence in a sample being anomalous. Therefore, even if the decomposition does not perfectly capture the normal sample distribution, as long as it better represents normal samples than anomalous ones, the model can still achieve accurate anomaly detection. Below, we will explain that decomposition is sufficient to express the normal samples and distinguish anomaly from normal samples.
>
> (2) We agree that the true distribution of normal samples in raw data space can be complex, especially in high-dimensional spaces. However, our approach remaps the raw data into a latent space, where we enforce the decomposition of representations. In other words, we learn more expressive, task-specific features from the raw data and perform the decomposition on these learned representations, rather than directly on the raw observations.
>
> (3) To accurately capture the statistical characteristics of normal samples and distinguish them from anomalies, we introduce a set of fixed and shared basis vectors to represent the global structure of the normal data, where the orthogonal basis vectors $\mathcal{B}$ are introduced to enforce the diversity and dependencies among them. Given basis vectors $\mathcal{B}$, each representation $\mathbf{h}$ in this space can be expressed as $\mathbf{w}\mathcal{B}$, where $\mathbf{w}$ is the weight vector (denoting the coordinates) and specific to each representation. We optimize the decomposition loss (Eq.3 in original paper) to expect that all normal samples' representations can reside in such subspace, where we show the convergence of the decomposition loss on the training set in Fig. 18 and 19 in Appendix 10 in revised version.
>
> (4) Besides, as shown in Fig. 12 in Appendix 10 of revised version, we provide the T-SNE visualization of representation $\mathbf{h}$ by feature extractor $f$ and the reconstructed representation $\tilde{\mathbf{h}}$ by linear combination of basis vectors $\mathbf{w}\mathcal{B}$ over all datasets. We can observe a notable overlap between $\mathbf{h}$ and $\tilde{\mathbf{h}}$ of normal samples, and a significant distinction between $\mathbf{h}$ and $\tilde{\mathbf{h}}$ of anomalous samples. This demonstrates that compared to anomalous samples, normal sample representations can be better modeled as a mixture of fixed basis vectors, making it reasonable to distinguish anomaly from normal samples with the decomposition loss (i.e., anomaly score).
>
> (5) To further verify the failure of modeling anomalous representations as a mixture of fixed basis vectors, we provide experiment results in Fig. 17 of Appendix 10 in revised version. We can find that the anomaly score of anomalous samples is significantly larger than that of normal samples, especially when we introduce the separation loss (minimizing the similarity between weight vectors of normal samples).

---

> > ### Author Response · Authors · 2024-11-24
> > **Response (Part 2/2)**
> >
> > W2: The reviewer is concerned about the efficiency of the proposed method. Although authors have shown the runtime in A.5, more comparison with other baselines in terms of runtime will improve the persuasiveness of this aspect.
> >
> > R to W2:
> > Thanks for your valuable suggestion!
> >
> > We agree that we need to compute a unique weight vector for each sample. However, we optimize a shared weight learner to calculate the weight vectors, rather than directly optimizing a unique weight vector for each sample individually. This significantly reduces the computational cost.
> >
> > Besides, we kindly remind you that the default number of basis vectors (i.e., the dimension of the weight vector) is set to 5 across all datasets. We also provide a sensitivity analysis in Fig. 9 of Appendix 6 in the original paper, where the results show that setting the number of basis vectors to 5 is sufficient to achieve high performance. Therefore, this number of basis vectors does not significantly increase the model complexity. Additionally, the weight learner is implemented as a simple two-layer fully connected MLP with Leaky ReLU activation, which does not require a large number of parameters.
> >
> > Following your suggestion, we now include runtime comparisons with other baselines, which is summarized into Table 8 in Appendix (A.5) in our revision. The results show that our proposed DRL method is computationally efficient. For example, among recent baselines, MCM requires the generation of multiple learnable mask matrices, which increases training costs. NPT-AD, on the other hand, involves a high computational cost in terms of memory and time, due to its reliance on the training set during inference.
> >
> > W3: Code needs to be released.
> >
> > R to W3:
> > Thanks for your valuable feedback!
> > Due to time constraints at the time of submission, the code was not available. However, we have now released the source code to facilitate reproducibility. We hope this addresses your concern.

---

> > ### Author Response · Authors · 2024-11-26
> > **A Gentle Reminder**
> >
> > Dear Reviewer Sgt1,
> >
> > Thank you very much again for your time and efforts in reviewing our paper.
> >
> > We kindly remind that our discussion period is closing soon.
> >
> > We just wonder whether there is any further concern and hope to have a chance to respond before the discussion phase ends.
> >
> > Many thanks, Authors

---

> > > ### Comment · Reviewer_Sgt1 · 2024-11-27
> > > **Follow-up feedback**
> > >
> > > I appreciate the authors' response. Their replies addressed my three concerns about this work, so I have updated my rating accordingly.

---

> > > > ### Author Response · Authors · 2024-11-27
> > > > **Response**
> > > >
> > > > Thank you for taking the time to review our response and for your positive feedback.

---

### Official Review · Reviewer_hbkx · 2024-11-03

**Soundness:** 2
**Presentation:** 3
**Contribution:** 3
**Rating:** 5
**Confidence:** 4

**Summary:**

This paper proposes a new method for tabular anomaly detection under the one-class setting via learning decomposed representations of normal samples. The key idea is to decompose normal representations into a weighted linear combination of data-free and training-free orthogonal basis vectors. Furthermore, a separation loss, supported by theoretical analysis, is designed to enhance model’s discriminative ability between normal and anomaly samples. Extensive experiments conducted on 40 datasets demonstrate the effectiveness of the proposed method.

**Strengths:**

1.	This paper is well-written and easy to read.
2.	The proposed decomposed representation learning method sounds reasonable for tabular anomaly detection.
3.	The authors provide a theoretical analysis to support the proposed separation loss.
4.	The comparative experiments conducted on 40 tabular datasets are quite extensive.

**Weaknesses:**

1.	I appreciate that this paper highlights the issue of normal and anomaly samples being entangled in raw space. However, the argument that normal and anomaly latent representations are entangled in reconstruction-based methods, leading to diminished discriminative power, seems somewhat unfounded. Since these models are biased towards reconstructing normal samples, it is expected that representations of the two classes would be entangled in latent space. Based on this, the motivation for learning decomposed normal representations is unclear to me. What are the specific advantages of learning decomposed representations for tabular anomaly detection?
2.	The visualized t-SNE results demonstrate that the proposed method can extract non-entangled normal and anomaly representations. However, the authors may need to explain the rationale for leveraging orthogonal basis vectors to learn such representations. Could vectors with other relational properties achieve the same effect? What are the specific advantages of using orthogonal basis vectors for decomposed representation learning?
3.	I appreciate the authors’ efforts in conducting extensive ablation studies. However, what is the performance when using cosine distance for the decomposition loss? Additionally, how does the model perform if only alignment losses are used as anomaly scores during inference?
4.	The sensitivity analysis of vector number is good. I wonder if the vector number is somehow related to the feature number of input data.
5.	I appreciate the theoretical support for the separation loss. However, as the separation loss is applied between samples, is it sensitive to the batch size? Additionally, if the training normal samples are highly similar to each other, does this loss face convergence challenges?
6.	Can the proposed method be applied to other data types (e.g., image or time series data)? In other words, why is the proposed method specifically useful for tabular data.
7.	This is a minor point, but the paper lacks a discussion on limitations and future work.

**Questions:**

see the weakness part.

---

> ### Author Response · Authors · 2024-11-24
> **Response (Part 1/4)**
>
> W1: The argument that normal and anomaly latent representations are entangled in reconstruction-based methods, leading to diminished discriminative power, seems somewhat unfounded. Since these models are biased towards reconstructing normal samples, it is expected that representations of the two classes would be entangled in latent space. Based on this, the motivation for learning decomposed normal representations is unclear to me. What are the specific advantages of learning decomposed representations for tabular anomaly detection?
>
> R to W1: Thanks for your valuable comment! We are sorry for any confusion.
> We would like to address your concern from two perspectives.
>
> **(1) Why the learned representations may exhibit entanglement in existing methods?** The main objective of anomaly detection in one class classification problem is to model the normal distribution, thus it is expected that the learned normal patterns are distinct from anomalous patterns. As real world data may exhibit observation entanglement between normal and anomalous samples (please refer to Fig. 7 of Appendix A.1 in original paper), using the reconstruction loss on the observed samples to distinguish the anomalous samples from the normal samples might be inefficient.
> Ignoring observation entanglement in tabular anomaly detection under the one-class classification setting can lead to diminished discriminative power between learned normal and anomalous patterns, as the overlap between normal and anomalous representations within the latent space of deep models may obscure the distinction between them, which is illustrated in (c) and (g) of Fig. 1 in original paper.
> We provide additional demonstrations of this phenomenon in Fig. 14 and 15 in Appendix 10 in revised version.
> We attribute this challenge to the intrinsic heterogeneity of features in tabular data, which aligns with recent findings [1] indicating that neural networks struggle to distinguish regular and irregular patterns, particularly when faced with a large number of uninformative features in tabular data.
> This is one of the reasons why distinguishing normal patterns from anomalous ones remains a challenging task for existing reconstruction-based methods.
>
>
> **(2) What are the advantages of learning decomposed representations for tabular anomaly detection?**
> The main objective of decomposed representation learning (DRL) is also to model the normal distribution during training.
> Considering the entanglement issue discussed above, we aim to capture the normal patterns in a constrained latent space, where the normal and anomalous patterns are enforced to be more discriminative.
> The advantage of learning decomposed representations lies in our ability to capture the shared statistical information within normal patterns, which helps distinguishes them from anomalies.
> Normal samples, which are drawn from the same distribution, are considered to represent the normal state. Thus, it is reasonable to assume that these normal samples share common statistical properties that distinguish them from anomalies.
> Inspired by techniques from dictionary learning, topic modeling, we can learn the shared information by enforcing that each normal sample's representation is decomposed into a linear combination of shared basis vectors (analogous to topics in topic modeling [2]) with sample-specific weights (analogous to topic proportion in topic modeling).
> Meanwhile, the separation constraint enforces the normal and anomalous patterns to be more discriminative, thereby facilitating the capturing of shared information within normal patterns.
>
> [1] Why do tree-based models still outperform deep learning on typical tabular data? [NeurIPS 2022]
>
> [2] A review of topic modeling methods. [Information Systems 2020]

---

> > ### Author Response · Authors · 2024-11-24
> > **Response (Part 2/4)**
> >
> > W2: The authors may need to explain the rationale for leveraging orthogonal basis vectors to learn such representations. Could vectors with other relational properties achieve the same effect? What are the specific advantages of using orthogonal basis vectors for decomposed representation learning?
> >
> > R to W2: Thanks for your valuable suggestion! We are sorry for any confusion and will explain the advantages in detail.
> >
> > (1) To accurately capture the statistical characteristics of normal samples while distinguishing them from anomalies, it is crucial that the shared basis vectors are sufficiently diverse to encapsulate the global structure of the normal data.
> > To this end, we eliminate the dependencies among basis vectors by leveraging a set of orthogonal vectors as basis vectors.
> >
> > (2) Given a subspace defined by a set of orthogonal basis vectors, each representation in this space can be expressed as $\mathbf{w}\mathcal{B}$, where $\mathbf{w}$ is the weight vector (denoting the coordinates) and $\mathcal{B}$ represents the fixed set of basis vectors.
> > To increase the discrepancy between different representations, as discussed in Sec. 3.2.2, we only need to enforce the separation in the weight vectors $\mathbf{w}$ due to the shared basis vectors, i.e., the separation loss (Eq.5 in original paper). It is easier than directly increasing the discrepancy between the representations  since we already exclude the shared information for all representations and the dimension of weight vector is extremely low (only 5 for all datasets).
> >
> > (3) Additionally, we have provided a comparison between using random basis vectors and orthogonal basis vectors in Table 1 of the original paper. From our experiments, we observe that the orthogonalization of basis vectors is crucial for effectively capturing normal patterns that distinguish them from anomalies.
> >
> >
> >
> > W3: What is the performance when using cosine distance for the decomposition loss? Additionally, how does the model perform if only alignment losses are used as anomaly scores during inference?
> >
> > R to W3: Thanks for your insightful suggestion! Following your suggestion, we include experiment results as follow to verify the performance of using cosine distance for the decomposition loss and using alignment loss as anomaly score for inference. We have also included these results in Table 2, Table 3 in main text and Table 11 to 14 in Appendix 10 in revised version.
> > We could observe that using cosine distance for the decomposition loss achieves comparable performance to using L2 distance (ours). During inference, DRL only uses decomposition loss as anomaly score as detailed in Section 3.2.3 in original paper.
> > When using alignment loss as the anomaly score for inference, we observed a degradation in performance. This is because the alignment loss is calculated in the observation space, which is prone to the observation entanglement issue discussed in Section 3.1 of the original paper. As a result, it might be inefficient to use alignment loss to distinguish the anomalous samples from normal ones.
> >
> >
> > |                                  | Backdoor        | Fault           | Imgseg          | Lympho          | Pendigits       | Vowels          | Wbc             | Average of 40 data |
> > |----------------------------------|-----------------|-----------------|-----------------|-----------------|-----------------|-----------------|-----------------|--------------------|
> > | Decomposition w/ cosine distance | 0.8808          | 0.6534          | 0.9036          | 0.9868          | 0.9029          | 0.4307          | 0.9417          | 0.7130             |
> > | Inference w/ alignment loss      | 0.8787          | 0.6059          | 0.8318          | 0.8278          | 0.7157          | 0.3160          | 0.9111          | 0.6100             |
> > | **DRL (ours)**              | **0.8915** | **0.6649** | **0.9238** | **1.0000** | **0.9360** | **0.4506** | **0.9742** | **0.7344**    |

---

> > ### Author Response · Authors · 2024-11-24
> > **Response (Part 3/4)**
> >
> > W4: The sensitivity analysis of vector number is good. I wonder if the vector number is somehow related to the feature number of input data.
> >
> > R to W4:
> > Thank you for your valuable suggestion! We agree that the number of basis vectors may have some relationship with the number of features in the input data. Considering the feature number when constructing the set of basis vectors is indeed a meaningful idea, and we see this as an important direction for our future work.
> >
> > In our DRL, we remap the sample from raw space to the latent space and enforce the representation of each normal sample in the latent space to be decomposed into a weighted linear combination of randomly generated orthogonal basis vectors. The hidden dimension for representation is fixed across all datasets, thus it is reasonable to use a default number of basis vectors for each dataset.
> > In our experiments, we set the default number of basis vectors to 5 for all datasets. We also provide a sensitivity analysis in Fig. 9 of Appendix 6 in the original paper, where we show that setting the number of basis vectors to 5 is sufficient to achieve high performance.
> >
> > W5: I appreciate the theoretical support for the separation loss. However, as the separation loss is applied between samples, is it sensitive to the batch size? Additionally, if the training normal samples are highly similar to each other, does this loss face convergence challenges?
> >
> > R to W5: Thanks for your valuable suggestion!
> >
> > (1) Following your suggestion, we include the sensitivity analysis w.r.t. batch size on AUC-PR, which is added into Fig. 9 (e) of Appendix (A.6) and summarized in table below for your convenience. The results demonstrate that the performance remains robust across different batch sizes.
> >
> > (2) Regarding loss convergence, the primary objective of DRL is to minimize the decomposition loss (Eq. 3 in the original paper), while the separation loss (Eq. 5 in the original paper) serves as an additional constraint. The separation loss is applied to the normal weights using cosine distance, ensuring that the values remain bounded within a small range. Moreover, the loss weight for the separation loss is set to 0.06 by default, which further constrains its range. Additionally, the weights for basis vectors belong to a probability simplex, which prevents cases where all weights become zero during updates. These mechanisms collectively contribute to stable loss convergence.
> > To verify this, we have added the experimental results in Fig. 18 and 19 in Appendix 10 in the revised version, illustrating the effect of the separation constraint on loss convergence. The results confirm that the separation loss does not negatively impact convergence.
> >
> >
> > | Batch size    | Backdoor        | Fault           | Imgseg          | Lympho     | Pendigits      | Vowels          | Wbc             | Average of 40 data |
> > |---------------|-----------------|-----------------|-----------------|------------|----------------|-----------------|-----------------|--------------------|
> > | 32            | 0.8692          | 0.65            | 0.909           | 0.9879     | 0.8671         | 0.4312          | **0.9762** | 0.7129             |
> > | 64            | 0.8809          | 0.6321          | 0.926           | 0.9972     | 0.8586         | 0.4465          | 0.9627          | 0.7149             |
> > | 128           | 0.8885          | **0.6699** | **0.9292** | **1** | 0.9012         | 0.4483          | 0.9742          | 0.7295             |
> > | 256           | 0.8718          | 0.639           | 0.9153          | **1** | 0.8931         | 0.4234          | 0.9742          | 0.7167             |
> > | 512 (Default) | **0.8915** | 0.6649          | 0.9238          | **1** | **0.936** | 0.4506          | 0.9742          | **0.7344**    |
> > | 1024          | 0.88            | 0.6006          | 0.9287          | **1** | 0.9237         | **0.4674** | 0.9742          | 0.7249             |

---

> > ### Author Response · Authors · 2024-11-24
> > **Response (Part 4/4)**
> >
> > W6: Can the proposed method be applied to other data types (e.g., image or time series data)?
> >
> > R to W6: Thank you for your insightful question!
> >
> > As classical methods struggle to capture complex patterns and relationships in high-dimensional spaces [1], recent studies have prompted a shift toward deep learning methodologies for tabular data.
> > Although fruitful progress has been made in the last several years, capturing the comprehensive normal patterns that are distinct from anomalous patterns for tabular data remains a challenging task, as real-world data may exhibit data entanglement between normal and anomalous samples.
> > Ignoring observation entanglement in tabular anomaly detection under the one-class classification setting can lead to diminished discriminative power between learned normal and anomalous patterns.
> > We attribute this challenge to the intrinsic heterogeneity of features in tabular data, which aligns with recent findings [2] indicating that neural networks struggle to distinguish regular and irregular patterns, particularly when confronted with numerous uninformative features present in tabular data.
> > Our method re-maps observations into a tailor-designed constrained latent space, where normal and anomalous patterns are more effectively distinguished, thereby alleviating the entanglement problem.
> >
> > In addition, when considering perceptual data (e.g., image and text), many methods have demonstrated significant success by leveraging the structure of the input data.
> > For example, images can be rotated, and the ability to distinguish between different rotations varies between anomalies and normal samples.
> > However, tabular data lacks such prior structural properties, and our DRL does not rely on prior knowledge of the data structure, making it especially effective for tabular data.
> >
> > We believe that applying the insights from DRL to other data types is meaningful and it will be an important direction of our future work.
> >
> > [1] Deep learning for anomaly detection: A review. [ACM Computing Surveys 2021]
> >
> > [2] Why do tree-based models still outperform deep learning on typical tabular data? [NeurIPS 2022]
> >
> >
> > W7: This is a minor point, but the paper lacks a discussion on limitations and future work.
> >
> > R to W7: Thank you for your valuable suggestion! We agree that discussing limitations and future work is important. Although our method can be used to different data types for the anomaly detection task by introducing representation decomposition, we may need to design the specific architecture for the weight learner and alignment learner due to the difference between data types. In the future, we plan to explore its application to other data types, where incorporating prior structural knowledge from these data types might be a possible solution. We have added the discussion on the limitations and future work in Appendix 11 in our revision.

---

> > ### Author Response · Authors · 2024-11-26
> > **A Gentle Reminder**
> >
> > Dear Reviewer hbkx,
> >
> > Thank you very much again for your time and efforts in reviewing our paper.
> >
> > We kindly remind that our discussion period is closing soon.
> >
> > We just wonder whether there is any further concern and hope to have a chance to respond before the discussion phase ends.
> >
> > Many thanks, Authors

---

> > > ### Comment · Reviewer_hbkx · 2024-11-27
> > >
> > > I appreciate that the authors address most of my concerns, but my main issue W1 remains unresolved. To me, empirical observations in the latent space fail to adequately explain the advantages of the proposed method, as reconstruction-based methods are expected to produce entangled latent features of normal and anomaly classes. Based on this, I am inclined to keep my previous rating. I will further discuss with the peer reviewers for a final recommendation.

---

> ### Author Response · Authors · 2024-11-27
> **Further explanation on W1**
>
> We sincerely appreciate your feedback! There might be some misunderstanding and we would like to further explain our motivation.
>
> We agree with you that for existing reconstruction-based methods, the representations of the two classes would be entangled in latent space since these models are biased towards reconstructing normal samples. We also agree that suppose these representations are already learned, further decomposing them cannot help with the OOD detection performance.
>
> In this work, we actually aim to learn less entangled latent representations without seeing anomaly samples. The way we achieve this includes two aspects. Firstly, we enforce the latent representations of normal observations to a constrained latent space defined by a set of orthogonal and fixed basis vectors, where each representation is decomposed into a weighted combination of the basis vectors (the weight vector is on a probability simplex). Besides, we enforce normal representations to span this constrained subspace by introducing the separation loss (minimizing the similarity between weight vectors of normal samples). Intuitively, the representation of an anomaly sample will be 'squeezed out' from the subspace. That is why our learned encoder can distinguish anomaly samples from normal samples better than the encoder of others.
> Therefore, the representation learning in ours is quiet different from existing reconstruction-based methods. We will make this motivation more clearer in our camera ready.
>
> We now give empirical evidence to illustrate the representation separation between the two classes.
> Let $\mathbf{w}_n$ and $\mathbf{w}_a$ denote the computed weights of normal and anomalous samples respectively.
> Fig. 16 in Appendix 10 of revised version indicates that as the training progresses, $\text{Var}(\|\mathbf{w}_n\|_2)$ increases, which leads to a corresponding increase in $\text{E}\left[\|\mathbf{w}_n - \mathbf{w}_a\|_2^2\right]$. This verifies that raising the lower bound effectively enlarges $\text{E}\left[\|\mathbf{w}_n - \mathbf{w}_a\|_2^2\right]$. In addition, Table 10 in Appendix 10 in revised version illustrates that when the separation loss is applied,
> the average distance of anomalous weight $\mathbf{w}_a$ from the center of normal weights, i.e., $\mu _ {\mathbf{w}_n} = \text{E}\left[\mathbf{w}_n\right]$, has a more greater growth than the average distance of normal weight $\mathbf{w}_n$ from $\mu _ {\mathbf{w}_n}$.
> We further visualize the T-SNE of the learned representations of encoder w/o and w/ separation loss over all datasets in Fig.13 in Appendix 10 of revised version to verify the representation separation. We can observe that with separation loss, the discriminative distinction between normal and anomalous patterns within the latent space can be enhanced.
> This supports the sufficiency of the constraint that anomalous representations are enforced to be distinct from normal representations.
>
> Therefore, in such a constrained subspace, we could facilitate the capturing of shared information within normal patterns that distinct from anomalous ones.

---

> ### Author Response · Authors · 2024-12-02
>
> Dear Reviewer hbkx,
>
> The discussion period will end soon (Dec 2nd), you have raised some further question on motivation, which we have provided further explanations. We want to check if our response has addressed your questions and concerns.  We also noticed that other reviewers have updated their ratings (Reviewer hqNX has increased the score from 5 to 6, Reviewer Sgt1 has increased the score from 5 to 6, and Reviewer TJDb has increased the score from 3 to 6). If necessary, please feel free to provide any additional feedback or ask further questions. Again, thank you for the time spent on reviewing and discussing the manuscript.

---

### Official Review · Reviewer_hqNX · 2024-11-03

**Soundness:** 2
**Presentation:** 2
**Contribution:** 3
**Rating:** 6
**Confidence:** 4

**Summary:**

This paper introduces Decomposed Representation Learning (DRL), a novel approach for addressing the entanglement issue in representation learning within reconstruction-based Tabular Anomaly Detection (TAD). DRL mitigates this problem by employing orthogonal bases in a training-free, data-free manner to remove latent space dependencies, implemented through the Gram-Schmidt process for Decomposition Loss. It also maximizes variance in the weights of normal latent bases, enhancing discrepancy with abnormal latents as supported by Proposition 1, and reconstructs the original input from latent representations to prevent task-irrelevant features. The paper validates DRL’s effectiveness with extensive experiments across 40 ADBench datasets and 15 baseline models, achieving state-of-the-art performance through detailed ablation studies and T-SNE visualizations.

**Strengths:**

1. The paper effectively addresses limitations in reconstruction-based methods by comparing DRL with recent state-of-the-art models such as NPT-AD and MCM, demonstrating its superior performance. This is a novel contribution to the field of tabular anomaly detection.
2. Theoretical support for the separation loss, specifically Proposition 1, provides a solid basis for the model's structure and validates its design choices.
3. Extensive experimentation across 40 ADBench datasets and 15 baseline models showcases the robustness and effectiveness of the DRL method, and ablation studies and T-SNE visualizations add valuable insights into its workings.

**Weaknesses:**

1. Lack of KNN Baseline Comparison: While the paper includes comparisons with several state-of-the-art methods, there is no performance comparison with KNN, a model often effective in tabular data tasks. Including KNN as a baseline would enhance the experimental section by providing insights into how DRL performs relative to a widely recognized tabular anomaly detection approach. I recommend the authors consider adding KNN results and explain how DRL’s design is particularly beneficial over KNN in anomaly detection.

2. Loose Bound in Separation Loss Calculation: In calculating the separation loss lower bound, the authors remove all terms involving squared expectations, which might lead to a relatively loose bound. It would be helpful if the authors could explore alternative derivations for this bound that might be tighter or justify the sufficiency of the current approach. Additionally, it would be useful to discuss how this looseness could potentially impact performance or theoretical guarantees of the DRL method.

3. Clarity in Section 2 and Lack of MCM Limitations: Some portions of the paper, particularly Section 2 (Preliminaries), could benefit from clearer explanations. For instance, a more detailed explanation of theta would aid readers in following the theoretical framework more easily. Additionally, while the authors discuss the disadvantages of NPT-AD, the limitations of MCM are not mentioned. Including a brief discussion on MCM's limitations would help balance the comparison and highlight DRL's unique advantages more clearly.

**Questions:**

1. What would happen if the basis vectors were set as unit vectors, for example, [1,0,0,0],[0,1,0,0],[0,0,1,0],[0,0,0,1] ?
2. The paper states that cosine distance improves performance as L2 distance leads to optimization issues. Could the authors explore how performance might change with L1 distance, or clarify why cosine distance was chosen specifically over both L2 and L1?
3. Proposition 1 proposes that maximizing variance of weights attached to basis vectors in normal latents enhances separation from abnormal latents. Are there trade-offs in terms of model stability or convergence when applying this approach?
4. How sensitive is DRL’s performance to hyperparameter choices, such as the number of orthogonal basis vectors or weights in loss terms? Are there specific recommendations or guidelines for tuning these parameters across different types of tabular datasets?
5. What are the main limitations of DRL as identified by the authors?

---

> ### Author Response · Authors · 2024-11-24
> **Response (Part 1/4)**
>
> W1: Lack of KNN Baseline Comparison.
>
> R to W1: Thank you for your valuable suggestion! In the revised version of the paper, we have included KNN as a baseline for comparison; please see Fig. 3, Fig. 4 of main text and Table 8 of Appendix 5 in revised manuscript.
> The advantage of DRL's design over KNN: Despite the effectiveness of KNN, it computes distances between samples in the observation space, where real-world data often exhibit entanglement between normal and anomalous samples as shown in Fig.1 in main text and Fig.7 of Appendix 1.
> Ignoring this entanglement in tabular anomaly detection, particularly in the one-class classification setting, usually result in reduced discriminative power between learned normal and anomalous patterns. Our proposed DRL alleviates this issue by introducing a decomposed representation learning framework.
>
> W2: The authors need to explore tighter lower bound or justify the sufficiency of the separation loss; additionally, it would be useful to discuss how it could potentially impact performance.
>
> R to W2: Thanks for your valuable suggestion!
>
> **(1) Exploring tighter lower bound.** We revise the derivations for this bound, and it is now tighter than the original ones, which is illustrated in Proposition 1 in the revised version. Below, we would clarify the sufficiency of separating normal and anomalous patterns and the impact on performance when applying this separation.
>
> **(2) The sufficiency of separating normal and anomalous patterns.** We have included experimental results in Fig. 16 and Table 10 in Appendix 10 of revised version to illustrate that increasing $\text{Var}(\|\mathbf{w}_n\|_2)$ in Proposition 1 can amplify the discrepancy between the normal and anomalous patterns. Let $\mathbf{w}_a$ and $\mathbf{w}_n$ denote the computed weights of anomalous and normal samples respectively. Specifically, Fig. 16 indicates that as the training progresses, $\text{Var}(\|\mathbf{w}_n\|_2)$ increases, which leads to a corresponding increase in $\text{E}\left[\|\mathbf{w}_n - \mathbf{w}_a\|_2^2\right]$. This verifies that raising the lower bound effectively enlarges $\text{E}\left[\|\mathbf{w}_n - \mathbf{w}_a\|_2^2\right]$. In addition, Table 10 illustrates that when the separation loss is applied,
> the average distance of anomalous weight $\mathbf{w}_a$ from the center of normal weights, i.e., $\mu _ {\mathbf{w}_n} = \text{E}\left[\mathbf{w}_n\right]$, has a more greater growth than the average distance of normal weight $\mathbf{w}_n$ from $\mu _ {\mathbf{w}_n}$. This supports the sufficiency of the proposed separation mechanism.
>
> **(3) The impact on performance of this separation.**
> It is worth noting that for anomaly detection, the key insight is that a higher anomaly score indicates higher confidence in a sample being anomalous. Therefore, even if the decomposition does not perfectly capture the normal sample distribution, as long as it better represents normal samples than anomalous ones, the model can still achieve accurate anomaly detection.
> Additionally, we provide the T-SNE visualization of representation $\mathbf{h}$ by feature extractor $f$ and the reconstructed representation $\tilde{\mathbf{h}}$ by linear combination of basis vectors $\mathbf{w}\mathcal{B}$ over all datasets, as shown in Fig. 12 in Appendix 10 of revised version.
> We can observe a notable overlap between $\mathbf{h}$ and $\tilde{\mathbf{h}}$ of normal samples, and a significant distinction between $\mathbf{h}$ and $\tilde{\mathbf{h}}$ of anomalous samples. This demonstrates that compared to anomalous samples, normal sample representations can be better modeled as a mixture of fixed basis vectors.
> To further verify the failure of modeling anomalous representations as a mixture of fixed basis vectors, we provide experiment results in Fig. 17 of Appendix 10 in revised version to show that the anomaly score (calculated by decomposition loss in Eq.3 in original paper) of anomalous samples is significantly larger than that without separation loss.
> This explains why separating the normal and anomalous patterns can result in performance improvement.

---

> > ### Author Response · Authors · 2024-11-24
> > **Response (Part 2/4)**
> >
> > W3: Unclear description of Section 2 and MCM limitations.
> >
> >
> > R to W3: Thanks for your insightful suggestion! We apologize for any confusion and we have made improvements to the clarity of Section 2 in revised version for better illustration.
> >
> > (1) Standard reconstruction-based approaches consider learning a mapping $A(\cdot;\Theta):\text{R}^D \to \text{R}^D$ to minimize the reconstruction loss within $\mathcal{D}_{train}$, where $D$ is the number of input features. Typically, $A(\cdot;\Theta)$ first maps the sample from observation space to latent space, and then maps it back to the observation space to obtain the reconstruction of the sample. The parameters $\Theta$ are optimized by minimizing the reconstruction loss on  normal training samples.
> >
> > (2) Based on reconstruction, MCM employs a learnable masking strategy to the input and aims to reconstruct normal samples well with access to only unmasked entries of the input, where how to produce  effective masks is challenging in this field.
> > NPT-AD incorporates both feature-feature and sample-sample dependencies to reconstruct masked features of normal samples by utilizing the whole training set. Therefore, NPT-AD involves a high computational cost in terms of memory and time, due to its reliance on the training set during inference.
> > Despite the effectiveness of MCM and NPT-AD, both of them design the reconstruction strategies in the observed data space, as illustrated in Section 3.1 in original paper, which usually suffer from the potential data entanglement in observed data space, where we showed the phenomenon on more real world tabular datasets in Fig. 7 of Appendix A.1.
> > Ignoring the issue in anomaly detection can lead to diminished discriminative power between normal and anomalous patterns. To solve the potential data entanglement issue, our proposed DRL introduces representation decomposition in the constrained latent space, where the normal and anomalous patterns are more discriminative. Additionally, DRL exhibits lower computational costs compared to both MCM and NPT-AD as illustrated in Table 8 in Appendix 5 of revised version.
> >
> >
> > Q1: What would happen if the basis vectors were set as unit vectors, for example, [1,0,0,0],[0,1,0,0],[0,0,1,0],[0,0,0,1]?
> >
> > R to Q1: Thanks for your comment! Following your suggestion, we include experiments as follows to evaluate the impact of using unit vectors as basis vectors in DRL.
> > The results indicate that initializing with unit vectors does not perform as well as the default basis vector generation method in DRL.
> > In our approach, the reconstructed representation is computed as a linear combination of orthogonal basis vectors: $\tilde{\mathbf{h}} = \sum_{k=1}^Kw^k\beta_k$, where the basis vectors are defined by $\mathcal{B} = \\{\beta_k\\}_{k=1}^K \in \text{R}^{K\times E}$ with $K < E$. By default, the $K$ and $E$ is set to 5 and 128 respectively. If the basis vectors were set as unit vectors, the reconstructed representation $\tilde{\mathbf{h}}$ would become highly sparse, containing $E-K$ zero entries in its $E$-dimensional space. The main objective of DRL is to minimize the decomposition loss $d(\mathbf{h}, \tilde{\mathbf{h}})$ (Eq.3), where $d$ is the distance measurement and $\mathbf{h}$ is the representation extracted by feature extractor $f$. However, this configuration could lead to inefficient optimization, as $\mathbf{h}$ would need to be overly sparse to match $\tilde{\mathbf{h}}$.
> >
> >
> > |                     | Backdoor        | Fault           | Imgseg          | Lympho          | Pendigits       | Vowels          | Wbc             | Average of 40 data |
> > |---------------------|-----------------|-----------------|-----------------|-----------------|-----------------|-----------------|-----------------|--------------------|
> > | Unit vector         | 0.6794          | 0.6175          | 0.9143          | 0.9484          | 0.8327          | 0.4381          | 0.8911          | 0.6688             |
> > | **DRL (ours)** | **0.8915** | **0.6649** | **0.9238** | **1.0000** | **0.9360** | **0.4506** | **0.9742** | **0.7344**    |

---

> > ### Author Response · Authors · 2024-11-24
> > **Response (Part 3/4)**
> >
> > Q2: How performance might change with L1 distance, and clarify why cosine distance was chosen specifically over both L2 and L1.
> >
> > R to Q2: Thank you for your thoughtful comment! We have added results using the L1 distance metric for comparison as follows.  We also included the results in Table 3 in main text, Table 13 and Table 14 in Appendix 10 of revised version. The results show that L1, L2, and cosine distances all yield promising performance, with cosine distance demonstrating relatively superior and more stable results.
> > In the original paper, we used cosine distance for both separation and alignment loss. We now provide further clarification on why cosine distance was chosen over both L1 and L2 distances.
> >
> > Our primary objective is to minimize the decomposition loss, which also serves as the anomaly score during inference. Separation loss acts as a constraint to enhance the discriminative power between normal and anomalous patterns within the latent space, thereby helping to better capture the distinct normal patterns through decomposition loss.
> > Since separation loss is implemented by enforcing separation among the weights of normal training samples, it has the potential to affect the convergence of the training process. To mitigate this, we use cosine distance for separation loss, as it ensures the values are bounded within a smaller range compared to both L1 and L2 distances, which helps avoid convergence issues during training.
> >
> > As for alignment loss, considering the potential entanglement between normal and anomalous samples in the observed data space, unlike previous methods minimizing the L2 distance between $\mathbf{x}$ and its reconstructed version $\tilde{\mathbf{x}}$ to accurately maintain the observation information, our strategy focuses on maximizing the cosine similarity between $\mathbf{x}$ and $\tilde{\mathbf{x}}$ to align the information of $\mathbf{h}$ with the intrinsic feature correlation of $\mathbf{x}$, while avoiding excessive retention of observational details that may contain entanglement patterns.
> >
> > |                                                          | Backdoor        | Fault           | Imgseg          | Lympho          | Pendigits       | Vowels          | Wbc             | Average of 40 data |
> > |----------------------------------------------------------|-----------------|-----------------|-----------------|-----------------|-----------------|-----------------|-----------------|--------------------|
> > | Separation w/ L1 distance                                | 0.8784          | 0.6391          | 0.9185          | 0.8391          | 0.882           | 0.44            | 0.9589          | 0.7007             |
> > | Alignment w/ L1 distance                                 | 0.8868          | 0.6433          | 0.9004          | 0.9762          | 0.9218          | 0.3696          | 0.9401          | 0.7063             |
> > | Separation w/ L2 distance                                | 0.8786          | 0.6444          | 0.8998          | 0.8900          | 0.8735          | 0.4635          | 0.9655          | 0.7080             |
> > | Alignment w/ L2 distance                                 | 0.8886          | 0.6576          | 0.9125          | 1.0000          | 0.9090          | 0.4425          | 0.9590          | 0.7134             |
> > | **Separation, Alignment w/ Cosine distance (ours)** | **0.8915** | **0.6649** | **0.9238** | **1.0000** | **0.9360** | **0.4506** | **0.9742** | **0.7344**    |
> >
> >
> > Q3: Proposition 1 proposes that maximizing variance of weights attached to basis vectors in normal latents enhances separation from abnormal latents. Are there trade-offs in terms of model stability or convergence when applying this approach?
> >
> > R to Q3: Thanks for your valuable comment! Regarding loss convergence, the primary objective of DRL is to minimize the decomposition loss (Eq. 3 in the original paper), while the separation loss (Eq. 5 in the original paper) serves as an additional constraint. The separation loss is applied to the normal weights using cosine distance, ensuring that the values remain bounded within a small range. Moreover, the loss weight for the separation loss is set to 0.06 by default, which further constrains its range. Additionally, the weights belong to a probability simplex, which prevents cases where all weights become zero during updates. These mechanisms collectively contribute to stable loss convergence.
> > To verify this, we have included experimental results in Fig. 18 and 19 in Appendix 10 in the revised version, illustrating the effect of the separation constraint on loss convergence. The results confirm that the separation constraint does not negatively impact convergence.

---

> > ### Author Response · Authors · 2024-11-24
> > **Response (Part 4/4)**
> >
> > Q4: How sensitive is DRL's performance to hyperparameter choices, such as the number of orthogonal basis vectors or weights in loss terms? Are there specific recommendations or guidelines for tuning these parameters across different types of tabular datasets?
> >
> > R to Q4: Thanks for your insightful comment! We apologize for any confusion. Actually, as mentioned in the last paragraph of the section about the experiments in the original version, we had provided the sensitivity analysis results in the Fig. 9 in Appendix (A.6) due to space limitations, including the sensitivity analysis for the number of basis vectors, the hyper-parameters about loss weight, the number of training epochs and the number of batch size. Below, we will give a more detailed explanation of the sensitivity analysis.
> >
> > As the very beginning, the performance increases rapidly as the number of orthogonal basis vectors increases and then the performance is stable. Thus, it is sufficient to set the number of orthogonal basis vectors to 5. For the loss weight $\lambda_1$ associated with separation loss, we find that the performance is generally robust across different values, but lower values of $\lambda_1$ tend to yield relatively better results.
> > Therefore, a lower $\lambda_1$ is more suitable.
> > Similarly, for the loss weight $\lambda_2$ associated with alignment loss, we observe that lower values of $\lambda_2$ lead to better AUC-ROC performance. Thus, a lower $\lambda_2$ is more suitable.
> > Regarding the number of training iterations, we found that the performance stabilizes after 200 iterations, so it is sufficient to set the total training iterations to 200.
> > The results also demonstrate that the performance remains robust across different batch sizes.
> > As noted in Line 372 of the original paper, the hyperparameters were kept consistent across all datasets.
> >
> > Q5: What are the main limitations of DRL as identified by the authors?
> >
> > R to Q5: The primary limitation of the current DRL method is that it is designed specifically for tabular anomaly detection. Although our method can be used to different data types for the anomaly detection task by introducing representation
> > decomposition, we may need to design the specific architecture for the weight learner and alignment learner due to the difference between data types. In the future, we plan to explore its application to other data types, where incorporating prior structural knowledge from these data types might be a possible solution. We have added the discussion on the limitations and future work in Appendix 11 in our revision.

---

> ### Author Response · Authors · 2024-11-26
> **A Gentle Reminder**
>
> Dear Reviewer hqNX,
>
> Thank you very much again for your time and efforts in reviewing our paper.
>
> We kindly remind that our discussion period is closing soon.
>
> We just wonder whether there is any further concern and hope to have a chance to respond before the discussion phase ends.
>
> Many thanks, Authors

---

> ### Author Response · Authors · 2024-12-02
>
> Dear Reviewer hqNX,
>
> The discussion period will end soon (Dec 2nd), we want to check if our response has addressed your questions and concerns regarding our paper. Please let us know if you have any follow-up comment or question regarding our manuscript.  We also noticed that other reviewers have updated their ratings (Reviewer Sgt1 has increased the score from 5 to 6, and Reviewer TJDb has increased the score from 3 to 6). If necessary, please feel free to provide any additional feedback or ask further questions. Again, thank you for the time spent on reviewing and discussing the manuscript.

---

> ### Author Response · Authors · 2024-12-02
> **Response**
>
> Thank you for kindly increasing the score and for taking the time to review our response.

---

### Meta-Review · Area_Chair_Aoqt · 2024-12-19

**Metareview:**

Based on the reviews, I recommend accepting the paper for its high technical and empirical quality, as well as its significance to the ICLR community. The paper received four reviews, three of which recommend acceptance with high confidence. Notably, one reviewer provided an exceptionally detailed review, offering many constructive suggestions and technical comments. Another reviewer rated the paper a 5 but acknowledged that the authors’ detailed rebuttal effectively addressed six out of the seven reported weaknesses. As an expert in this topic, I find that the remaining concern is unfounded, with sufficient evidence provided in parts (c) and (g) of Fig. 1 and Figs. 14 and 15 in Appendix 10 to disprove it convincingly.

**Additional Comments On Reviewer Discussion:**

The main reviewer concerns centered on technical aspects and the key assumptions underlying the proposed method:

- **Reviewer hqNX** questioned the lack of important baselines, a loose bound in the separation loss, and a missing discussion on the limitations of the method. The authors’ detailed rebuttal addressed these points, leading the reviewer to increase their rating to 6.

- **Reviewer hbkx** raised doubts about the motivation for decomposing normal representations and the absence of discussion on limitations and future work. Despite added experimental results and discussions, the reviewer remained unconvinced, maintaining a rating below the acceptance threshold.

- **Reviewer Sgt1** expressed concerns about decomposing normal samples into orthogonal bases, high-dimensional data challenges, computational costs, and reproducibility. The authors addressed these concerns effectively, resulting in a rating increase to 6.

- **Reviewer TJDb** initially questioned the theoretical basis and performance of the method. The authors provided additional statistics and results, leading the reviewer to revise their rating from 3 to 6.

In conclusion, the authors’ comprehensive rebuttals resolved most concerns. Despite continued reservations of Reviewer hbkx, the overall feedback was positive, with the paper’s empirical strength and detailed responses being key factors in recommending acceptance.

---

### Decision · Program_Chairs · 2025-01-22

Accept (Poster)